# Behavioral decomposition reveals rich encoding structure employed across neocortex in rats

Bartul Mimica [1,4] ✉, Tuçe Tombaz[2,4], Claudia Battistin [2,3],
Jingyi Guo Fuglstad[2], Benjamin A. Dunn [2,3] & Jonathan R. Whitlock [2] ✉

The cortical population code is pervaded by activity patterns evoked by movement, but it remains largely unknown how such signals relate to natural behavior or how they might support processing in sensory cortices where they have been observed. To address this we compared high-density neural recordings across four cortical regions (visual, auditory, somatosensory, motor) in relation to sensory modulation, posture, movement, and ethograms of freely foraging male rats. Momentary actions, such as rearing or turning, were represented ubiquitously and could be decoded from all sampled structures. However, more elementary and continuous features, such as pose and movement, followed region-specific organization, with neurons in visual and auditory cortices preferentially encoding mutually distinct head-orienting features in world-referenced coordinates, and somatosensory and motor cortices principally encoding the trunk and head in egocentric coordinates. The tuning properties of synaptically coupled cells also exhibited connection patterns suggestive of area-specific uses of pose and movement signals, particularly in visual and auditory regions. Together, our results indicate that ongoing behavior is encoded at multiple levels throughout the dorsal cortex, and that low-level features are differentially utilized by different regions to serve locally relevant computations.

Our knowledge of sensory and motor cortical processing is founded largely on approaches in which neural systems are studied in isolation[1–5], using laboratory tasks where animals perform a priori defined subsets of behaviors in response to experimenter-defined stimuli[6]. While such approaches bring essential reliability and control, they also restrict the scope of actions animals can express, which limits understanding of the broader array of features to which these systems respond when animals engage in natural behaviors[7–9]. This knowledge gap is underscored by observations in head-fixed animals showing that self-generated movements, independent of behavioral tasks, profoundly affect cortical activity patterns[10,11], including in primary visual cortex, with animals in darkness and under no explicit cognitive burden[12]. Great strides have been made in the last decade linking self-generated movements with gain modulation and predictive processing in sensory cortices[13–20], but sensory systems are rarely studied in tandem, and most systematic progress has been achieved by exploiting the advantages of head-fixed preparations. Consequently, it is not well understood how movement-associated signals reflect the animals' natural behavioral repertoire or if the features encoded vary by region to support local forms of processing.

[1]Princeton Neuroscience Institute, Princeton University, Washington Road, Princeton 100190 NJ, USA. [2]Kavli Institute for Systems Neuroscience, Norwegian University of Science and Technology, Olav Kyrres Gate 9, 7030 Trondheim, Norway. [3]Department of Mathematical Sciences, Norwegian University of Science and Technology, 7491 Trondheim, Norway. [4]These authors contributed equally: Bartul Mimica, Tuçe Tombaz. ✉e-mail: bmimica@princeton.edu; jonathan.whitlock@ntnu.no

It is becoming possible to address such questions in freely behaving animals owing to advances in quantitative pose estimation[21–24], as well as unsupervised machine learning approaches[25–28] for classifying unique actions based on underlying structure in tracking data. When paired with neural recordings, such techniques have afforded a range of recent discoveries, including how subcortical circuits generate sub-second patterns of behavior[29] or encode action space[30], the characterization of escape behaviors[31], or how different pharmacochemical substances leave tractable traces on the behavioral landscape[32]. Furthermore, machine learning has also enabled researchers to infer rodents' emotional states from facial videos[33] or control virtual rodent behavior[34], demonstrating the full promise in carefully quantifying animal actions.

Here, we sought to leverage such advances to determine the extent to which momentary behavior was represented across four major sensory and motor cortical regions, and whether the coding was uniform or varied depending on the region in which it occurred. Whereas naturalistic actions were encoded ubiquitously in sensory and motor areas alike, finer-grained features of pose and movement varied from one region to the next, following extended topographies that overlaid neighboring areas. In relation to sensory processing, pose and movement signals were integrated to similar degrees among sound- and light-sensitive cell populations in auditory and visual cortices, but closer analyses of putative synaptically connected pairs of neurons suggested different uses of pose and movement signals in each region. Thus, by considering neural tuning at different levels of behavioral complexity and among functionally connected neurons, we show that action representation may be a global feature of sensory and motor cortical systems, but that different regions preferentially encode different physical aspects of posture and motion, presumably to support locally unique computations during active sensing and movement.

## Results

### Naturalistic actions are represented across sensory and motor cortices in freely foraging rats

We combined 3D motion capture with chronic Neuropixels recordings to track the heads and backs of freely moving rats while recording large ensembles of single units from primary motor and somatosensory cortices (Fig. 1a, top; 4 animals, 1532 and 792 cells, respectively; "Methods") or visual and auditory cortices (Fig. 1a, bottom right; 3 animals, 1633 and 526 cells, respectively). Recording sites were localized to different cortical regions using a custom pipeline (Supplementary Fig. 1a–e and "Methods"), which allowed us to triangulate the anatomical position of individual channels along each probe (Supplementary Fig. 1f, g). Single units were classified as regular- or fast-spiking (RS or FS) by their spiking profiles (Supplementary Fig. 2 and "Methods") and were assigned to cortical sub-regions (e.g., S1 hindlimb region, S1 trunk, primary or secondary auditory, etc.)[35] in four overarching areas (motor, somatosensory, visual, auditory). Due to probe implantation angles, single units across visual and auditory cortices were sampled unequally across layers (V1 (L2/3 to L6), V2L (L6), A2D (L5 and 6) and A1 (L5 and 6; L4 in one animal); Supplementary Fig. 3). Somatosensory and motor cortical recordings included both superficial and deep layers (L2/3 and L5), but only superficial layers in somatosensory cortex (S1Tr; S1HL) were sampled sufficiently for analysis.

Animal behavior can be assessed at different levels of complexity, ranging from momentary poses to species-typical actions like rearing, and over longer time scales to capture hierarchical structure following shifts in internal states such as arousal or hunger[27,36,37]. We focused here on individual actions expressed over 20 minute sessions while rats were actively engaged in open field foraging. Actions were identified using existing approaches[27,38] to transform raw tracking data pooled across rats into sets of postural and movement features, and ultimately into a time-resolved ethogram (Fig. 1b and "Methods"). The

animals' combined ethogram consisted of 44 independent modular actions (Fig. 1c and Supplementary Figs. 4 and 5) comprised of unique composites of rudimentary pose and movement features (Supplementary Fig. 6a, b), that followed characteristic transition probabilities which were conserved across light and dark recording conditions (Supplementary Fig. 4b). The action "running, head up", for example, was most often followed by "running, head level, scanning", whereas "still, back hunched, head down" was followed by "still, curled right, head down", such as during grooming. As expected with freely behaving rodents[26,27], actions comprising the ethogram were not sampled equally (Fig. 1c, center) and varied from one session to the next (Supplementary Fig. 7a, top), though with greater commonality within individuals than between them (Supplementary Fig. 7b, top). The behavioral dynamics observed were also likely influenced by the foraging task, differing from purely exploratory behavioral patterns expressed in novel environments[39].

With the behavioral phenotype of the rats profiled, we characterized how neural responses in different cortical regions mapped onto the spectrum of identified actions, with analyses limited to recordings in darkness to minimize visual confounds. We found stable encoding of nearly every considered action by individual neurons in each cortical region (51% of neurons in visual cortex, 55% in auditory, 58% in motor and 56% in somatosensory; Fig. 1c), with most cells responding to multiple actions and fewer to single actions (Supplementary Fig. 7c, Supplementary Movies 1–6, and "Methods"). This was observed in all animals, irrespective of the specific placement of the recording probe, and the distributions of encoded actions were similar across regions (Fig. 1d and Supplementary Fig. 7b, bottom). Since sensory and motor coding properties can vary between FS and RS neurons[17,40,41], and between superficial and deep cortical layers[16,40,42], we further examined whether action encoding varied by cell type or lamination. We found that FS and RS neurons encoded similar ranges of actions within regions (Supplementary Fig. 8a), and that comparable fractions of cells encoded actions in superficial and deep layers in motor cortex, whereas action encoding in visual and auditory cortices was more common in layers 5 and 6 (Supplementary Fig. 8b). Crucially, decoding analyses at the ensemble level revealed that nearly any of the 44 independent actions with sufficient sampling could be predicted beyond chance in any of the four overarching areas (Fig. 1e and see Supplementary Fig. 9 for confusion matrices). Overall decoder accuracy was lower in visual cortex relative to other regions, but increased in all areas with larger numbers of simultaneously recorded cells (Supplementary Fig. 10). Moreover, decoding accuracies from sessions with more than 100 neurons were positively correlated (Spearman's $\rho = 0.39 \pm 0.11$ (mean ± std), $p = 0.007$; permutation test), underscoring cross-regional similarities in action modulation.

### Expression and statistical modeling of elementary pose and movement features

Although action representation was widespread in sensory and motor cortices alike, the uniformity of encoding gave few clues as to how such signals might influence processing in one region *versus* another. To gain traction on this question we more closely inspected neural tuning to finer grained behavioral features in each area, including posture and movement of the head, neck and trunk (along Euler axes of pitch, azimuth and roll), as well as whole-body movements such as self-motion and running speed. Head posture and movement were further divided into egocentric (relative to the trunk) or allocentric (relative to the world) reference frames, for a total of 23 features ("Methods"). Subsets of neurons in superficial, granular or deep layers of each region exhibited stable tuning curves for at least one of the considered features (Fig. 2 and Supplementary Fig. 11), similar to tuning reported previously in posterior parietal and motor cortices[43,44]. Coding properties of the cells were established by a statistical model

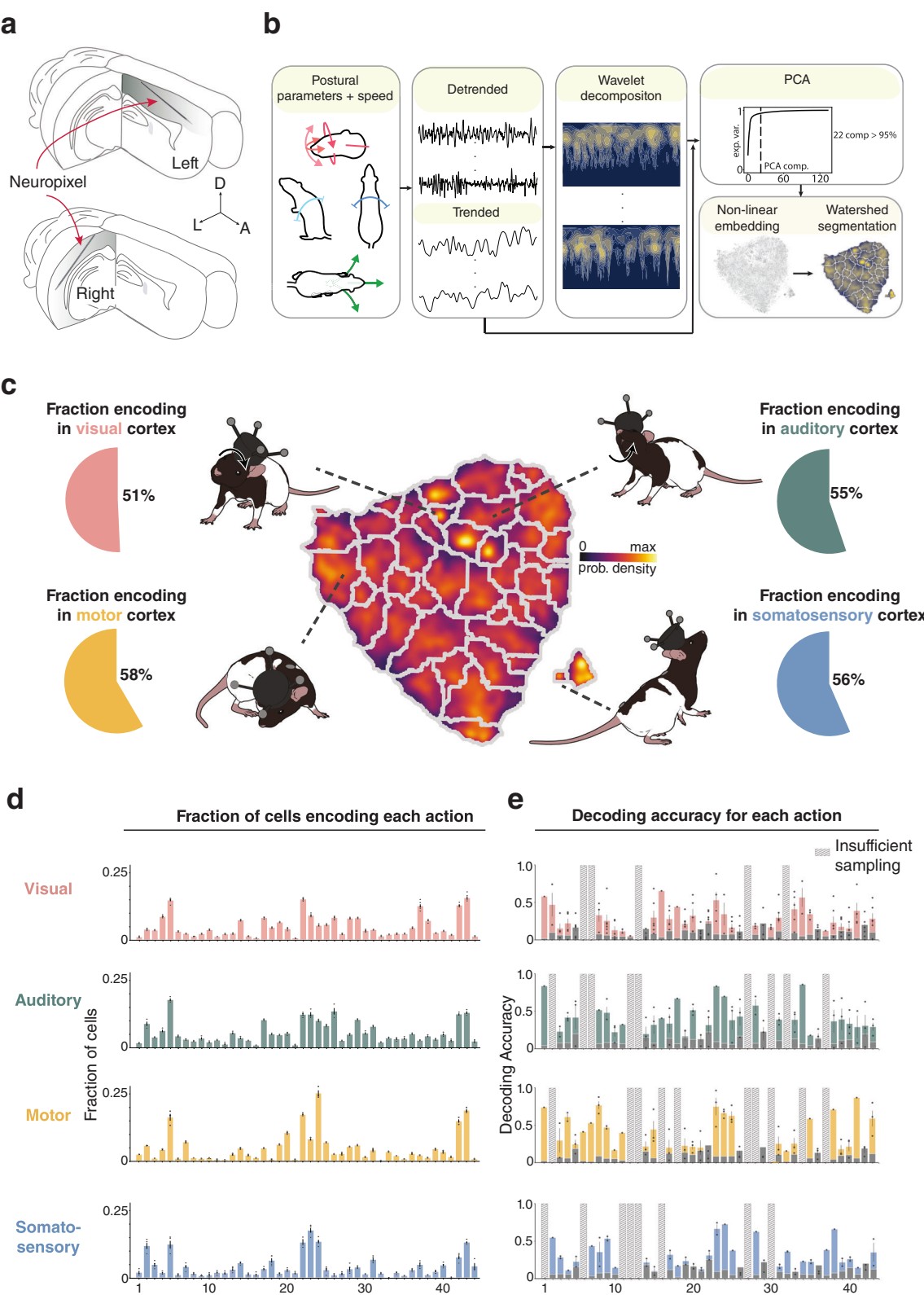

**d** Fraction of cells encoding each action

**e** Decoding accuracy for each action

selection framework (Supplementary Fig. 12a and "Methods"), again using recordings conducted in the dark. For each region we quantified (i) the fraction of neurons with selected covariates providing the single best out-of-sample fit relative to the intercept-only model (Fig. 3, inset pie charts), (ii) the proportion to which each covariate was selected among all selected covariates (Fig. 3, polar plot wedge widths), (iii) mean cross-validated relative log-likelihood ratio (rLLR, "Methods") for

each selected feature (Fig. 3, polar plot wedge heights), and (iv) the distribution of covariate counts, i.e., model sparsity (Fig. 3, gray histograms). The rLLR values in the polar charts illustrate how much the predictive power of the GLM suffered when a given feature was removed, i.e., the "importance" of that feature to the model. A graphical breakdown of the framework is provided on data pooled across regions (Supplementary Fig. 12b), as is the distribution of mean cross-

**Fig. 1 | Encoding and decoding of natural actions is robust in visual, auditory, motor and somatosensory cortices. a** (Top) Neuropixels probes implanted in the left hemisphere were tilted 60-70° in the sagittal plane, traveling anteriorly from S1 to M1. (Bottom) Separate animals were implanted with probes in the right hemisphere tilted 45--50° in the coronal plane, progressing laterally through primary and secondary visual and auditory cortices. **b** (Left) To extract discernible actions, time series were generated for each postural feature of the head (pitch, azimuth, roll) and back (pitch, azimuth), together with running speed and neck elevation. (Middle) Data were then detrended and decomposed spectrally using a Morlet wavelet transform. (Right) Features from all animals were sub-sampled and co-embedded in two dimensions using t-SNE, followed by watershed segmentation, producing a map in which distinct actions separated into discrete categories. **c** (Middle) A map of identified actions, color-coded for occupancy, shows cumulative sampling of the 44 actions. **d** The fraction of cells encoding each action, arranged by numerical labels (1 to 44), in each overarching region; all regions encoded both still and active behaviors. In visual cortex, the most common actions included #5, "still, head down left, curl left" and #22, "running, head level, scanning". Action #5 was also most common in auditory cortex, followed by #26, "walk, clockwise head roll". Motor cortical neurons most commonly encoded action #24, "still, CW head roll, slight up" and action #22, "running, head level, scanning", while somatosensory neurons most frequently encoded action #23, "running, head up". Dots indicate encoding rates for each animal in each session, bars and error bars denote the mean and ± standard deviation (stdv), respectively. **e** Decoding accuracy for each action in each cortical area, with analyses restricted to 60 cells to match sampling across regions. Dots denote decoder performance in each session for each rat, bars denote the mean, error bars denote ± stdv; gray bars show decoder performance consisting only of a prior distribution; actions with insufficient sampling shown as striped bars. Rat illustrations by Falconieri Visuals. ⓒ All Rights Reserved. Source data are provided as a Source data file.

validated pseudo-$R^2$ ("Methods") values across selected models for each area (Supplementary Fig. 13).

## Visual and auditory cortices encode the head in world-referenced coordinates

The majority of cells in visual cortex (62%) were selected as coding for one or more behavioral covariates (Fig. 3a, pie and polar charts), the strongest of which was for combinations of features capturing allocentric head movement along the horizontal plane (specifically, azimuthal head movement and planar body motion; Fig. 3a and Supplementary Table 1, bottom), as well as egocentric head posture. The same features were represented in superficial and deep layers, but by larger fractions of cells in layers 5 and 6 (Supplementary Fig. 14), consistent with laminar trends for encoded actions. Regular- and fast-spiking neurons also encoded the same covariates, though among a slightly larger proportion of FS than RS neurons (72% vs. 60%, respectively; Supplementary Fig. 15).

Behavioral tuning in auditory cortex was similarly widespread, with 63% of cells being explained by at least one covariate. Unlike visual regions, auditory neurons principally encoded features conveying gravity-relative head orientation (Fig. 3b, pie chart), with spiking activity best fit by models for allocentric head roll and pitch, followed by egocentric head posture (Fig. 3b, polar chart and Supplementary Table 1, bottom). Cells in different layers encoded the same features, but again in larger proportions in deep layers (Supplementary Fig. 14), and coding was expressed evenly across RS and FS neurons (Supplementary Fig. 15). In both visual and auditory regions, the proportion of selected covariates and their contribution to model performance depended little on whether the cells had high pseudo-R2 values (Supplementary Figs. 16 and 17). The overall number of tuned cells and their tuning properties were also largely similar across primary and secondary subareas, but neurons in primary areas consistently exhibited higher rLLRs (Supplementary Fig. 18a), and were better fit by sparser models, which was clearly contrasted by more complex behavioral modulation in secondary cortices (Supplementary Fig. 18b).

## Somatomotor regions encode the trunk and head in largely egocentric coordinates

Primary motor cortex had the largest fraction of cells encoding low-level features (79%), which principally included planar body motion, back movement and egocentric head posture (Fig. 3c, pie chart and Supplementary Table 1). This corresponded well with back movement and self-motion being among best represented features among classified cells (Fig. 3c, polar chart). Mean cross-validated rLLRs were moderate and strikingly similar across the most prominent features (Fig. 3c, polar plot), in agreement with complex models best accounting for motor cortex spiking activity (Fig. 3c, gray histogram). These models also generally exhibited higher explanatory power compared to those selected in other areas (Supplementary Fig. 13,

median cross-validated pseudo-$R^2$ for auditory (0.02), visual (0.01), motor (0.03) and somatosensory (0.02) areas), as could be expected from a cortical region involved primarily in generating movement.

Our recordings also uncovered dense representation of head kinematics, particularly in the deep layers of motor cortex (Supplementary Fig. 14), which presented the opportunity to determine if neural encoding of spatial kinematics changed with an added load on the head[45,46]. We therefore performed additional recordings with a 15 g weight added to the animals' implants (Supplementary Fig. 19a) to determine if this changed the shape or stability of tuning curves for head posture in motor cortex or, for comparison, visual cortex. The added weight caused the animals' heads to roll slightly towards one side (Supplementary Fig. 19a), but had little effect on behavior or tuning in either motor or visual areas (Supplementary Fig. 19b,c), except for higher firing rates and stability of azimuthal egocentric head posture tuning across weight-free sessions (Supplementary Fig. 19d and Supplementary Table 2). Likewise, similar proportions of behavioral covariates were selected by GLMs trained on recordings from the first weight-free session and cross-validated separately on recordings with or without the head weight (Supplementary Fig. 20).

Somatosensory cortex primarily responded to planar body motion, back movement and posture (Fig. 3d). It had the largest overall proportion of unclassified units (51%; Fig. 3d, pie chart), though a notably larger fraction of FS neurons were tuned than RS neurons (69% vs. 46%, respectively), particularly for back movement and planar body motion (Supplementary Fig. 15). Compared to other areas, somatosensory neurons were better fit by sparser models containing either one or two covariates (Fig. 3d, gray histograms). Lastly, having also captured head kinematics with head-mounted accelerometers, we compared modeling results based on accelerometer-generated covariates to those based on optical tracking. Although both approaches revealed encoding of allocentric head features in auditory and visual cortices, only optical tracking, which included tracking of the back, was suited to distinguish the dominant egocentric head and back-related receptivity in motor and somatosensory areas (Fig. 3a–d and Supplementary Fig. 21).

## Topographical mapping of behavioral features across cortical regions

Up to this point in the study we found that discrete, naturalistic behaviors were represented throughout the cortical areas recorded, and were composed of simpler posture and movement primitives whose expression appeared to vary regionally. We therefore next sought to establish (i) how the neural coding of pose and movement was organized within and between regions, (ii) the extent to which pose and movement signals were integrated with sensory input in visual and auditory cortices, and (iii) how pose and movement signaling might be utilized in different areas.

For the first question, we assessed whether region-specific differences revealed by the GLM analysis emerged abruptly between

## a

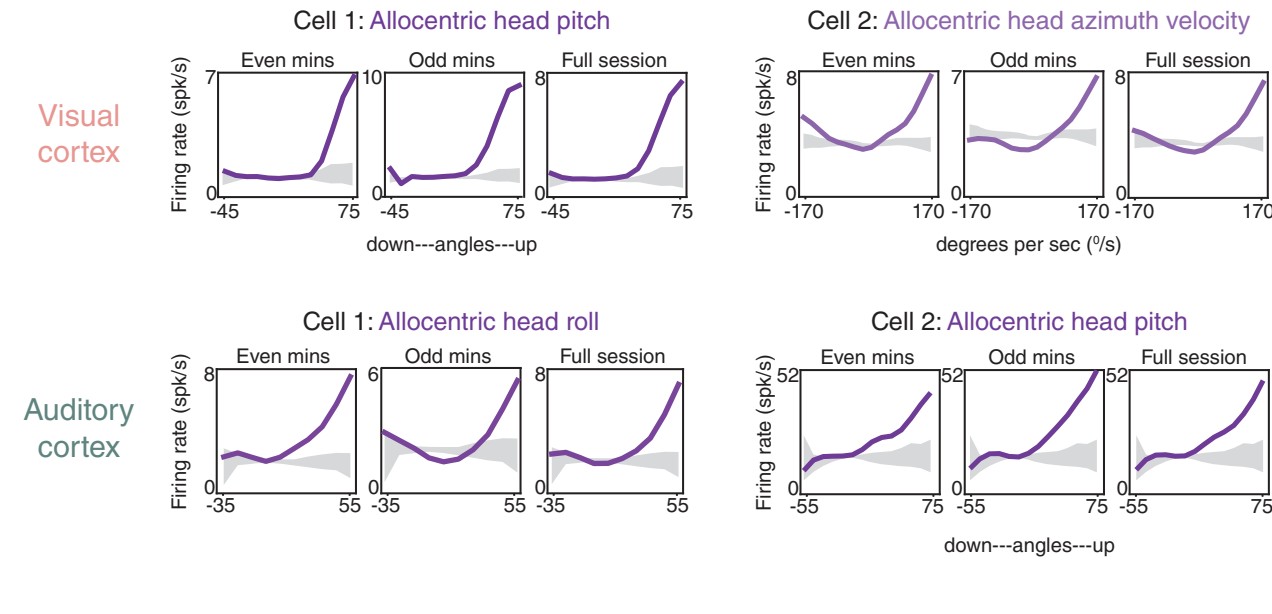

## b

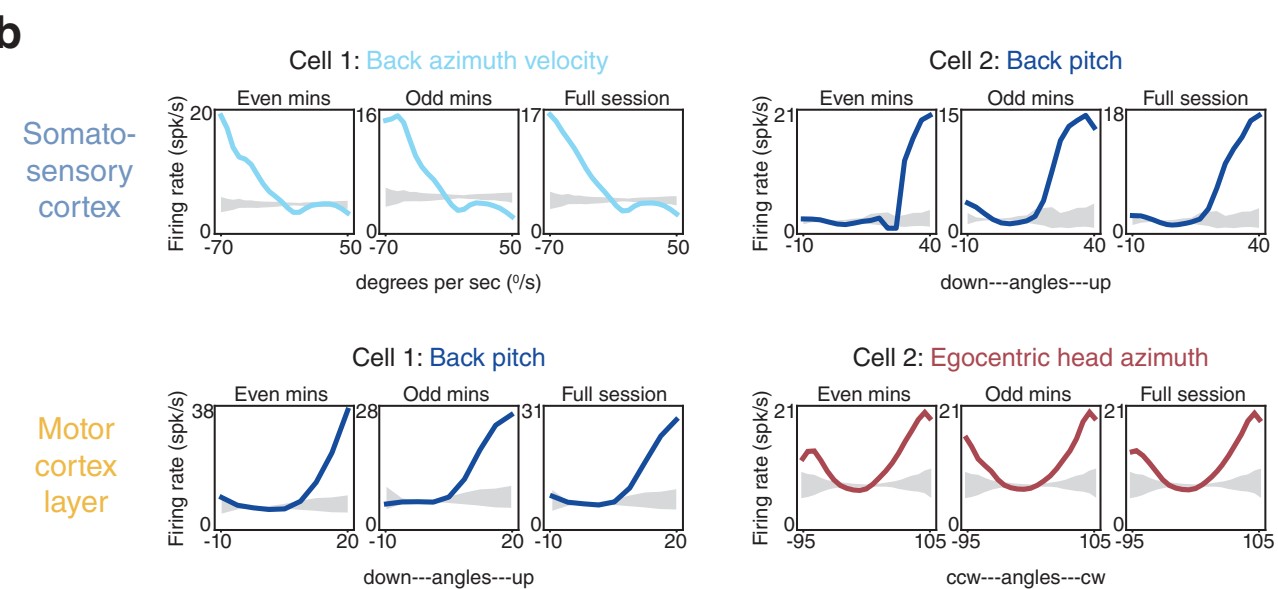

**Fig. 2 | Tuning curves for posture and movement in superficial or granular layers in each overarching cortical area. a** (Top left) Tuning curves from a layer 2/3 visual cortical neuron (Cell 1) preferring upward pitch of the head in allocentric coordinates (all examples recorded in darkness). Data from even and odd minutes of a 20 min recording session are shown adjacently (left and middle), and full-session data are shown to the right; the 99% CI of shuffled data are shown in gray. A total of 30% of visual cortical neurons had even-odd minute tuning curve stability higher than the 95th quantile of shuffled data ("Methods"). In relation to the subsequent GLM analyses (in Fig. 3 and Supplementary Figs. 12 and 13), the pseudo-$R^2$ value for allocentric head pitch for Cell 1 was 0.003. (Top right) A layer 2/3 visual cortical neuron tuned to right head azimuth velocity in allocentric coordinates;

pseudo-$R^2$ value of 0.01. (Lower left) A layer 4 auditory cortical neuron preferring rightward head roll (pseudo-$R^2$, 0.004), and (lower right) a L4 auditory neuron tuned to upward pitch of the head in allocentric coordinates (pseudo-$R^2$ of 0.001). 37% of auditory cortical neurons had tuning curves exceeding the shuffled distribution (as described above). **b** Somatosensory cortical neurons in layer 2/3 tuned to leftward flexion velocity of the back (top left; pseudo-$R^2$ of 0.03), and upward pitch of the back (top right; pseudo-$R^2$, 0.02). 43% of S1 neurons were stable beyond the shuffled data. (Lower left) Tuning curves from layer 2/3 motor cortical neurons preferring upward pitch of the back (pseudo-$R^2$ of 0.06), and (lower right) left and right head roll relative to the trunk (pseudo-$R^2$ of 0.01). Tuning curves from 66% of M1 neurons were stable beyond the 95th quantile of shuffled data.

areas, or followed continuous topographies that spanned cortical boundaries. The first covariates we considered were allocentric head posture and movement in visual and auditory areas, since they were represented most prominently and had previously only been studied within V1[16,17,40]. This revealed a graded increase in allocentric head posture coding that progressed laterally from V1 to V2L and peaked in A2D, ($\chi^2(7)$ = 29.5, $p$ = 4.8e−5), as well as a peak in allocentric head

movement tuning nearby in V2L ($\chi^2(7)$ = 13.09, $p$ = 0.04; Fig. 4a and "Methods"; data from individual animals shown in Fig. 4b). The representation of planar body motion features (e.g. self-motion and turning direction) also increased laterally across V1 and reached a maximum in deeper cortical layers at the border of V1 and V2L ($\chi^2(7)$ = 18.2, $p$ = 0.006; Fig. 4a, b and Supplementary Fig. 14). Together, these covariates largely accounted for the apex of coding cells around the

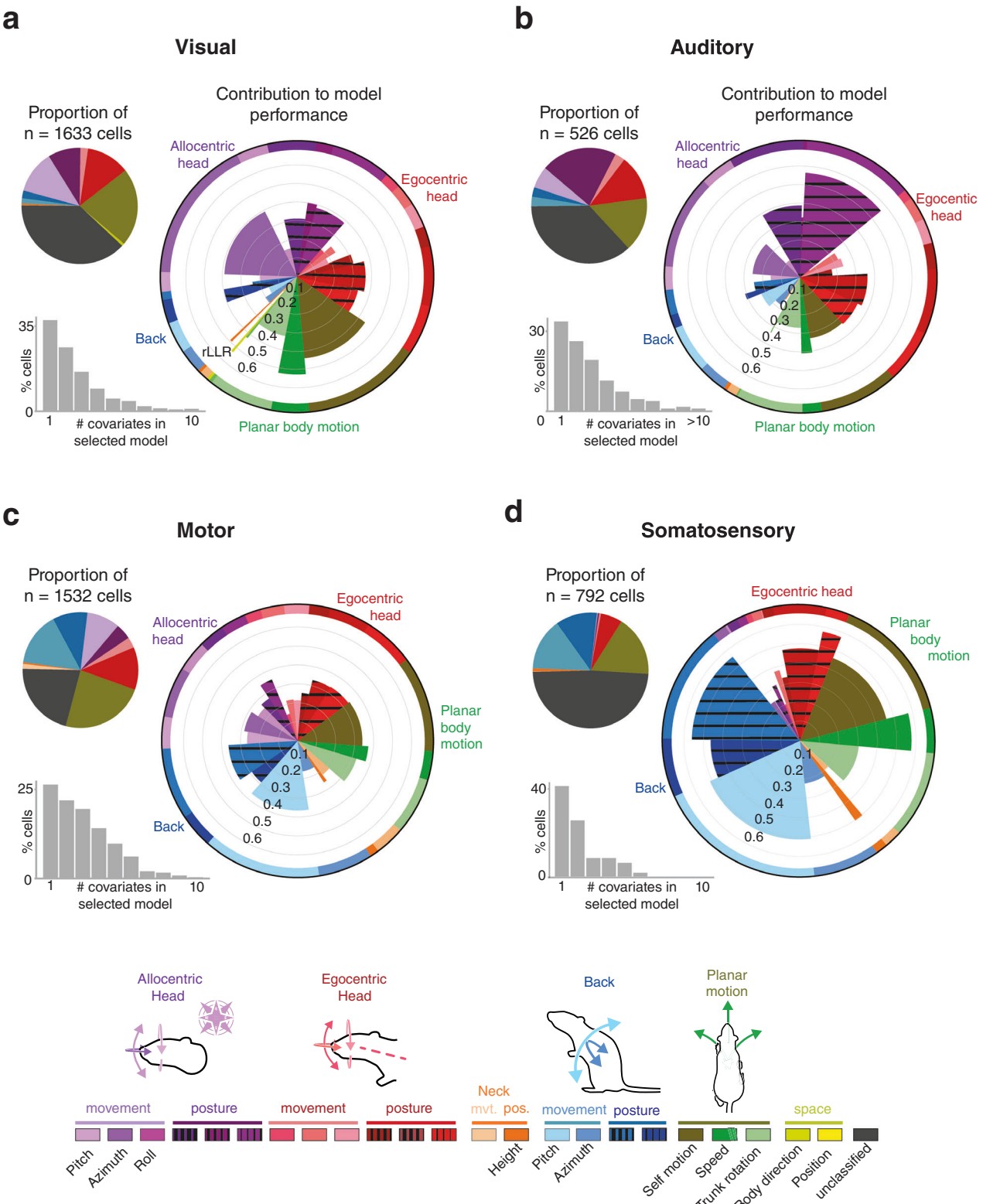

visual-auditory cortical border (Fig. 4a, pie charts). Across S1 and M1, on the other hand, we found continual gradients for egocentric head features and the back (Fig. 4c). Specifically, neurons encoding egocentric head posture and movement were more frequent in anterior than posterior M1 or S1 (posture: $\chi^2(7) = 37.7$, $p = 1.3e{-}6$; movement: $\chi^2(7) = 106.9$, $p = 8.8e{-}21$), whereas back representations dominated in posterior motor areas and S1HL (posture: $\chi^2(7){=}18.8$, $p = 0.004$; movement: $\chi^2(7) = 41.02$, $p = 2.9e{-}7$) (Fig. 4c, d). The total fraction of

classified cells was higher in anterior locations, (Fig. 4c, pie charts) peaking in M1.

### Sensory and behavioral tuning overlap to similar extents in visual and auditory populations

We next sought to characterize the degree to which sensory and behavioral signals overlapped in sensory cortices, focusing on visual and auditory modalities since these could be manipulated without

**Fig. 3 | Distinct cortices show differential tuning to posture, movement and self-motion. a** (Top left) Pie charts showing the fraction of single units in visual cortices (V1, V2L and V2M) that incorporated specific behavioral features as the first covariate (largest mean cross-validated log-likelihood among single covariates models) in model selection (see color-coded legend at bottom for feature identification). (Lower left) Bar graph indicating the percentages of single units statistically linked to one or any larger number of behavioral covariates. (Right) Polar charts denoting the relative importance of individual covariates included in the full model, after model selection, with "unclassified" cells excluded. Wedge length denotes the mean cross-validated rLLR ("Methods") of each covariate across the set of models it was included in. The width of each wedge reflects the fraction of times that feature was selected among the other covariates. Wedge length and width are independent (i.e. the area does not reflect effect size). **b** Same as (**a**) but for auditory cortices (A1 and A2D). **c** Same as (**a**) but for the primary motor cortex (M1). **d** Same as (**a**) but for the primary somatosensory cortex (S1HL and S1Tr). (Bottom) Two color gradients were used to convey GLM results: one in pie charts (denoted by elongated lines) with related features grouped together, and one in polar plots (denoted by clear and striped rectangles) with each representing an individual feature. Rat illustrations by Falconieri Visuals. © All Rights Reserved. Source data are provided as a Source data file.

disrupting the animals' movement. Visual and auditory receptivity were first assessed across subsequent recording sessions with the room lights on or off and, separately, with intermittent presentation of 5s white noise sequences (Fig. 5a, b and Supplementary Fig. 22a, top). Sound and luminance modulation indices (Supplementary Fig. 22b, c and "Methods") were used to identify sound-suppressed and sound-activated neurons (35.3%, Fig. 5a, b), which were concentrated near the tip of the probe (i.e., in auditory areas), and luminance-suppressed and -activated neurons further up the shank in visual areas (26.9%, Fig. 5c and Supplementary Fig. 22c). Decoding analyses confirmed that auditory, but not visual, units predicted sound stimulus presentation (Supplementary Fig. 22b, bottom). In contrast, population vectors of visual neurons occupied distinct locations in a non-linear UMAP embedding (Supplementary Fig. 22a, bottom) and could be used to reliably decode the luminance condition of different sessions, whereas A1 neurons could not (Supplementary Fig. 22c, bottom). Seventy-five percent of auditory cortical neurons were modulated by behavior or white noise, of which >3-fold more were exclusively modulated by behavior (40%) than sound (12%), and 23% were tuned to both (Fig. 5d, left). Nearly identical proportions were observed in visual cortices (though using luminance, a coarser measure of sensory receptivity), with 42% of neurons tuned exclusively to behavior, 12% exclusively luminance-sensitive, and 20% tuned to both (Fig. 5d, right). The proportion of behaviorally- and sensory-modulated RS and FS neurons was nearly uniform across visual and auditory areas. We next compared the stability of behavioral representation across light and dark conditions, and found that GLM-selected features in visual regions were more stable between light sessions than between light and dark sessions (Supplementary Fig. 23). Planar body motion, allocentric head movement, and allo- and egocentric head posture were the least stable covariates, but were still maintained in darkness, presumably by vestibular, proprioceptive or efference copy signals[16,17,40,47]. Tuning in auditory cortex, on the other hand, was stable across light and dark conditions (Supplementary Fig. 23; $p > 0.05$, two sample $Z$-test for proportions).

### Functional coupling suggests different uses of behavioral signals in different regions

The overlap of sensory and behavioral tuning, while substantial, was not informative as to how behavior-related signals were utilized within the networks, so we sought to characterize which behavioral signals were expressed between putative synaptically-connected cells. This allowed us to discern whether feedforward excitatory and inhibitory signaling differed in relation to specific types of perceptual processing or motor behavior, and to test whether the likelihood of synaptic connections was higher between similarly tuned cells, as reported among orientation-selective neurons in visual cortex[48–50]. We identified a limited, but informative number of putative excitatory connections (driven by RS units) and inhibitory connections (driven by FS units) of varying strength within auditory ($n = 107$ total connections) and visual cortices ($n = 247$) (Fig. 6a, b and "Methods") as well as somatosensory ($n = 35$) and motor cortices ($n = 181$) (Supplementary Fig. 24a, b). Cells were classified as encoding posture ("Po") or movement ("Mo") based on best-fit single covariate models from a dark recording session

("Methods"), and visual and auditory neurons were identified as luminance- ("LM") or sound-modulated ("SM"). Since the firing rates of neurons tuned to multiple features varied in higher dimensions, low dimensional embeddings were generated with UMAP (Supplementary Fig. 24c) to visualize the functional similarity between neuron pairs.

Different functional connection subtypes were uncovered in each area (Fig. 6c and Supplementary Fig. 24c–f), with the strongest connections in motor and somatosensory cortices being feed-forward excitation between movement-responsive neurons (movement → movement), and excitatory (posture → movement) connections in somatosensory cortex (Supplementary Fig. 24f). Relative to motor and somatosensory areas, visual pairs tended to be more homogeneous, and both visual and auditory synapses were weaker (Supplementary Fig. 24e, top). Furthermore, in visual areas, aside from excitatory (movement → movement, posture → posture) and inhibitory (movement ⇢ movement) communication between functionally homogeneous units, we found excitatory (posture → luminance modulated, posture → movement) and inhibitory (movement ⇢ posture) drive in functionally heterogeneous units (Fig. 6c, top). A large majority of connections between heterogeneous units appeared in V2L (106/122 units, 86.8%), which was significantly larger than in V1 ($\chi^2(1) = 66.39$, $p = 3.7e−16$). Auditory areas, in contrast, exhibited completely different patterns of connectivity (Fig. 6c, bottom). On one hand, we found a significant amount of movement-inhibited posture-encoding units (i.e., movement ⇢ posture), and this connection subtype occurred most prominently in A2D (14/18 units, 77.8%), as opposed to A1 ($p = 0.015$; one-sided binomial test). On the other, pairs of movement-inhibited sound-modulated units (i.e., movement ⇢ sound modulated), featured more frequently in A1 (8/13 units, 61.5%), though the low total number of units precluded the difference with A2D to reach statistical significance ($p = 0.29$; one-sided binomial test). We also noted that results obtained on synaptically connected pairs stood in stark contrast to those obtained on pairs receiving common input, which consisted almost exclusively of functionally homogeneous units (Supplementary Fig. 24g). Lastly, we assessed whether functionally similar neurons were more likely to exhibit stronger synaptic connections, but found no correlation between synaptic strength and functional distance in any of the four cortical areas (Supplementary Fig. 24e, bottom). Similarly tuned neurons did not form functional subclasses, and strong connections were found between all functional cell types, indicating that communication between both functionally similar and dissimilar cells was important.

## Discussion

In this study, we strove to address one of the core challenges in understanding ethologically relevant neural computations, namely, how cortical networks differentially represent freely-composed behavior. Visual and auditory areas were of particular interest because of their pervasive modulation by behavioral state[13,51–53], movement during sensorimotor tasks[10,11,14,16,40,54–63], or while animals cognitively idle in the dark[12]. To gain a clearer grasp on which actions were encoded by cortical activity, we focused on what rats do when allowed to explore a familiar space without constraints. We converted tracked head and back points into series of postural and movement features, quantified

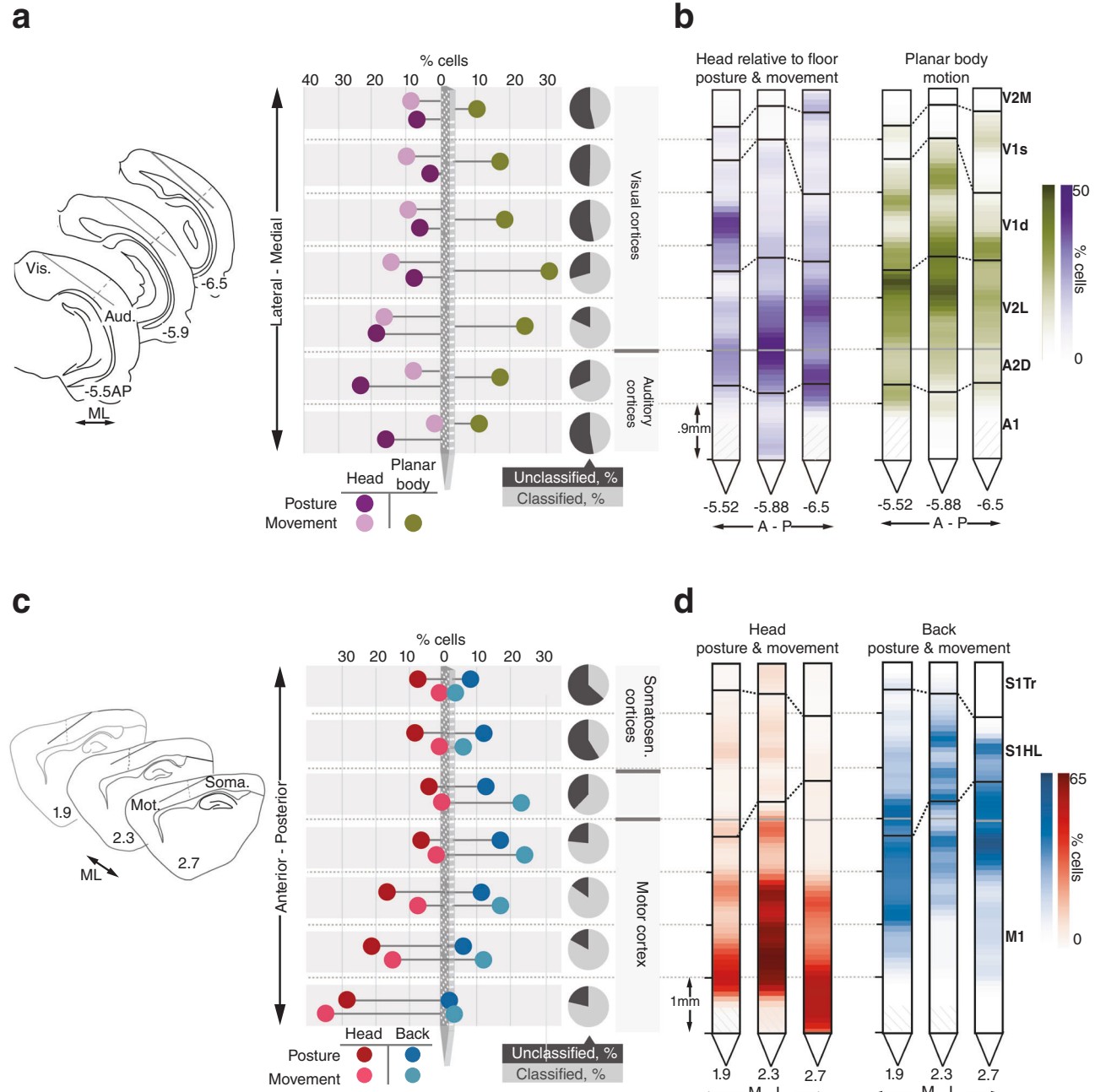

**Fig. 4 | Anatomically organized behavioral tuning gradients span sensory and motor regions. a** (Left) Probe positions relative to corresponding coronal atlas sections in all right hemisphere implanted animals; the dotted line marks the border between auditory and visual cortices. (Middle) Percentage of all recorded cells responsive to allocentric head posture, allocentric head movement and planar body motion along the mediolateral axis. (Right) The fraction of cells showing any type of behavioral tuning at a given location is indicated by the gray and black pie charts. **b** Probes from each animal showing the percentage of cells encoding behavioral features along their length. **c** (Left) Same as in (**a**) only for left hemisphere implanted animals, with the dotted line marking the border between primary somatosensory and motor cortices. (Middle) percentage of all recorded cells responsive to egocentric head posture and movement, and back posture and movement along the rostrocaudal axis. (Right) same as in (**a**), but for cells classified along the length of S1-M1. **d** Same as (**b**), but using the probe view along S1 and M1 for each animal. Source data are provided as a Source Data file.

the joint rat ethogram and characterized how recognizable modular actions were represented across cortical regions. The four overarching areas differed considerably in their connections, cytoarchitecture, and layers sampled, but our finding that every region carried sufficient information to decode nearly any action suggests that ongoing behavior continually modulates computations throughout dorsal cortical systems. This could provide a foundation both for contextualizing environmental inputs[64] and informing sensory predictions generated by internal models[18,65–69], since an animal's behavior at any moment

profoundly affects the spatiotemporal statistics of ongoing and impending sensory signals.

However, ethograms are descriptive of only one level of behavioral organization. When we "zoomed in" and modeled spiking responses on continuous elementary features, like rotational movements or angular positioning of the head, we encountered a wealth of encoding variety across cortical structures. The response variance of motor and somatosensory units was best captured by kinematic parameters and included both egocentric head and back movements

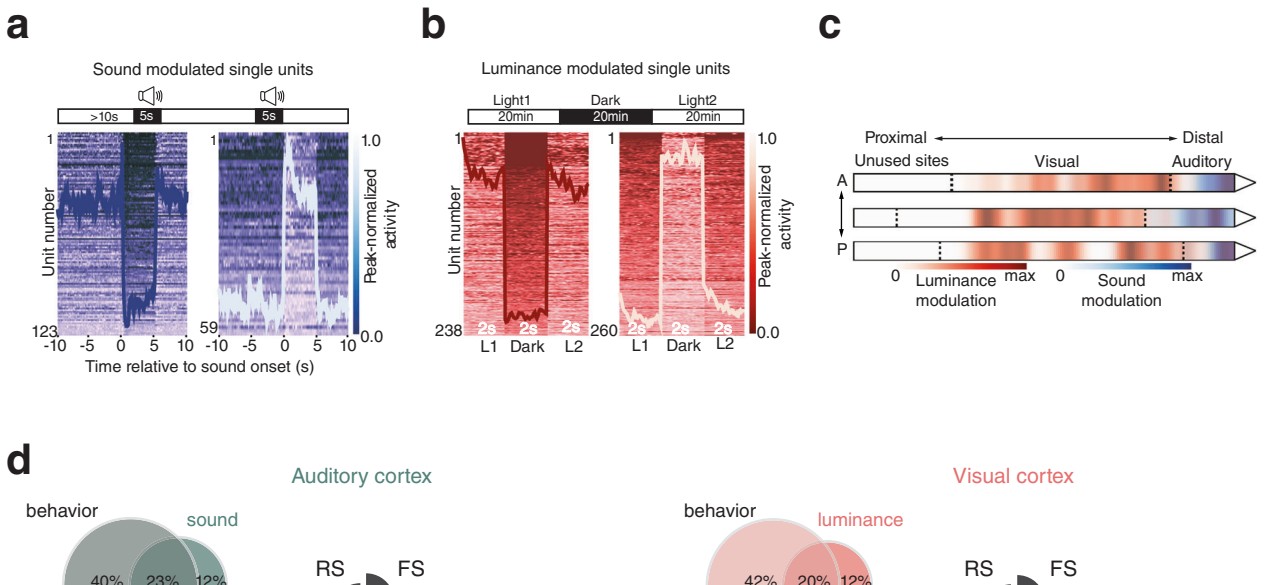

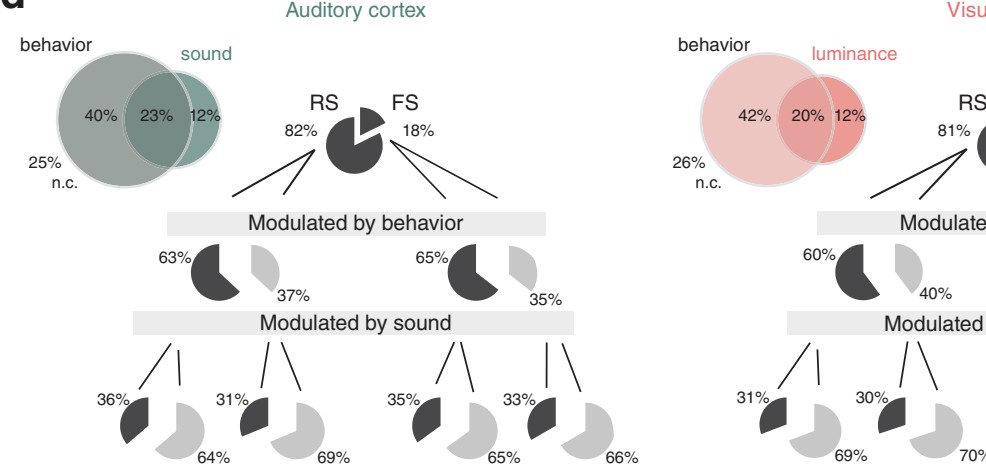

**Fig. 5 | Auditory and visual cortices show similar prevalence and overlap of sensory and behavioral signals. a** (Top) White noise stimulation paradigm schematic. (Bottom)Trial averaged activity for single cells (arranged by unit number, left axis) and ensemble averaged activity (overlaid trace) of all significantly suppressed (left) and sound-excited (right) single units in auditory cortices during sequences of sound stimulation. Colorimetric axis denoting activity relative to max are shown to the right. **b** (Top) Recording paradigm for light/dark sessions. 2 s segments of trial averaged and ensemble averaged activity of all significantly dark-suppressed (left) and dark-excited (right) single units in visual cortices. Axis labels are the same as in (**a**). **c** Distribution of recording sites of single units modulated significantly by sound (blue) or luminance (red) across sensory cortices (opacity represents concentrations of units). Sound- and light-modulation within auditory and visual

cortices, respectively, were highest in layer 4 (L4). In A1, 48% of L4 neurons had significant sound modulation indices (SMIs), followed by layers 5 and 6, with 26% and 25%, respectively. In V1, 30% of layer 4 neurons had significant luminance modulation indices (LMIs); layer 5 had 27%, layer 6 had 24%, and layer 2/3 had 23%. **d** (Left) Venn diagrams and pie charts summarizing the overlap between and breakdown of spiking profile, sound modulation and behavioral tuning (as determined by the GLM analysis; n.c. non coding) in auditory cortices (A1 and A2D); the fraction of fast spiking (FS) and regular spiking (RS) cells in each category are indicated on either side of each pie chart. (Right) Same as (left) but for luminance modulation and visual cortices (V1, V2L and V2M). Source data are provided as a Source data file.

and poses, respectively. Motor neurons most frequently encoded head rotations around two or three axes, and spiking activity was accounted for better than any other area, as could be expected of a population generating signals that control movement. The robust encoding of posture and kinematics of the head allowed us to test whether the addition of weight affected how neurons encoded these features, a question approached previously in relation to hand and arm movements in primates[45,46,70,71]. The added weight had only minor effects on the tuning properties of M1 neurons (Supplementary Figs. 19 and 20), though signals related to sensory feedback, planning or dynamic pattern generation could also have contributed since the animals were moving freely[72,73].

Somatosensory neurons, on the other hand, were better characterized by sparser models related to the trunk, and tuning was more common among FS than RS cells. Fast spiking neurons in particular encoded dynamic features such as trunk movement and self-motion, which fits broadly with known circuitry for movement-driven disinhibition of parvalbumin interneurons in S1, described in barrel cortex during active whisking[41]. Overall, however, model performance in S1

was the lowest of all regions, which could be traced to different factors. It was not likely due to the recordings being confined to layer 2/3, since only 22% of layer 2/3 neurons in adjacent motor regions were unclassified. In this case, unclassified neurons could have been driven by features we did not track, such as the whiskers, face, limbs or tail. The hindlimbs may have been more influential due to the probes being placed in the hindlimb and trunk regions of S1, medial to barrel cortex, and the fact that trunk kinematics in quadrupeds are steadily affected by gross dynamics of the hindlimbs and hips[74]. This interpretation also fits the somatotopic organization of the trunk and limbs in rats (Hall and Lindholm[75]; Neafsey et al.[76]), as well as the anatomical gradients we found for trunk and head features, and the higher overall fraction of classified cells anteriorly in motor cortex (Fig. 4c, right).

In visual and auditory cortices, by contrast, the common denominator was encoding of the head in world-centered coordinates, consistent with a role in processing sensory signals from the environment via sense organs embedded in the head. The activity of visual ensembles was best described by allocentric horizontal motion of the head and movement of the body over the surface of the arena, which

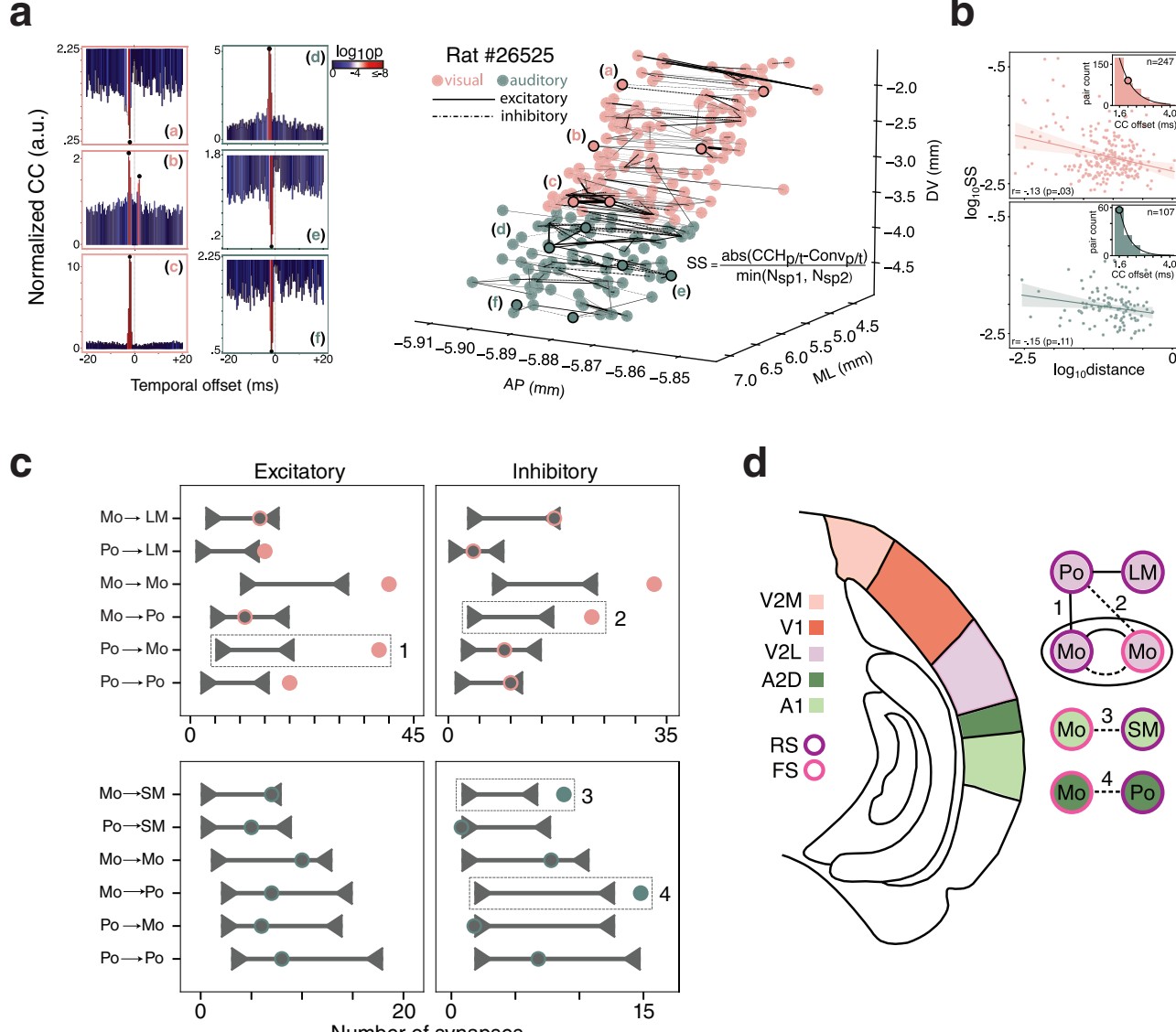

**Fig. 6 | Synaptic connectivity patterns reveal behavioral information is employed differently across auditory and visual subregions. a** (Left) Three example temporal spiking cross-correlograms from visual (a–c) and auditory cortices (d–f). Significance threshold was set at the 99.9999% or 0.0001% of the cumulative Poisson distribution for excitatory or inhibitory connections, respectively; P-value calculations described in detail in "Methods" sub-section "Functional connectivity." (Right) All putative synaptic connections including outlined examples (a–f) from one example rat's (#26525) visual (pink) and auditory (green) areas localized in anatomical space, with the width of each connection weighted by synaptic strength (SS). **b** (Top) The log-log relationship of anatomical distance and synaptic strength (SS) revealed weak effects of anatomical distance on SS (two-sided $r = -0.13$, $p = 0.03$); (inset) distribution and median (colored circle) of cross-correlogram peaks/troughs. (Bottom) Same as above, only for auditory cortices; SS did not vary as a function of anatomical distance (two-sided $r = -0.15$, $p = 0.11$). Lines

in the scatter plots denote the least squares regression fit, shaded areas denote the 95% confidence intervals; the circle in inset histograms denotes the median of the distribution. **c** (Left) Shuffled connection distributions (horizontal line with carets) and experimentally observed excitatory and inhibitory connections (circles) for various functional subtypes in visual (above) and auditory (below) cortices. **d** (Top) The V2L network utilizes excitation and inhibition between postural and movement ensembles (connections 1 and 2, respectively; dashed boxes in (**c**)) which could serve to coordinate impending movements with visual flow. (Middle) A1 FS ensembles inherit movement information, which could enable gain-modulation of local sound modulated RS units (connection 3) in response to self-generated sounds. (Bottom) A2D FS movement modulated ensembles inhibit gravity-relative, posture-responsive units (connection 4), which could facilitate sound localization. Source data are provided as a Source data file.

confirms and extends earlier observations that features like angular head velocity[16,17,40], head direction[77], and running[54] each modulate spiking activity in V1. The combination of these signals could support computations for stabilizing and predicting motion of the visual field, as well as optimizing visual processing during active movement[55,78]. The preferential expression of dynamic lower-level behavioral features in layers 5 and 6 is also supported by earlier work demonstrating head-motion signaling in deeper visual cortical layers during head rotations in darkness[16,40]. Those and our findings point to the deeper layers as a

potential site of integration for movement and vestibular cues conveyed by midline motor[79] and retrosplenial[16] cortices, respectively.

Unlike visual cortical neurons, neurons in auditory cortex were most robustly triggered by gravity-relative head orientations. Such differences were made more prominent when we quantified connectivity differences between subregions within each area (Fig. 6d). The lateral part of the secondary visual cortex (V2L), specialized for processing visual motion[80–82], exhibited extensive cross-talk between movement-modulated cells, where putative feedforward excitation

happened between similarly tuned cells, and apparent feedforward inhibition occurred between units encoding movements in opposite directions. More importantly, such fast spiking movement-responsive cells also inhibited posture-encoding units (Mo ⇢ Po; e.g., a left turn cell inhibiting a cell encoding rightward posture), and posture-modulated neurons were shown to excite luminance-modulated and movement-responsive cells (Po → Lm; e.g., a rightward pose cell exciting a leftward-rotation neuron). This might assist visual networks in distinguishing whether optic flow is likely generated by self-movement[83]: if postural cells convey static head position, their direct inhibition would indicate that visual scene changes are likely self-generated. Furthermore, when the animal's head is fully extended in one direction, it can only really move in the opposite way; in this case, postural cells exciting opposite movement cells (Po → Mo) could aid in predicting the direction the visual scene will drift next[84,85].

The dissimilar connectivity patterns across primary and secondary auditory cortices may also indicate regional specializations in how behavioral information is employed. Fast spiking cells encoding horizontal movements (i.e., turning clockwise and counterclockwise) and inhibiting sound modulated units (Mo ⇢ Sm) were mainly found in A1, which is strongly reminiscent of gain mechanisms for suppressing self-generated sounds identified in head-fixed animals[60]. Many such cells also showed evidence of receiving common input, consistent with efference copy circuits described in A1[79,86,87], but here demonstrated in freely moving and sensing subjects. The finding that homogeneously and heterogeneously tuned pairs showed different connectivity motifs, and that the analysis was agnostic to the functional properties of the cells, suggests that functional groupings were not an artifact of the analysis. Movement-encoding fast spiking cells were also shown to inhibit allocentric posture-encoding cells (Mo ⇢ Po), which mostly occurred in A2D (e.g., a cell encoding leftward rolling of the head inhibiting a cell for right head roll posture). Circuits with this pattern of connectivity would report minute changes in posture whenever the animal moved and, conversely, maintain a strong readout of the current postural state when the rat was still. The roll and pitch of the head could be heavily represented because these features influence the detection of interaural loudness differences (ILDs) in a 3D environment. That is, if a rat's head is perfectly level, any ILD corresponds to a change in horizontal location, and the animal is 'blind' to changes in vertical localization. Once the head rolls, this changes, and strong roll- and pitch-modulation would facilitate detection of ILDs in vertical space, allowing for 3D sampling (Lauer et al.[88]). Further work is needed to test if this is the case in rodents, but the interpretation is supported by observations in different species of owl that have vertically offset ear openings in the head which facilitate sound localization using vertical ILDs[89,90]. Additional work is also required to define how postural signals contribute to sound localization at the level of cortex, though auditory cortical neurons in many mammalian species encode sound source locations uniformly and over broad ranges of spatial locations[91,92], so added modulation by posture and motion of the head would help discriminate the location and direction of a sound relative to the individual[93].

Lastly, the most abundant connection types in motor and somatosensory cortices supported excitatory cross-talk between movement-modulated neurons, though defining the functions of these connections here is difficult given the lack of a more elaborate task structure for the former, and the lack of additional sensory manipulations for the latter. Previous work in motor areas has shown that weak feedforward connections, in conjunction with synaptic noise, help sustain behavioral variability, as demonstrated in simpler organisms (e.g., ref. 94), but the strengthening of such connections with sustained repetition leads to motor outputs being less variable (e.g., refs. 95,96). In that sense, the occurrence of feedforward excitation of similarly tuned cells may be a feature of a network mechanism which ensures kinematic stability during different iterations of the same

actions. Furthermore, excitatory posture → movement connections between somatosensory neurons, particularly those representing the trunk or limbs, could assist in predicting expected movements following a given starting posture or gait, and signal postural instability when those expectations are violated[97,98].

In summary, we found that momentary actions are encoded broadly across dorsal sensorimotor cortices, and that coding of underlying pose and movement primitives appears regionally optimized for different types of sensory processing. The widespread expression of such features speaks to the importance and computational demand of keeping sensory systems oriented during unrestrained movement, and may reflect a general property of sensory coding in cortex. Although we did not test how behavioral and sensory information are combined here, our observations could guide future work investigating how pose and movement signals inform perceptual operations like visual self-motion subtraction or sound localization. Pinpointing exactly how this integration happens at the circuit level may prove challenging in freely-behaving subjects, requiring sufficiently resolved techniques, such as miniature 2-photon imaging[99,100] and holographic stimulation[101], to identify and manipulate behaviorally-classified neurons in vivo. Other outstanding questions are how internal states, such as hunger or thirst[102], or how the hierarchical organization of behavior[27,37] influence the way in which relevant actions are encoded over longer time scales. For now, defining how sensory cortices encode behavioral states and precise kinematics is a critical first step toward understanding how these systems respond in more naturalistically engaging environments, and establishing how behavioral modulation is implemented to solve locally relevant problems.

## Methods
### Subjects and electrode implantation
Experiments were performed in accordance with the Norwegian Animal Welfare Act and the European Convention for the Protection of Vertebrate Animals used for Experimental and Other Scientific Purposes. The study contained no randomization to experimental treatments and no blinding. Sample size (number of animals) was set a priori to at least three per recorded brain area, given the expected cell yield necessary to perform unbiased statistical analyses. A total of 7 male Long-Evans rats (age: 3–4 months, weight: 350–450 g) were used in this study. The rats were housed with their male littermates prior to surgery, and single housed in cages (45 × 44 × 30 cm) after surgery to avoid potential damage to the implants. All animals were kept on a reversed 12 h light–dark cycle and recordings were performed during their light cycle.

All 7 rats were implanted with silicon probes (Neuropixels version 1.0, IMEC, Belgium) developed for high-density extracellular recordings[103], targeting primary sensory and motor cortices. At surgery, animals were anesthetized in a ventilated Plexiglas box with 5% isoflurane vapor, and maintained on 1.0–2.5% isofluorane for the duration of surgery. Body temperature was maintained at 37 °C with a heating pad. Once unconscious, animals received s.c. injections of analgesic (Metacam 2.5 mg/kg weight, Temgesic (buprenorphine) 0.05 mg/kg). Local anesthetic (Marcain 0.5%) was injected under the scalp before making the incision. The skull was then exposed, rinsed and sterilized using 0.9% saline and 3% hydrogen peroxide. A high-speed dental drill with 0.8mm burr was used to drill holes for skull screws and craniotomies over sensory and motor cortices (coordinates below). A single bone-tapping stainless steel screw was inserted securely into the skull, serving as the ground and reference for the recording probe.

Each probe was coated with CM-DiI (Vybrant DiI, catalog #V22888, Thermo Fisher Scientific, USA) by repeatedly drawing a 2 µL droplet of CM-DiI solution at the tip of a micropipette up and down the length of the probe until the liquid dried, slightly changing the coloration of the

shank. Subsequently, probes were, with electrical contacts facing up, stereotactically inserted into either the left hemisphere with a ~65° backward tilt in the sagittal plane to penetrate tissue from the posterior parietal cortex to primary motor cortex ($n$ = 4, AP: − 3.5 to − 4.5 mm, ML: 1.9 to 2.7 mm) or, in the right hemisphere, with a ~50° lateral tilt in the coronal plane to target secondary and primary visual and auditory cortices ($n$ = 3, AP: − 5.52 to − 6.5 mm, ML: 2.1 to 2.5 mm; see Supplementary Fig. 1 for bregma-relative insertion coordinates of each probe). Probe insertions ranged from 3.9 to 7.2 mm in length across animals (Supplementary Fig. 1). External reference and ground wires were mechanically attached to a single skull screw (positioned at AP: +7 mm, ML: +2 mm) and sealed with a drop of SCP03B (*i.e.*, silver conductive paint). The remaining probe outside the brain was air sealed with a silicone elastomer (DOWSIL 3–4680 Silicone Gel Kit) and bead-sterilized Vaseline, and shielded by custom-designed black plastic housing to accommodate probes positioned at intended angles. Finally, the implant was statically secured with black-dyed dental cement to minimize light-induced electrical interference during recordings. Following surgery, rats were subcutaneously administered fluids and postoperative analgesics and placed in a 37 °C heated chamber to recover for 1–2 h prior to recordings.

## In vivo electrophysiology and behavior

Electrophysiological recordings were performed using Neuropixels 1.0 acquisition hardware, namely the National Instruments PXIe-1071 chassis and PXI-6133 I/O module for recording analog and digital inputs[103]. Implanted probes were operationally connected via a headstage circuit board and interface cable above the head. Excess cable was counterbalanced with elastic string which allowed animals to move freely through the entire arena during recordings. Data were acquired with SpikeGLX software (SpikeGLX, Janelia Research Campus), with the amplifier gain for AP channels set to 500×, 250× for LFP channels, an external reference and AP filter cut at 300 Hz. In every session, signal was collected from all channels in the brain, typically from the most distal 384 recording sites (bank 0) first, followed by the next 384 recording sites (bank 1), consecutively.

Behavioral recordings were performed as individual rats foraged for food crumbs (chocolate cereal or vanilla cookies) scattered randomly into an octagonal, black open-field arena (2 × 2 × 0.8 m), with abundant visual orienting cues above and around the arena. All rats underwent a habituation phase prior to surgery during which they were placed on food restriction (to a minimum of 90% pre-deprivation body weight) to stimulate foraging behavior and were allowed to explore the arena daily. They were also acquainted and accustomed to the white noise presentations explained below. Food restriction was halted one day prior to surgery and recordings, by which time the animals were familiar with the environment. The entire data set for each animal was collected during 7–8 recording sessions (20 min each) within the first 12 h ($n$ = 5) or 72 h ($n$ = 2) after recovery from surgery. The experiments were divided into two 4-session schedules in which recordings were made from bank 0 and bank 1, respectively. Each schedule consisted of the same ordering of conditions (light, dark, weight, and light/sound session). Each schedule started with a "light" session, where animals were run in dim lighting, followed by a "dark" session, in which all sources of light were either turned off or covered with fully opaque materials. Then, at the start of the "weight" session, a small copper weight (15 g) was attached to the animals' implants before neural data was acquired. The last session of each schedule was either a "light" session or, when recording from auditory cortices, a "sound" session. During the latter, room lights were dimmed and white noise (5s duration) was played throughout the session at a pseudo-random inter stimulus interval (>10 s ISI) by a Teensy 4.0 Development Board controlled miniaturized Keyestudio SC8002B Audio Power Amplifier Speaker Module, running on custom-developed code. Between each schedule animals were returned to their home cage to rest.

## Perfusion and magnetic resonance imaging (MRI)

After recordings were completed rats received an overdose of Isoflurane and were perfused intracardially with saline and 4% paraformaldehyde. The probe shanks remained in the brains to give enhanced contrast and visibility during subsequent MRI acquisition. MRI scanning was performed on a 7T MRI with a 200 mm bore size (Biospec 70/20 Avance III, Bruker Biospin MRI, Ettlingen, Germany); an 86 mm diameter volume resonator was used for RF transmission, and a phased array rat head surface coil was used for reception. Brains were submerged in fluorinert (FC-77, 3M, USA) to remove background signal on the MRI. A 3D T1 weighted FLASH sequence was acquired at 0.06 mm³ resolution (TE: 10 ms, TR: 40 ms, NA: 4, matrix size: 360 × 256 × 180, FOV: 21.6 mm × 15.4 mm × 10.8 mm, acquisition time: 2 h 20 min).

## Histology and immunohistochemistry

After MRI scanning the shanks were carefully removed and brains were transferred to 2% dimethyl sulfoxide (DMSO, VWR, USA) for cryoprotection for 1–2 days prior to cryosectioning. All brains were frozen and sectioned coronally in 3 series of 40 µm on a freezing sliding microtome (Microm HM-430, Thermo Scientific, Waltham, MA). The first series was mounted directly onto Superfrost slides (Fisher Scientific, Göteborg, Sweden) and stained with Cresyl Violet. The second series was used to visualize Neuropixel tracks, labeled with CM-DiI, against neuronal nuclear antigen (NeuN) immunostaining, which provided ubiquitous labeling that enabled delineation of cortical layers. For immunostaining, tissue sections were incubated with primary anti-NeuN antibody (1:1000 dilution; catalog no. ABN90P, Sigma-Aldrich, USA), followed by secondary antibody-staining with Alexa 647-tagged goat anti-guinea pig antibody (1:300 dilution; catalog no. A21450, Thermo Fisher Scientific, USA), after which the sections were rinsed, mounted, coverslipped and stored at 4 °C. A more detailed immunostaining protocol is available per request. The third series of sections were collected and kept for long-term storage in vials containing 2% DMSO and 20% glycerol in phosphate buffer (PB) at − 20 °C. Using a digital scanner and scanning software (Carl Zeiss AS, Oslo, Norway), all brain sections were digitized using appropriate illumination wavelengths. The images were visualized with ZEN (blue edition) software and used subsequently along with MRI scans to locate recording probes in each brain.

## Probe placement

MRI scans were taken to locate the probes in 3D and to calculate the angle of each probe in the dorsal-ventral (DV) and medial-lateral (ML) axes. Since MRI scanning and histological staining were performed after perfusion with PFA, which cause a non-uniform reduction in brain volume, we reasoned the probe terminus would appear to have penetrated further in the tissue than was actually implanted. We therefore estimated the length of the implanted probe using the number of recording channels in the brain during the experiments. To locate the channel at which the probe exited the brain, we Fourier-transformed the median subtracted local field potential (LFP) signals at each channel along the probe and calculated power differentials between adjacent channels in the lower range frequencies ( < 10 Hz). We then located the largest shift in power between successive channels, which we identified as the point of exit from the brain (this analysis was adapted from the Allen Institute's Modules for processing extracellular electrophysiology data from Neuropixels probes). The final length estimate for each probe was based on the identified surface channel and the physical geometry of the probe.

Probe placement was reconstructed in 3D (Supplementary Fig. 1) by first locating the entry point of the probe in the brain in CM-DiI-stained histological sections and their corresponding MRI scans. Given the probe length (calculated as elaborated above) and angles of the inserted probe relative to the tissue in different planes, we used

trigonometry to calculate the rostral terminus of the probe (Supplementary Fig. 1). Anatomical coordinates were obtained by overlaying images of histological sections on corresponding sections from the rat brain atlas[35]. Probe tracks in the left and right hemisphere were followed from one coronal section to the next until the expected tip of the probe was reached, and area boundaries from the atlas were applied to determine the span of the probe in each brain region (gray line, Supplementary Fig. 1d). Using the within-region span and angle of each probe, we calculated the length, in micrometers, of each probe in each brain region in 3D, which allowed us to determine the number of channels in each region (with two channels spaced every 20 μm).

### Spike sorting and determining the spiking profile of single units

Given that the sessions were recorded in close temporal proximity, raw signals from recording files in each schedule (4 sessions) were concatenated in a unitary binary file, in order to keep the identity of each cluster across sessions. Spike sorting was performed with Kilosort 2.0 software using default parameters, followed by manual curation in Phy 2.0, where *noise* clusters were additionally separated from *good units* and *multiunit activity* based on inter-spike interval distributions, waveform features and the value of the Kilosort contamination parameter. Furthermore, good units were split into fast-spiking (FS) and regular-spiking (RS) subtypes by performing K-means clustering (where $k = 2$) on spike width, peak-to-trough ratio, full width at half maximum and hyperpolarization (or end) slope data (Supplementary Fig. 2).

### 3D tracking, head-mounted accelerometer, and animal model assignment

The rats were tracked with seven retroreflective markers: four 9.5 mm spheres were affixed to a rigid body attached to the head (OptiTrack, catalog no. MKR095M3-10; Natural Point Inc., Corvallis, OR, USA), and three 9 mm circular cut outs of 3M retroreflective tape (OptiTrack, catalog no. MSC 1040) which were affixed to cleanly shaved locations on the trunk[43]. Their precise positioning was optimized to minimize interference in picking up signals from individual markers. 3D marker positions were recorded at 120 fps with an eight camera (seven infrared and one B/W) 3D motion capture system (OptiTrack, Flex13 cameras & Motive v2.0 software). Additionally, a 9-DOF Absolute Orientation IMU [Inertial Measurement Unit] Fusion Sensor (Adafruit,BNO055) was affixed to the implant chamber, such that angular velocities could be sampled directly and compared to tracking-derived features. The IMU (accelerometer) data was acquired via custom-developed code through serial port terminal freeware (CoolTerm 1.7) at 100 Hz via another Teensy 4.0 Development Board, upsampled to 120 Hz post hoc, and rotated to match the reference axes defined by the tracking system. For precise alignment of acquired data streams, three additional infrared LED light sources were captured by the motion capture system. LED flashes (250 ms duration; random 250 ms ≤ IPI ≤ 1.5 s) were controlled by an Arduino Microcontroller C++ code which generated unique sequences of digital pulses transmitted to different acquisition systems throughout the recording and save the IPIs via serial port terminal freeware (CoolTerm 1.7). The detailed model assignment procedure has been described previously[43]. Briefly, all seven individual markers associated with the animal were labeled in a semi-supervised way using built-in functions in Motive. A rigid body was created using 4 markers on the head, and three markers on the body were labeled as separate markers. In addition to the markers on the animal, the three synchronizing LEDs were labeled as a separate rigid body (only marker sets with fixed distance over time can be labeled as a rigid body). After each session was fully labeled, remaining unlabeled markers were deleted and data were exported as a CSV file. The CSV file was converted to a format (pickle) compatible with our in-house graphical user interface[43] for reconstruction of the coordinate system of the head from tracked points. Finally, tracking data was then merged with spiking data for further processing.

### Extracting postural variables from tracking data

Following the recordings, we labeled tracked points within the Motive (OptiTrack) interface, and imported labeled data into a custom script in Fiji. Using the four tracked points on the animal's head, the geometry of the rigid body was estimated using the average pairwise distances between markers. We found the time point at which this geometry was closest to the average and used that time point as the template. We then assigned an XYZ coordinate system to the template with the origin located at the centroid of the four points, and constructed coordinate systems at each time point of the experiment by finding the optimal rigid body transformation of the template to the location of the head markers. To find the likely axis of rotation for the head (i.e. the base of the head), we found the translation of the coordinate system that minimized the Euclidean distance between the origin at time point $t−20$ and $t+20$, where $t$ is measured in frames from the tracking system (120 fps). Next, the coordinate system was rotated to most closely match the $Z$-direction with the vertical direction of the room, and X-direction with that of the running direction, which was defined by horizontal movements of the origin from t-50 to t+50. Time points where the speed exceeded 10 cm/s were used to center the coordinate systems for the animals' head direction and running direction. The two objectives were combined by considering the sum of squared differences of the two sets of angles. This definition of running direction was used only to rotate the head direction, and was not used in subsequent analyses. Hyperparameters were chosen such that head placement using the resulting coordinate system visibly matched experiments.

To compute the postural variables for relating tracking to neural activity, we first denoted the allocentric angles of the head (pitch, azimuth and roll) relative to room coordinates, computed assuming the $XYZ$ Euler angle method. The $XYZ$ Euler angle method indicates the three elemental rotations are intrinsic rotations about the axes of the rotating coordinate system $XYZ$, solidly with the moving body, which changes its orientation after each elemental rotation. We next denoted body direction as the vector from the marker above the root of the tail to the neck point. The egocentric angles of the head (pitch, azimuth and roll) relative to body direction were then computed assuming the $XYZ$ Euler angle method. The back angles (pitch and azimuth) were determined relative to the horizontal component of body direction using standard 2D rotations, which were optimally rotated such that the average peak of occupancy was close to zero. The point on the neck was then used to determine neck elevation relative to the floor, as well as the horizontal position of the animal in the environment. Movement variables were estimated from the tracked angles using a central difference derivative with a time offset of 10 bins. Running speed was then estimated using a moving window of radius 250 ms. The values for self-motion were computed as the speed of the animal multiplied by the $X$ and $Y$ component of the difference in angles between the body direction at $t−15$ and $t+15$.

### Tuning curves to posture, movement and navigational variables

Angular behavioral variables were binned in 5°, with the exception of back angles, which were lowered to 2.5°. Movement variables were binned in 36 equally spaced bins, spanning the range of recorded variables such that there was a minimum occupancy of 400 ms in both the first and last bins. Neck elevation bins were 1 cm, while position in the environment was estimated using 6.67 cm bins. Finally, self-motion used a bin size of 3 cm/s. For all rate maps, the average firing rate (spk/s) per bin was calculated as the total number of spikes per bin, divided by total time spent in the bin. All smoothed rate maps were constructed with a Gaussian filter with a standard deviations of 1 bin. Only bins with a minimum occupancy of 400 ms were used for subsequent

analysis. To shuffle receptive field distributions, we shifted the neural activity 1000 times on the interval of ±[15,60] s and recomputed tuning curves for each variable. For each cell, tuning curve stability was calculated for the feature which had the highest mutual information[104] with that cell's spiking activity. Tuning curves for which the Pearson correlation across even and odd minutes of the recording was higher than the 95th percentile of the shuffled distribution were considered stable.

## Defining composite actions

The behavioral clustering pipeline is sketched in Fig. 1b and adapted from prior work[25]. The starting point is the time series of postural parameters and the running speed of the animal, Fig. 1b (panel 1). Running speed, neck elevation and back Euler angles (pitch and azimuth) were defined as explained above (see "Extracting postural variables from tracking data"), while 3D head direction relative to the body direction was parameterized using the exponential map[105]. The time series of each of the 7 variables (6 postural parameters plus running speed) was detrended using third degree splines with equally spaced knots at 0.5 Hz, Fig. 1b (panel 2). Time frequency analysis was then performed on the detrended time-series for each of the original variables using Morlet wavelets, at 18 Fourier frequencies dyadically spaced between 0.5 and 20 Hz, Fig. 1b (panel 3). The square root of the power spectral density was centered and rescaled dividing it by the variance of the smoothed signal (fit resulting from the spline interpolation). The smoothed signal was $z$-scored and concatenated with the rescaled spectrogram yielding a 133D feature vector for each tracked time point (120 Hz). Feature vectors were downsampled to 1 Hz, and data were pooled across animals and conditions. Downsampling was performed only for embedding, and did not affect behavioral analyses or correlations with neural activity (e.g. encoding, decoding), which used 120Hz tracking data. Redundant dimensions were removed using Principal Component Analysis (PCA), which indicated that the first 22 principal components explained 97.2% of the variance. Only these 22 principal *eigenmodes* were retained and the dimensionality was further reduced to 2 via tSNE[106] embedding (Euclidean metric, perplexity=200), Fig. 1b (panel 4). The embedding was then used to estimate a probability mass function (PMF) on a 60x60 lattice in the 2 tSNE dimensions by convolving the raw histogram with a two-dimensional Gaussian (width=1). We segmented the 60x60 lattice by applying a watershed transform[107] to the additive inverse of the PMF, Fig. 1c. All data points falling within a watershed-identified region were assigned the same action label. Timepoints not belonging to the dataset used for PCA were classified by minimizing the Euclidean distance of the feature vector in the 22-dimensional PC space from the datapoints used for training. Names were attributed to the action labels after post-hoc visual inspection of individual timestamps in one session per animal in a graphical user interface (GUI) and by comparison to the postural decomposition of the behavior (Supplementary Fig. 6a, b and Supplementary Videos S1–S6).

## Encoding and decoding of actions

The average firing rate (spk/s) of each cell per attributed label was calculated as the total number of spikes emitted during the action, divided by total time spent in it. Additionally, the average firing rate of the cell was computed separately in two halves of the dataset: the odd vs. even time bins within each action. To compare with shuffled data, we shifted the neural activity 1000 times on an interval of ±[15,60] s. Shuffled distributions were also constructed for each of the two halves of the dataset. A cell was classified as encoding a behavior if these 2 criteria were met: [i] its average firing rate at the behavior on the whole dataset was either (a) below the 0.01th percentile, or (b) above the 99.99th percentile of the shuffle distribution [ii] if (a), its average firing rate for the action in both halves of the dataset was below the 2.5th percentile of the shuffle distribution of each half respectively, while if

(b), its average firing rate for the behavior in both halves of the dataset was above the 97.5th percentile of the shuffle distribution of each half respectively. For [i], the 99% significance level was Bonferroni-corrected for multiple comparison.

Spike counts time series were constructed by counting the number of spikes fired by a cell in each 8.33 ms time bin. Behavioral decoding from spike count data was performed on every session for which more than 10 cells were simultaneously recorded. A naive Bayes classifier was trained on all the actions with an occupancy larger than 16 s, resulting in 34 ± 3 (mean ± SEM) actions per session to be decoded. Decoding was performed on 20 samples of 400 ms each (50 bins) per action, while the rest of the dataset was used for training. The classifier consisted of a binomial likelihood and a categorical prior determined by the occupancy of actions in two different randomly selected sessions of the same animal. We defined the decoding accuracy for an action as the fraction of samples whose label was correctly classified. For comparison we also classified actions using only the prior distribution.

## GLM and model selection

We binned the spike train of all neurons with 8.33 ms time bins to match the tracking frequency of 120 Hz. Let $y_t, t = 1, \cdots, T$ be the binarized spike count of a neuron in time bin $t$ of a total of $T$ in the whole recording session; $y_t = 1$ indicates that the neuron emits one or more spikes in bin $t$, whereas $y_t = 0$ indicates that the neuron does not fire in bin $t$. The probability of $y_t$ is given by a Bernoulli distribution,

$$y_t \sim f(y_t|p_t) = p_t^{y_t}(1-p_t)^{1-y_t}, \quad y_t = 0 \text{ or } 1 \tag{1}$$

where $p_t$ is the probability that $y_t = 1$. Let $\mathcal{X}_t = \{x_1(t), \cdots, x_m(t)\}$ represent the $m$ tracked and factorized features at time t: nine postural features (pitch, azimuth and roll of the head in allocentric and egocentric reference frames, back pitch and azimuth, neck elevation), their first derivative values, body direction, speed, position and self-motion. For each feature $i$, let $x_i(t)$ be a binary vector of length $N_i$ (number of bins used to factorize covariate $i$: 15 for 1D features; bin size of 5 cm/s for self-motion and 10 cm for position), whose components are all 0, but the one corresponding to the bin in which the features falls at time $t$. To study how well a neuron can be explained by one or more features, we fit the activities of a single neuron using generalized linear models $M$[108] with logit link function,

$$p_t = \text{logit}(X_t^T \beta) \tag{2}$$

where $\beta$ are the parameters of the model and $X_t = (1, x_{M_1}(t), x_{M_2}(t), \ldots, x_{M_n}(t))$ is the vector of $n$ features included in model $M$. Thus the log-likelihood of the model is,

$$l(M|y) = \sum_{t=1}^{T} (y_t X_t^T \beta) - \log(1 + \exp(X_t^T \beta)). \tag{3}$$

We estimated all models with an additional L1 regularization with the learning rate $\lambda = 10^{\backslash -4}$.

To determine which subset of these features best explain the neural activity, we performed a forward selection procedure[109] combined with a 10-fold cross-validation scheme. For each neuron, we first partition the data into 10 approximately equally sized blocks $a = 1, \ldots, 10$, where each block $\{y^a, X^a\}$ consists of consecutive time bins. We then computed the average held-out scores across folds. The initial simple model consisted only of an intercept and features were added sequentially through three-steps:

1. For each feature not included in the model, and each fold $a$, we fitted the GLM with the feature added, $M$, on 9 data blocks and computed the log-likelihood $l(M|y^a)$ for the test data. After iterating over folds we took the average over folds of

$LLR^a = (l(M|y^a) - l(M_0|y^a))/n^a_{spikes}$, where $n^a_{spikes}$ is the number of spikes in the test data of fold $a$, while $l(M_0|y^a)$ is the out-of-sample log-likelihood of the intercept model in fold $a$. We determined which feature had the largest value of the average LLR and selected it as a candidate feature to be included in the model.

2. For the given candidate feature, we employed a one-sided Wilcoxon signed rank test on the out-of-sample log-likelihood across folds $l(M|y^a)$ of the more complex model and the current model. The null hypothesis is that the more complex model yields smaller or equal values of $l(\beta|y^a)$ with respect to the less complex model.

3. If the null hypothesis was rejected ($\alpha = .01$), the new feature was added to the model and the forward selection procedure continued. The selection process stopped when the null hypothesis was not rejected or no more features were available.

After a final model is selected for each cell, we calculated the contribution of each feature belonging to the final model via 10-fold cross-validation. Assuming that the final model $M_{full}$ includes n covariates, for each selected covariate $x_i$, $i = 1, ..., n$, we considered the partial model $M_{x_i}$ which includes all the covariates except $x_i$, and the intercept model $M_0$. Then for each partition $a$ of the data we trained the three models and computed the out-of-sample log-likelihoods $l(M_{x_{full}}|y^a)$, $l(M_{x_i}|y^a)$, $l(M_0|y^a)$. Finally we define the contribution of each covariate to the final model as the relative log-likelihood ratio

$$\text{rLLR}(x_i) = \frac{l(M_{full}|y) - l(M_{x_i}|y)}{l(M_{full}|y) - l(M_0|y)}. \quad (4)$$

where $l(M|y)$ is the average across folds $a$ of $l(M|y^a)$.

We measure the prediction accuracy of a model $M$ relative to the intercept model $M_0$ as the average across folds $a$ of McFadden's pseudo-$R^2$[110]

$$\text{pseudo-R}^2(M)^a = 1 - \frac{l(M|y^a)}{l(M_0|y^a)}. \quad (5)$$

## Anatomical topography of tuning features

Data from the three left hemisphere-implanted animals were used to compute anatomical gradients of behavioral tuning across adjacent brain regions (the fourth rat, #26148, was excluded due to limited anatomical coverage). Since the exact anterior-posterior (AP) placement of the probes differed across animals, the overall extent of the three probes was calculated using the most posterior and the most anterior anatomical locations. This physical distance comprised the rows of a matrix with 3 columns, in which each column represented data from individual animals. Based on where the probe was located in this physical space and how many cells of a particular tuning type (e.g., allocentric head roll) were recorded, multiple matrices were created for each feature. These matrices were then divided into 7 equal segments, each corresponding to 1 mm of tissue, and the numbers of cells tuned to particular features were summed. Since the absolute number of cells varied across animals, the data were presented as a proportion of the total number of cells recorded in a given 1 mm segment. We applied a $\chi^2$ test to determine if the observed distribution of cells tuned to each feature was significantly different from a uniform distribution over the cortical surface. The features with significant differences were plotted in Fig. 4.

All three right hemisphere-implanted animals were used for this analysis. Instead of calculating the absolute spatial extent of the three probes along the medial - lateral (ML) axis, the probes were aligned based on the auditory/visual border. The same approach was applied as above, and the distribution of the proportion of cells tuned to each feature over the anatomical surface was calculated, and significance was assessed using the $\chi^2$ test. Features showing anatomical differences were plotted in Fig. 4.

## Sensory modulation indices and decoding

To obtain peri-event time histograms (PETHs) relative to sound stimuli onset, each spike train was zeroed to tracking start, purged of spikes that exceed tracking boundaries and binned to match the tracking resolution. It was further resampled (to 50 ms bins) to encompass a large window (10 s) before and after every event onset (the start of the white noise stimulation). Spike counts were converted to firing rates (spk/s) and smoothed with a Gaussian kernel (sd=3 bins). To identify sound responsive units, we calculated the sound modulation index (SMI) for each cell on PETHs averaged across all trials (Supplementary Fig. 22b, top). SMI is the difference between the "sound" (500-1000 ms post-stimulation) and the "baseline" firing rate (1000-500 ms pre-stimulation) divided by the sum of the two, such that a negative SMI signifies higher firing before, and a positive SMI signifies higher firing following sound onset. The statistical significance of each SMI was determined with a Wilcoxon signed-rank test ($p < .05$) performed on all "sound" and "baseline" trial sequences.

We used a nearest neighbor decoder to query whether we could predict the sound being "on" or "off" given only auditory or visual ensemble activity, for each rat separately (Supplementary Fig. 22b, bottom). The sound event vector ("on" or "off"), together with the spike train of each single cell, was resampled to 10 Hz resolution and the latter were convolved with a Gaussian kernel (sd=1 bin). In each run (for a total of 100 runs per unit number) we pseudorandomly subsampled either 5, 10, 20, 50 or 100 different cells and divided the data into 3 folds, where each third of the data once served as the test and the other two thirds as the training set. We calculated Pearson correlations between every test set ensemble activity vector and every ensemble vector in the training set. This enabled us to obtain a predicted sound stimulus value ("on" or "off") for each test frame by assigning it the sound stimulus value of the highest correlated training set vector. Decoding accuracy was defined as the proportion of correctly matched stimulus states across the entire recording session (theoretically varying from 0 to 1). To obtain the null-distribution of decoded accuracy we shuffled the spike trains of each subsample in the first run 1000 times (as described above) and followed the same described procedure that resulted in shuffled accuracy distributions.

Since "light" and "dark" conditions were not varied on a trial, but on a session basis, we computed PETHs by first searching for all ≤2 s time windows where the speed of the animal was ≤5 cm/s, effectively equating to quiescence or epochs of slow movement. We did this in three sessions: light1, dark and light2 by subsampling the number of events from the session that had the fewest such events in the other two sessions. The firing rates (spk/s) in each window bin were calculated using the same method described above. To identify luminance responsive units, we calculated the luminance modulation index (LMI) for each cell on PETHs averaged across all trials (Supplementary Fig. 22c, top). LMI is the difference between the "dark" (full 2 s window) and the "light1" firing rate (full 2 s window) divided by the sum of the two, such that a negative LMI signifies higher firing in light conditions, and a positive LMI signifies higher firing in the dark condition. The statistical significance of each LMI was determined with a Wilcoxon signed-rank test ($p ≤ .05$) performed on all "dark" and "light1" trial sequences, provided that the same test yielded no difference in firing rates ($p > 0.05$) between "light1" and "light2" conditions. To visualize these differences, we concatenated the population vectors (all recorded cells in A1 or V) of all three sessions (in one example animal), z-scored them, and performed the principal component analysis (PCA). We determined the vertex (or the "knee") of the scree plot and selected all components preceding it for non-linear low-dimensional

embedding of individual timepoints with UMAP (Supplementary Fig. 22a, bottom).

To determine the relative strength of the sound and luminance modulation along the recording probe, we counted all significantly modulated units (both suppressed and excited) at their respective peak channels, joined all channels of 2 successive rows in one count (totaling 4 channels every 2 rows), and normalized this count by the maximal count obtained.

The nearest neighbor decoding was also adjusted to accommodate for the lack of a trial based structure. First, due to the fact that the secondary auditory area (A2D) had multisensory properties, i.e., units which were sensitive to both sound and luminance change, we focused our analysis on primary auditory (A1), together with all recorded visual (V1, V2L and V2M) neurons. The data were downsampled and smoothed as described above, and synthesized by taking the last quarter of timepoints in the light1 session, the temporally adjacent first quarter of timepoints in the dark session and the temporally distant second half of the light2 session. Similarly to the sound decoder, in each run (for a total of 100 runs per unit number) we pseudorandomly subsampled either 5, 10, 20, 35 or 50 different cells (adjusting for the lower total number of cells in A1). For each test set ensemble activity vector (i.e., the second half of light2) we computed Pearson correlations to every ensemble vector in the training set (light1 + dark) and obtained a predicted condition status by assigning it the condition status of the most highly correlated training ensemble vector. Since luminance did not change within a session, shuffling spike trains would not suffice (because it would not eliminate the overall lower/higher rate relative to the other session), we randomly permuted the ensemble vectors across the training set (i.e., light1 + dark) at each time point 1000 times in the first run which resulted in the null-distribution of decoded accuracy.

### Weight and behavioral tuning

To ascertain whether weight had a behavioral effect on the measured variables, we primarily focused on head-related features, neck elevation and speed, assuming these would be affected the most. For each feature we computed differences between the total occupancy in every bin between the weight and light2 sessions, across all rats and looked whether the 99% CI of these differences overlapped with zero.

To estimate whether adding the weight on the head had any affect on the neural coding or activity, we performed several analyses. Since our recordings were performed over multiple successive sessions, we analyzed the change in the overall activity of spiking through time. Therefore, in each cluster, spikes are allocated to broad 10 s bins and smoothed with a Gaussian kernel (sd=1 bin). They were then concatenated into a single array and a rolling mean (size=50 bins) was calculated over the whole window for display purposes. The "baseline" firing rate was defined as the total spike count within a session divided by the total session time. A "stable" baseline rate was the weight session rate above .1 spk/s whose difference to the reference session rate (light2 sessions were picked as reference sessions as overall rates tended to be more similar to the weight sessions) was smaller than 20% of its own rate. To obtain the shuffled distribution of rate differences, we pseudorandomly permuted individual cells' rate identities across light1, weight and light2 sessions 1000 times. Our subsequent analyses focused on the effect of weight on different tuning features, namely the observed differences in areas under the tuning curve (AUC), the observed differences in information rate[104], the observed differences in the stability of tuning curves, and the observed differences in tuning peak positions. To determine whether any of these difference distributions were significantly different compared to a null-distribution, we created shuffled distributions of differences by pseudorandomly permuting session identities of the data 1000 times and recomputing the differences.

### Functional connectivity

Spike trains were binned in .4 ms wide bins and dot products (cross-correlograms, CCG) were computed between every spike array and any other jointly recorded spike array with temporal offsets spanning the [−20, 20] ms range with 0.4 ms steps. To generate a low frequency baseline cross-correlation histogram for comparison, the observed CCG was convolved with a "partially hollowed" Gaussian kernel[11], with a standard deviation of 10 ms, and a hollow fraction of 60%. The observed coincidence count (CCG) is compared to the expected one (low frequency baseline) which is estimated using a Poisson distribution with a continuity correction, as previously described[112]. A putative connection was considered synaptic if the following conditions were met: (1) 99.9999/0.0001 (for excitatory/inhibitory connections, respectively) percentile of the cumulative Poisson distribution (at the predicted rate) was used as the statistical threshold for significant detection of outliers from baseline, (2) two consecutive bins needed to pass the threshold within the ±1.6−4 ms window[113], and (3) there should be no threshold passing in the ±1.6−0 ms range. Alternatively, if the peak/trough occurred in the ±1.6−0 ms range, and two consecutive bins passed the threshold for detecting outliers, the units were considered as receiving common input. A total of total of 557,620 possible connections were considered in the right hemisphere (auditory and visual cortices), and 606,301 possible connections were considered in the left hemisphere (somatosensory and motor cortices).

The 3D position of each neuron in a connected pair was determined by first computing its center of mass on the probe surface, based on peak absolute template waveform amplitudes on the peak waveform channel and 20 adjacent channels below and above the peak. The exact DV, ML and AP positions were then computed taking into account the insertion site, the total length of the probe in the brain and its angles in the tissue, as explained above. Synaptic strength was defined as the absolute difference between the spike coincidence count at the CCG peak/trough (for excitatory/inhibitory connections, respectively) and the slow baseline at peak/trough, normalized by the minimum number of spikes between the two spike trains (i.e., the theoretical maximum number of coincidences).

All neurons in the dataset were assigned with a variable that best fit its spiking variability in the dark session, based on the mean cross-validated relative log-likelihood ratio (rLLR) of single covariate models relative to the null model (23 behavioral features + unclassified cells). The "functional space" map was obtained for visualization purposes only by performing PCA on a matrix containing such values for all 23 covariates, in addition to the SMI and LMI estimates and p-values. The first n components that cumulatively accounted for 90% of the variance were then embedded on a 2D plane by uniform manifold approximation and projection (UMAP). The 24 feature list was further simplified by grouping variables in 11 categories: unclassified, position, speed-related, egocentric head posture, egocentric head movement, allocentric head posture, allocentric head movement, back posture, back movement, neck elevation, neck movement. Variables were plotted on log-log scales for visualization purposes only, but all presented statistics (Mann–Whitney U test was chosen, as Levene's test established groups had unequal variances) were performed on the original data, and "functional distance" was calculated across the original 28 variables (23 covariates + SM and LM indices and p-values).

Excitatory and inhibitory connection pairs were classified in 6 broad functional categories: (1) movement modulated neuron preceding a sensory modulated cell (either sound or luminance, for auditory and visual ensembles, respectively), (2) posture modulated neuron preceding a sensory modulated cell, (3) movement modulated neuron preceding a movement modulated cell, (4) movement modulated neuron preceding a posture modulated cell, (5) posture modulated neuron preceding a movement modulated cell, and (6) posture modulated neuron preceding a posture modulated cell. Assessment of whether the connection pair numbers in each category could have

been observed by chance was done by subsampling pseudorandomly paired units 1000 times, provided that: (1) the anatomical distance between cells was shorter or equal to the maximal one observed in the true data, (2) there were equal numbers of excitatory and inhibitory connections as in the real data in each run, (3) the connection was physiologically plausible (excitatory/inhibitory connections could only be formed in the RS/FS cell was the presynaptic neuron, respectively). Lastly, we statistically tested whether the observed difference in neuron counts between subregion categories (either A1-A2D, or V1-V2L) was larger than theoretically expected by an equal split (50–50%), using the $\chi^2$ goodness-of-fit test if the expected frequencies exceeded 5 in each category, or the binomial test otherwise.

### Reporting summary

Further information on research design is available in the Nature Portfolio Reporting Summary linked to this article.

## Data availability

All data generated in this study are available in a fig**share** database under the accession code https://figshare.com/articles/dataset/Rat_3D_Tracking_E-Phys_KISN_2020_Dataset/17903834. Source data are provided with this paper.

## Code availability

The code pertaining to the experimental pipeline for data acquisition and preprocessing can be found at https://github.com/bartulem/KISN-PyLab[121]. The code used to analyze the data and make the figures can be found at https://github.com/bartulem/KISN-pancortical-kinematics. All codes are publicly available.

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

## Acknowledgements

We thank D. Deutsch, A.L. Falkner, M. Long, S.R. Datta and M. Murthy for helpful comments on the manuscript; M. Andresen for tissue sectioning; R. Gardner, T. Waaga and V. Normand for the initial assistance with Neuropixels recordings; W. Zong for assistance in designing the 3D printed cap housing the Neuropixels probes; S. Gonzalo Cogno, R. Gardner, V. Normand, T. Waage, M. Nigro, A. Lauer and J. Bizley for fruitful discussions along the way; K. Haugen and H. Waade for technical and IT assistance; M. Witter and M. Nigro for assistance with anatomical delineations; D. Hill and M. Widerøe of the Norwegian University of Science and Technology (NTNU) MRI Core Facility and S. Eggen for veterinary oversight. This project used the high performance computing infrastructure, IDUN, provided through the NTNU[114]. We also wish to express our gratitude towards individuals, teams, organizations and funding programs behind fundamental packages for scientific computing and visualizations in Python which relieved our workload substantially: Matplotlib[115], Numba[116], NumPy[117], Pandas[118], Scikit-Learn[119], and SciPy[120]. This work was supported by a Research Council of Norway FRIPRO (grant #300709) to J.R.W., the Centre of Excellence scheme of the Research Council of Norway (Centre for Neural Computation, grant #223262), the National Infrastructure scheme of the Research Council of Norway NORBRAIN (grant #197467), The Kavli Foundation and the Department of Mathematical Sciences at NTNU (to B.A.D.). The MRI core facility is funded by the Faculty of Medicine at NTNU and Central Norway Regional Health Authority.

## Author contributions

B.M. and T.T. contributed equally to this work. C.B. and J.G.F. contributed equally to this work. Conceived the project: B.M., T.T., J.R.W.; funding acquisition: B.A.D., J.R.W.; methodology: B.M., T.T., C.B., B.A.D.;

data collection: B.M., T.T.; data analysis: B.M., T.T., C.B., J.G.F.; software: B.M., C.B., J.G.F.; writing (original draft): B.M.; writing (editing): B.M., T.T., C.B., J.G.F., J.R.W.

## Funding

## Competing interests
The authors declare no competing interests.
