## [Peer Review File · Nature Communications]

Behavioral decomposition reveals rich encoding structure employed across neocortex in ratsREVIEWER COMMENTS

Reviewer #1 (Remarks to the Author):

The authors chronically recorded simultaneously from either motor and somatosensory (left cortex; N=4) or auditory and visual cortical areas (right cortex; N=3) with neuropixels probes in freely foraging rats while monitoring their movements with an array of imaged 3D markers and accelerometers. Recording locations were confirmed by decoding analysis of passive stimulation epochs, and a post-mortem combined MRI and immunohistochemical / histological approach. All data were resampled and aligned at 120 Hz, then detrended and decomposed by Morlet wavelet. T-distributed stochastic neighbor embedding (t-SNE) was then used to identify 44 recurring action classes / behavioral motifs, and generalized linear models (GLM) used to determine which components of neural and inferred synaptic activity best predicted the action classes at each time point (binned at 8.33 ms or 1/120Hz). Just over half of neurons in all recorded areas predicted each action above chance. However, components of movement, posture, and self-motion were differentially represented across the four main cortical areas recorded from. Visual cortex represented horizontal scanning head movements and head position, auditory cortex represented gravity relative head position, motor cortex represented overall body and back position, and somatosensory cortex represented back position posture but also had a large proportion of “unclassified” neurons. Putative synaptic connections were identified by statistically enriched cross-correlogram peaks or troughs (i.e. excitatory or inhibitory). Synaptic connections within visual cortex tended to connect neurons with homogenous coding of behavioral subcategories (i.e. movement, posture, self-motion), although there was also an enriched probability of excitatory connections from posture encoding neurons onto movement encoding neurons, and of inhibitory connections from movement neurons onto posture neurons (feedback inhibition motif?). Movement neuron inhibition of posture neurons was the dominant motif in auditory cortex, and pairs of neurons in general that received common input (i.e. CCG peak at ~0 ms) tended to encode the same class of behavioral components.

Overall this study seems to be taking the right approach to better understand the cortical neural encoding of spontaneous, ethological behaviors. However, the main findings come across as somewhat preliminary and disjoint, lacking an overall cohesive message. Furthermore, confidence in the main findings in terms of differential action encoding across cortical areas would be enhanced if recordings were simultaneous across all four main areas, and if comparable data existed for these areas in terms of the cortical layer/depth from pia distribution of recorded neurons. Inclusion of data regarding fine facial movements, which are known to be directly indicated arousal state and strongly predict cortical neural activity, would likely strengthen the validity of behavioral action classification. Lastly, use of longer recording sessions would likely enhance the validity of behavioral action transition matrices and allow for observation of the full span of the hierarchy of arousal states / behavioral actions. Were some or all

of these concerns to be addressed by the authors, this reviewer may recommend this study for publication. Following are some additional major and minor points of critique.

Major Points:

1.-The identified behavioral actions are too granular and brief, perhaps missing organization from one level higher in the presumed hierarchy of behavioral organization. This can be seen as the lack of apparent sensible organization of the behavioral action transition diagram (Extended Figure 3c).

2.-tSNE organization of actions seems to have inherent organization unmentioned by authors – three vertically oriented strips (left, middle, right) seem to encode arousal level / locomotion speed, rightward orienting postures and movements, and leftward postures and movements, respectively

3.-Explanation for why auditory cortical areas would primarily encode gravity relative head movements is not convincing.

4.-The main finding that visual areas strongly encode head movements and head posture seems to not be novel, and may merely represent subtraction of changes in the visual scene due to self-movement, rather than representation of ethological behaviors in these areas, per se.

5.-Examination of recurring synaptic motifs across cortical areas is extremely interesting, but the n used here are perhaps insufficient to make strong statistical conclusions concerning their relative prevalence. In addition, showing that manipulation or modulation of these synapses during particular behavioral actions and/or transitions would greatly contribute to the cohesiveness of the main findings of the study.

6.-Extracted behavioral primitives do a relatively poor job of explaining neural activity in somatosensory cortex (Fig 2D), and there is no good explanation for this. Is this due to a different layer recording position in somatosensory cortex, or to the lack of inclusion of fine whisker and facial movements in the overall model, for example?

7. Could the differences between areas result from different neurons being recorded? What if the same neurons (e.g. regular spiking cells in layer 5) were compared between areas – does that affect the results? What are the differences between layers within the same area? Are these differences similar to differences between areas?

8. How much of the widespread nature of activity related to behavior simply the result of the animal experiencing widespread changes in multiple inputs during behavior? E.g. when a animal moves, even in the dark, there is large scale changes in motor, somatosensory, auditory, vestibular, eye movement, etc. signals.

Minor Points:

-It's not clear what the partial foraging paradigm adds to the study, other than to ensure a minimum amount of locomotion during the recordings. Also, it seems that the probability of naturalistic behaviors not directly related to foraging may have been decreased by this aspect of the experimental design.

-The vagueness of the final sentence of the abstract speaks to the overall suggestive nature of the results and the lack of a cohesive overall finding.

-More in-depth treatment of behavioral action transition times may allow for analysis of activity motifs and synaptic activity that predict these transitions.

-It's not clear if the authors looked for inter-areal synaptic connections (i.e. visual to auditory), or if they focused only on intra-areal connections (i.e. A1 to AuD).

-Use of alternative manifold embedding techniques, such as UMAP (see "BSOiD", Hsu et al, 2021), may allow for better inter-subject identification of high-level behavioral motifs.

-Is it possible to achieve a clearer alignment between the identified elementary poses and species typical actions with assigned ethological meaning?

-A clearer explanation of the rationale for use of relative log likelihood ratio as a principal metric for assessing the degree to which each elemental behavioral is represented in each cortical region would be helpful to the general readership.

-Use of a 15 g head weight to control for the possible encoding of muscle contractions as proxies for head position seems unnecessary and perhaps a bit cruel.

-It would have been nice to provide passive somatosensory stimulation so that a hierarchical examination of behavioral vs. sensory encoding (see Fig 3e) could have been performed in somatosensory cortex as well as auditory and visual cortices, particularly given the preponderance of unclassified neurons in S1.

-It is not specifically mentioned that the activity of visual cortex neurons could not be used to decode the presence or absence of the white noise stimulus, although it is mentioned that the activity of auditory cortex neurons could not be used to detect the luminance condition.

-Has it already been shown in other studies that back movements tend to be represented in caudal motor and somatosensory cortex, while head movements tend to be represented in corresponding rostral areas? If this is a novel finding, perhaps it warrants more discussion (page 3).

-Does 12% seem like a low percentage of neurons in auditory cortex to be modulated “exclusively” by sound? What is the corresponding percentage for all other recorded areas?

-The increased prevalence of “movement inhibited posture units” in auditory cortical areas seems to be of particular interest, and to merit further discussion, particularly in terms of the ability of the model to explain cortical activity during periods of low movement / low arousal.

-Is it possible that the observed increased probability of synaptic connection between neurons with heterogeneous behavioral encoding in “V2L” may actually be due to the mis-assignment of neurons that are actually in posterior parietal cortex (PPC)?

-Is it trivial that pairs of neurons receiving common synaptic inputs appear to be homogeneously tuned for behavior (see final point in results)? Or, is this merely an artifact of the analysis? Please provide brief additional discussion on this point.

-Do your results suggest that studies not using video of the back and hindlimbs of rodents will have dramatically insufficient ability to account for variance in neural activity of primary somatosensory cortex? (See Discussion, paragraph 2). Please provide brief additional discussion on this point.

-Separation of regular -and fast- spiking units (Extended data Figure 2) based on spike duration, peak-to-trough ratio, and end-slope is not very convincing.

-Was head position really only updated when the rat was running faster than 10 cm / s (Methods, page 12)? This seems like too high of a threshold and likely results in ignoring many real changes in head position in the overall model.

-Use of the acronym “IMU” for “inertial measurement unit” is unnecessarily confusing. These instruments are typically referred to as “accelerometers”.

-Please provide a definition of “XYZ Euler angle method”.

-Behavioral feature vectors seem a bit overconstrained with 133 dimensions (Methods, page 13). Comparison of results after removal of redundant dimensions may be informative.

-Downsampling of feature vectors to 1 Hz (Methods, page 13) may be excessive and prevent proper high-resolution alignment of neural activity with behavioral action transitions. A comparison with results after downsampling to 10-30 Hz, a range that better matches the frequency components of natural rodent movement, would be informative.

The authors often state that movement drives or influences activity – however, this is only a correlation. I recommend the authors use more circumspect language such as “movement is associated with....”.

The results from the fast spiking category of cells is commented upon in the discussion but I couldn't find it presented in the main body of the results section.

Reviewer #2 (Remarks to the Author):

General comments:

The goal of the study is to examine the neural representation of natural behaviors - both segmented 'momentary behaviors' (e.g., rearing, turning), and more "elementary" behavioral features (posture, movement, position, etc.). The reported findings are that (1) momentary behaviors were represented across all areas recorded, (2) behavioral features showed organization across areas, and (3) tuning properties of synaptically coupled cells differed across areas. The authors collected high-quality video recordings of freely behaving rats while recording single unit activity across multiple brain areas using Neuropixels probes. The methodology to quantify behavior appears sound and consistent with literature, and the statistical methodology behind fitting the array of decoders and GLMs in this paper appears sound.

The main weaknesses in the paper are that the primary evidence used to support claim (b) is based on models that appear to have very low performance in capturing neural responses and further that the differences in recording depth across different areas could be confounding the conclusions. Second, the data and analysis supporting claim (c), as presented in Figure 4, are somewhat weak. Finally, too much of the substance of the paper is in the Extended Data, making it very challenging to grasp the significance of the results as presented. This is a novel and extensive dataset, and the results of these analyses are of interest, but there are serious shortcomings in its current form.

Strengths of the paper:

It is clear that a tremendous amount of work went into acquiring, processing, and analyzing the data shown here, and the thoroughness of the documentation of technical details through the extended data figures is excellent. Figures 1 and 3 clearly convey results of interest.

To highlight some of the strengths in these parts of the paper: in Figure 1, the authors outline the methodology for segmenting natural behavior of freely moving rats, which produces a map of identifiable momentary behaviors (center of panel (c)). These momentary behaviors can be decoded from neural activity in all the brain areas recorded, with varying degrees of accuracy. Most momentary behaviors are decoded above a chance-level set by the prior distribution of momentary behaviors, some are decodable with very high accuracy. This is interesting, in that most previous decoding analysis have not focused on representation of a wide range of specific actions, and the fact that specific actions are represented is a novel finding. Figure 3 shows how visual inputs and auditory inputs modulate activity, and that neurons across both areas have similarly broadly distributed representation of behavior variables and sensory variables. This broad representation of behavioral alongside sensory information is important for understanding what cortical computation is – at the most basic level, the inputs and the outputs of these computations are more complex than typically appreciated.

Detailed comments on major concerns:

1. Regarding evidence supporting claim (b)

The analysis supporting claim (b) is primarily in Figure 2 ("Distinct cortices show differential tuning to posture, movement, and self-motion"). The authors fit individual cells with GLMs through a model selection procedure to identify which behavioral features contributed the most to the model performance. This figure represents how the distribution of those features differs across different brain areas.

Before discussing further, Figure 2 is missing any information on how well the GLMs explained spiking activity, which appears in Extended Data Fig. 7. The measure of model performance is the pseudo- R^2 . Pseudo- R^2 would be 0 if the model explained no variance, and 1 if it exactly predicted the spiking of cell. Ext. Data Fig. 7 shows that a small minority of cells have a pseudo- R^2 greater than 0.1. The medians (reported in the main text) across different areas range from 0.01 to 0.03. Thus, to interpret these values, the authors need to explain what a "good" value of pseudo- R^2 is for single units during freely moving behavior. This is challenging, as there is no repeated trial structure to use to assess variance across repeated conditions to provide a ceiling to the possible model variance explained. One possibility is to follow a similar approach to the Stringer (Harris, Carandini) 2019 paper, in which they compared feature-derived predictions of neural activity to predictions generated by simultaneously recorded cells. In other words, if the GLM instead used neighboring cell activity as a predictive feature, how well would it perform? This could at least provide a scale by which to assess model performance. Further, I want to know, if you restrict analysis to cells for which the GLM explained a meaningful amount of variance, how does the feature analysis in Figure 2 change?

Turning now to the presentation as-is in Figure 2, there are four groupings of plots, each consisting of a pie chart ('proportion of $n = \#\#$ cells'), a bar plot, and a polar chart ('contribution to model performance'). The proportion of cells refers to the proportion of cells for which each of these behavioral features was the top covariate; between ~20% (Motor) and ~50% (Somatosensory) of cells are "unclassified," and presumably not included in the wedge plot at right or the bar plots below. The bar plots show how many covariates were included in the selected model for each cell. This is where one possible difference across areas becomes apparent, where some 40% of somatosensory cells only have one covariate and < 2% have 6 or more covariates; in motor areas especially, cells tend to have more covariates. This is potentially interesting, except that the cells sampled in the somatosensory areas are reported to be primarily Layer 2/3, while the other areas are all sampling deeper layers, either instead of or in addition to L2/3 (manuscript, lines 82 - 85). Related, extended data Figure S8 shows V1 vs V2 and A1 vs A2D differences in the number of covariates, but also tractography showing the probe crossing into different layers. If the authors have enough cells sampled from L5/6 in multiple areas, could they repeat this analysis restricted to that subset? Similarly, with L2/3? Do these trends persist?

Finally, the polar chart in each subpanel is represents 'contribution to model performance' of behavioral features. I am somewhat confused on how it was constructed. The angular extent of each wedge is 'the fraction to which each covariate was represented among all those selected.' Does this mean that you take every model for every single unit in the selected brain area, enumerate all the behavioral features that were ever used, and then report the fraction of that pool by feature? Is the contribution weighted by how much additional predictive power each feature contributed? Which cells are included in this analysis -- for instance, a substantial fraction of cells have "unclassified" top features, are these excluded? and are cells with pseudo-R² of 0.01 included on the same footing as cells with pseudo-R² of 0.1 or 0.2?

As a somewhat minor point, I would argue that it is perceptually confusing to set the 'length' of each wedge (rather than the area) to denote the relative log-likelihood ratio. Is 0.6 really four times higher than 0.3 for rLLRs? Again, this is averaged across all the models the covariate was included in, but how does it change if restricted to models with high pseudo-R²?

It may be worth considering if polar charts are the best way to convey this information in this figure. As is, given the concerns described above, I do not know how to interpret the data shown.

2. Regarding evidence supporting claim (d)

In the abstract, the authors state, "tuning properties of synaptically coupled cells also exhibited connection patterns suggestive of different uses of behavioral information within and between regions." This is primarily addressed in Figure 4.

In Figure 4, the authors use cross-correlograms between pairs of units to identify putative synaptic connections. Figure 4a shows an anatomical projection of auditory (green) and visual (pink) units, with grayscale lines showing putative connections. (The color bar (labeled $\log_{10} p$) uses the same darkness to represent a p-value of 1 and of 10^{-8} and should be reconsidered; at any rate, I cannot see where this colorscale is used in the figure.) The connectivity drawn in anatomical space is difficult to interpret; are cells truly unlikely to connect to their nearest neighbors, but likely to be connected across long A-P distance (roughly, left to right on this plot)? Figure 4b shows a scatter of synaptic strength against distance, with no significant trends.

Figure 4c is where the main point is made: that tuning-specific connectivity suggests different uses of behavioral information within and between regions. First, with respect to "between regions," I do not see in this figure any analysis of synapses between regions, so perhaps something else is meant. The question is then, are there different tuning-specific patterns of connection within regions? The evidence for this is thin. Fig 4c represents the number of synapses found between functionally identified pairs of cells, where "function" is Mo (movement-responsive), Po (posture-responsive), LM (light-modulated) and SM (sound-modulated). (The authors claim that these subsets are largely non-overlapping, but this is not shown.)

In Fig. 4c, a shuffle test within the synapses found in each area was performed. Note that there were different numbers of cells in visual vs auditory areas, and about 2x as many synapses found in visual areas. The overall number of detected synapses is also small: from 5 to 15 in any auditory cortex group, and from about 10 to 40 in visual areas. If this is out of the ~10k possible synapses in the 100s of simultaneously recorded cells, this is a very small fraction, and I do not know how much weight I should give these results.

The shuffle test represents the null hypothesis that synapses occur with equal rate independent of functional cell class. In visual areas, there appear to be more Mo to Mo synapses (both excitatory and inhibitory), more excitatory Po to Mo synapses, and more inhibitory Po to Mo synapses. Comparing that result to the auditory result is challenging. Due to the smaller number of cells recorded in auditory areas, it is harder to detect functional patterns of synaptic connectivity over the shuffle baseline. Slightly more inhibitory Mo to Po synapses and Mo to SM synapses were found. The cells and significant connections, for both areas are shown in a functional space, which cannot be understood without referring to the details in Ext. Data Figure 13.

I would suggest that the authors consider whether Figure 4 truly belongs in this paper, and if it does, whether Ext. Data Fig. 13 should be included as well. How does it support the main point about 'encoding structure employed across neocortex'?

Minor concerns:

A general minor concern: burying the lede

There are several interesting statements that are almost throw-aways in the paper as written. For instance, recordings were performed in dark and light conditions, with the decoding analysis restricted to the dark condition [lines 53-54] In Figure 3, the authors show a dramatic shift (in functional UMAP space, at least) of neural activity in visual areas in lights on vs lights off conditions. The analyses in Figure 2 also utilized the dark condition (lines 70-71), reporting that visual areas maintain information about head position in world coordinates. If this is really in the absence of visual inputs, how is a world-coordinate representation maintained? Potentially through some sort of angular path integration or triangulation with auditory inputs (which, like visual inputs, have sources that could provide stable world-reference coordinates). At the least, are the GLM features and GLM performances in visual areas different when trained on dark vs. lights-on recordings?

Another result that seemed deserving of more attention was the perturbation experiment in which a 15 g weight was added to the head mount, with results reported in Extended Data Figure 10 and summarized in a few lines (106-113) in the results.

Lines 44 -46: "The animals combined ethogram consisted of 44 independent modular actions ... whose sequence order was best described by a set of transition probabilities"

- Is it shown that transition probabilities are the best description? My understanding of these ethograms is that the behavioral state dynamics tend to be highly non-Markovian, suggesting that transition probabilities alone are insufficient. See, e.g., Alba, Berman, Bialek and Shaevitz, arxiv, 2020.

Figure 1:

The top panel is relatively clear, as is the behavior map and example postures in the bottom panel. I do not know how to interpret actions 1 to 44 without a map, and I don't understand what new information the map provides that bar heights do not. Could you pick one or the other?

Further, it is a non-intuitive choice to designate dark bars for accuracy of decoding with the prior and light bars for accuracy of decoding using neural activity, while in the maps of decoding, it appears to be light for low accuracy and dark for high accuracy.

A confusion matrix would be good to see in the extended data to understand how distinguishable similar behaviors are.

Minor grammar: "of each covariate across the set of models it was included in" -> "of each covariate across the set of models in which it was included"

Figure 3:

I do not understand why the average lines in the A1 plots in (B) are blue and yellow; do these correspond to blue and yellow clusters in the UMAP plots? I don't think so, as those seem more likely to correspond to the blue and yellow in panel (C), where light-modulation is demonstrated. Related, why is there no corresponding UMAP plot for sound modulation?

The gradients in (D) are intriguing, but also seem to move in and out of different layers in different areas (as portrayed in Extended Data Fig 1D and f). This makes it difficult to determine whether it is a proximal-distal gradient, or a laminar gradient.

Figure 3(E) is nice: it appears there is a distributed representation of sensory inputs and behavior across auditory and visual areas, and that this is more or less similar between FS and RS units. It would be helpful to have unit counts at each branch.

Figure 4:

In Fig. 4C, use the same x limits on all subplots. It would also be useful to know the number of possible connections.

In Fig. 4 D, do not use the same color for FS/RS outline that you use in Fig 4B to indicate auditory regions. A1 and A2D are yellow and brown - why not shades of green to link to panel B and C?

The caption says "We note that the outlined pair groups are largely separated." for Fig 4C. This separation is not clear to me in the UMAP plot at right. More details on the functional subspace would

be helpful for interpreting this plot. Extended Data Fig. 13 is useful in this regard, but Figure 4 needs to stand on its own.

Extended Data Fig. 6

All x-axes should be the same

The x-label says cross-validated log-likelihood, but the caption says cross-validated rLLRs.

Reviewer #3 (Remarks to the Author):

Summary:

Mimica*, Tombaz* et al present an impressive and unique study that examines the functional coding properties of neurons in somatosensory, motor, visual, and auditory cortices during natural behavior. By first parsing the natural behavior into actions (e.g. rearing), the authors find that most actions can be decoded in each region. By then parsing the behavior into continuous movements, such as head orientation in egocentric and allocentric coordinates, the authors uncover coding properties specific to each region. E.g. visual cortex encodes allocentric head movement, while motor cortex encodes egocentric movement. The authors then show that neurons in visual and auditory cortices encode visual and auditory information alongside behavioral information, and that there is structure to the putative synaptic connections in visual and auditory regions.

Overall, the data collected are quite impressive, as are the wide range of presented analyses. Understanding neural activity during natural behavior is of high interest to many but an incredibly difficult task. Here the authors have very bravely waded into those waters and come out with some interesting results. However, I felt there were some issues with the manuscript in its current form. First, while in general I can get behind the sharing of results even if their full impact or interpretation isn't entirely clear yet (as every paper is a small part of a bigger story), I felt that some of the results presented here really didn't have quite enough contextualization, interpretation, or motivation. There are a lot of facts in this paper, and in some places it was hard to know what to take away, or what exactly was learned that wasn't clear before. More details on this point are listed below, but overall, I felt this applied most to results in Figures 3 and 4. This issue is also exacerbated by the current short format of the article (which I'm not sure is necessary, although I admit I don't know the character limits here). Further, some of the figures were hard to parse, and some analyses or methodological details were either missing or hard to find. These issues and others are detailed below.

Major comments:

1. As mentioned above, I felt that in a couple places the figures were difficult to parse. While some figure panels are mentioned in other comments, issues pertaining to the rest have been collected into the list below.

a. Figure 1- Artistically speaking, the circle looks nice. But it is very hard to visualize the decoding accuracy or compare the size of the bars in different regions in this format. While less exciting, it might be easier on the reader (and better convey the points you make in the text) to show rows of bar plots. In addition, the comparison to the control decoder, in which decoding is based on the prior, is vital, but difficult to see here.

b. Figure 3

i. I find the lines in Figure 3b,c to be visually confusing - I see it is a double axis, but it makes it quite hard to interpret.

ii. The proportion breakdown in the bottom of Fig 3b,c is really difficult to parse.

c. Extended Data Figures –

i. Extended Data Fig 4a,b are difficult to interpret. Can they be summarized or restructured to highlight the most meaningful relationships between behaviors?

ii. Extended Data 10 - this figure has a ton of information in it, but with the number of panels and subpanels, it's difficult to pull out the relevant facts. In some cases, I was not sure what was being plotted - e.g., what the 'information rate' in panel e is being computed from (the encoding model?). I would suggest substantially simplifying this figure.

2. How were spikes sorted across multiple 'schedules' (the group of 4 sessions)? It's my understanding that the Neuropixels shank was implanted and in the same place across these schedules, but were the spikes sorted together? Or were cells sorted separately in each schedule? Unless I have missed something, it seems this latter method would lead to double-counting of the same cells.

3. In Line 51, the authors claim that each action is encoded by some population of neurons in each region. However, in Extended Data Figure 3, it looks like there are quite a few actions that are not encoded by neurons in any region. (To me, this figure actually highlights that only a subset of actions seem to be encoded for in each region.) Perhaps this can be directly quantified somewhere, either in the main text or the figure legend, to make it clearer that there are neurons encoding these actions.

4. Related to the decoding result in Figure 1 - the number of neurons will affect decoding accuracy (e.g. as in Figure 3b-c). However, I could not find the average number of simultaneously recorded cells (or how much this number varied across sessions), or whether this was controlled for when presenting decoding accuracy across regions. Also, given that motor regions encode egocentric movement the strongest, did the authors also observe higher decoding accuracy for actions here as well?

5. Regarding the model fits - 20 minute sessions are quite short for fitting spike trains to natural behavior data, and the resulting pseudo-R2 values are quite low with medians ~ 0.02 . While this is to be expected to some degree for natural behavior no matter how long the recordings are, the authors may be able to provide additional support for the encoding properties they present if they are indeed tracking cells across schedules (or even across light1 and light2 sessions). In this case, are the encoding properties (variables encoded and their tuning) generally similar? If not, the data presented in Extended Data Figure 5 is relevant and convincing; however, there are only 4 example cells shown. How often are the tuning curves similar across session halves?

6. Regarding the result regarding the addition of a weight - in the main text, it seems that the addition of the weight did not change the encoding properties of motor cortex cells. However, in extended data fig 10, it is difficult to see the extent to which this is the case. In particular, while the proportion of cells encoding different features is similar across the population in panel G, it is unclear whether other aspects of the encoding, such as the weights or log-likelihoods, are similar. Perhaps a more straightforward comparison would be to train on a light1 session, test the model on the weighted and light2 session, and then compare the resulting model fits. Further, one could directly compare the glm tuning for models fit with the same variables in each session. For what it is worth, I think this potentially a very interesting result, since this relates to ongoing arguments of encoding kinematics versus dynamics in motor cortex (which could also be referenced more directly in the text).

7. Regarding the results in Figure 3 - currently, much more attention is paid to the overlap in sensory and behavioral variables in visual and auditory cortices than motor and somatosensory cortex. While I somewhat understand the logic for looking at sensory variables in sensory cortex, this has the unfortunate effect that the main results seem to be that visual cortex responds to visual stimuli, and auditory cortex responds to auditory stimuli. It's unclear what we've learned from this result, given that this would be expected. However, given that authors recorded from somatosensory and motor cortex during these trials as well, it might be interesting to include all four regions in these analyses and discuss the differences and similarities. For example, does the behavioral encoding in motor or somatosensory neurons depend on light/dark conditions? Are neurons in these regions modulated by luminance/sound and/or behavior?

8. Extended Data Figure 12b does not include p-values for the changes in encoding properties of neurons in the light and dark conditions. Do you think that the results reflect an actual change in the

encoding of movement features, or that the actual features that the neurons are responding to are visual in nature and are only correlated with movement in the well-lit condition?

9. There is a thread regarding FS/RS cells that I am not following. I think that part of the confusion is that FS/RS breakdowns in Figure 3 are not mentioned in the text, and then FS/RS properties relating to Figure 4 are mentioned in the discussion, but not mentioned in Figure 4 or the related main text in the results (unless I missed something). Further, it's not clear to me what the main results are regarding FS/RS cells. For example, in Figure 3, both types of cells seem to have very similar properties. What is the result that the authors are trying to highlight here?

10. In Figure 4, the take home message to me is not clear. I think perhaps part of the issue is that it is unclear what should be expected here, or how the results presented specifically update our knowledge of functional connectivity in visual and auditory cortex. I think additional text is needed to motivate and contextualize these results. There has been prior work on functional connectivity in at least visual cortex (Ko*, Hofer* et al 2011), which can be discussed here. Further, it is unclear what the functional space embeddings are supposed to convey.

a. In addition, it is not immediately clear to me why the focus is on auditory and visual cortex. The results in Extended Data 13 F-G could be compared to those in Figure 4 C-D. For example, does the increased precedence of Po->Mo excitatory connections in visual and somatosensory cortices as opposed to auditory and motor cortices align with a model similar to those described in the caption of Figure 4?

Minor comments:

1. There were a few tiny typos:

a. Line 442 – should be 'represent'

b. Line 450 – 'features' is misspelled

c. Line 454 -- there is an extra parentheses

d. Line 65 – should be 'nearly all'

2. Line 51 - it is not completely clear what the numbers in the parentheses refer to. Is this the number of neurons that encode at least behavior in each region? I think the wording should be changed, or text added, so this is more transparent.

3. To help compare the number of simultaneously encoded variables across regions, the axis of the bar plot in Figure 2d should be the same as the other panels.

4. Lines 94-98, Lines 118-122 - the authors report a gradient in encoded features, but it is unclear to me whether the observed gradient is surprising or fits with the current literature. I think a sentence or two contextualizing these results would help.

5. Either the legend or the panel in Fig 3a is not correct – the legend says that the FS spikes are in green, but I do not see any green spikes.

6. Extended Data Fig 3 - it seems that panel c has actions in a different configuration than the rest of the panels? This makes it difficult to visually compare with the others.

7. Extended Data Fig 6 - the x-axis should be labeled as 'relative log-likelihood' – just listing 'cross validated log-likelihood' had me confused for a little bit.

8. Extended Data Fig 7 - it would help if all were plotted on the same axis.

9. Extended Data Figure 13 – I believe the colors describing panels A and B in the caption are incorrect.

10. Lines 418-422 - it is unclear how the two halves of the split-half analyses were obtained.

Response letter, Mimica et al. 2022.

We thank the reviewers for their constructive input and suggestions for how to improve the prior version of our manuscript, 'Behavioral decomposition reveals rich encoding structure employed across neocortex'. Accordingly, the manuscript has undergone substantial modifications, with the inclusion of new sets of expanded and supporting analyses, revision and inclusion of new main and supplementary figures, and the addition of text (changes written in **bold**) throughout the manuscript to provide a clearer narrative and context for the results. New analyses include examination of tuning properties across cortical layers (including superficial and deep layers in visual and motor cortices, and granular and deep layers in auditory cortex) in relation to naturalistic actions (Supplementary Fig. 7) and low-level pose and movement variables (Supplementary Fig. 13). This extended our original results by showing that regionally distinct tuning to lower-level features was conserved across layers, but was more abundant in deeper layers in visual and auditory cortices. We also included examples of stable tuning curves for posture and movement features in superficial and granular (Figure 2) and deep (Supplementary Fig. 10) layers. In addition, we analyzed fast- and regular-spiking cell types in relation to higher- and lower-level behavioral features (Supplementary Figs. 7 and 14), which again showed largely similar tuning preferences, but with fast-spiking cells preferentially encoding movement features in somatosensory and visual cortices. The revision also includes confusion matrices to convey the specificity of the decoding results (Supplementary Fig. 8), and additional GLM analyses comparing tuning properties in the head-weight condition (Supplementary Fig. 17) and across light and dark recordings (Supplementary Fig. 20). To make to make key supporting data more easily available, we moved figures showing tuning curves and the topographical analysis to the main text (Figs. 2 and 4, respectively). The revised manuscript, including new analyses, consists of 6 main figures and 21 supplementary figures.

The rebuttal also provides several critical tests of our analytical methods requested by the Reviewers, which validated the results of our GLM framework as well as our methodology for extracting individual actions. We have provided point-by-point responses to each of the Reviewers' concerns below.

Reviewer #1 (Remarks to the Author):

The authors chronically recorded simultaneously from either motor and somatosensory (left cortex; N=4) or auditory and visual cortical areas (right cortex; N=3) with neuropixels probes in freely foraging rats while monitoring their movements with an array of imaged 3D markers and accelerometers. Recording locations were confirmed by decoding analysis of passive stimulation epochs, and a post-mortem combined MRI and immunohistochemical / histological approach. All data were resampled and aligned at 120 Hz, then detrended and decomposed by Morlet wavelet. T-distributed stochastic neighbor embedding (t-SNE) was then used to identify 44 recurring action classes / behavioral motifs, and generalized linear models (GLM) used to determine which components of neural and inferred synaptic activity best predicted the action classes at each time point (binned at 8.33 ms or 1/120Hz). Just over half of neurons in all recorded areas predicted each action above chance. However, components of movement, posture, and self-motion were differentially represented across the four main cortical areas recorded from. Visual cortex represented horizontal scanning head movements and head position, auditory cortex represented gravity relative head position, motor cortex represented overall body and back position, and somatosensory cortex represented back position posture but also

had a large proportion of “unclassified” neurons. Putative synaptic connections were identified by statistically enriched cross-correlogram peaks or troughs (i.e. excitatory or inhibitory). Synaptic connections within visual cortex tended to connect neurons with homogenous coding of behavioral subcategories (i.e. movement, posture, self-motion), although there was also an enriched probability of excitatory connections from posture encoding neurons onto movement encoding neurons, and of inhibitory connections from movement neurons onto posture neurons (feedback inhibition motif?). Movement neuron inhibition of posture neurons was the dominant motif in auditory cortex, and pairs of neurons in general that received common input (i.e. CCG peak at ~0 ms) tended to encode the same class of behavioral components.

Overall this study seems to be taking the right approach to better understand the cortical neural encoding of spontaneous, ethological behaviors. However, the main findings come across as somewhat preliminary and disjoint, lacking an overall cohesive message.

We appreciate the Reviewer’s supportive view of our approach in the study, and have modified and added several portions of text throughout the manuscript to impart a more cohesive message than before, including the endings of the Abstract and Introduction, at the beginning of each of the Results section, and in the Discussion. The cohesive message we hope reads more clearly now is that, by quantifying behavior carefully and at multiple levels, we can generate sensible insights into how such signals are employed by different cortical systems during natural behavior, particularly in sensory areas.

Furthermore, confidence in the main findings in terms of differential action encoding across cortical areas would be enhanced if recordings were simultaneous across all four main areas, and if comparable data existed for these areas in terms of the cortical layer/depth from pia distribution of recorded neurons. Inclusion of data regarding fine facial movements, which are known to be directly indicated arousal state and strongly predict cortical neural activity, would likely strengthen the validity of behavioral action classification. Lastly, use of longer recording sessions would likely enhance the validity of behavioral action transition matrices and allow for observation of the full span of the hierarchy of arousal states / behavioral actions.

We thank the Reviewer for suggesting ways in which the action classification and transition analyses could be strengthened. This prompted us to include several new analyses and figures (listed below) which we feel have strengthened the study, though we are not able to oblige all suggestions. We provide responses below for each point raised.

Regarding simultaneous recordings across cortical areas: we wish to clarify that we do not claim to show differential coding of actions across cortical areas. On the contrary, we find action encoding is similar across regions, which is supported with an equivalence test (Supplementary Fig. 6b). We contend that simultaneous recordings of all 4 regions would not make the claim of equivalence stronger because one could attribute regional similarities to behavioral confounds with recordings coming from the same animal.

Regarding recordings across cortical layers and cell types (RS vs. FS): this is an excellent suggestion, and we have included breakdowns of the data accordingly in a new Supplementary Fig. 7, related to action encoding, and in new Supplementary Figs. 13 and 14, related to GLM results. Additional details on the results are included in the point-by-point responses below.

Both facial tracking and longer recordings were suggested as ways of validating our action classification and transition matrices using 20-minute recordings. We investigated this issue more closely on the assumption that, if our methods and recording durations were insufficient to extract meaningful actions, then the classification results should be arbitrary from one recording session to the next. However, we found that both action classification and the associated transitions were highly consistent across sessions (discussed below for point M1), which supports that our behavioral analysis approach is valid.

We otherwise agree that facial tracking would give added insight into internal states, and that longer recordings would add insight into the hierarchical organization of behavior. These are related and fundamental questions facing the field currently, but addressing them would require major technical and analytical developments which we feel fall beyond the scope of the present work. One is that the field has not yet solved how long behavioral recordings in rodents need to be before higher levels of the behavioral hierarchy emerge. The one such study of which we are aware that quantified rat behavior over long time scales (Marshall et al., *Neuron*, 2021) recorded from animals in a home environment on a 24/7 schedule for >1 week. We feel that repeating our recordings following such a schedule, and in a necessarily new chronic recording setup, exceeds the scope of the present study. Likewise, inclusion of facial tracking would require starting over with new sets of recordings with facial cameras. This would necessitate the new design and incorporation of implants that house facial cameras alongside the electrophysiology hardware, as well as synchronization of the data streams, and expansion of our rate-coding, GLM and action classification pipelines. We respectfully argue that this would constitute grounds for a new stand-alone study.

We contend that our study, by focusing on time scales of seconds-to-minutes, still makes critical steps toward understanding how primary sensory and motor cortices represent unrestrained behavior at multiple levels of complexity. Such work is needed as a precursor before investigating more complex levels of hierarchical organization, and prior to considering additional variables such as facial features. We now make the distinction in the Results (lines 46-49) and at the close of the Discussion (lines 328-333) that our study focuses on action coding at shorter, rather than longer, time scales.

Major Points:

M1.-The identified behavioral actions are too granular and brief, perhaps missing organization from one level higher in the presumed hierarchy of behavioral organization. This can be seen as the lack of apparent sensible organization of the behavioral action transition diagram (Extended Figure 3c).

We regret that the structure of the behavioral transitions was obscured by the way it was presented originally, giving the impression that the data lacked a sensible organization with too-brief behaviors. To address this, we (i) arranged the action transition matrices to match the tSNE embedding in Figure 1, so the actions and their transitions appear less arbitrary (shown in a new Supplementary Figure 4; see below). We (ii) checked the stability of the transition matrices in light vs. dark sessions and found they were very highly conserved across the recordings (Pearson's R of 0.968, $p = 0.00058$), which speaks very much against them being arbitrary. The average duration of actions across conditions was also similar across conditions (0.47 ± 0.33 s (\pm SD) in light sessions vs. 0.49 ± 0.36 s in dark sessions). We note that these

durations are in line with sub-second durations of behaviors published previously (durations on the order of ~0.5s were reported by Wiltschko et al. 2015 and Hsu & Yttri 2021, and >1s by Huang et al. 2021). We include the stability result in the Results section, lines 53-55, and action durations are included in the legend of the new Supplementary Figure 4.

Supplementary Figure 4

M2.-tSNE organization of actions seems to have inherent organization unmentioned by authors – three vertically oriented strips (left, middle, right) seem to encode arousal level / locomotion speed, rightward orienting postures and movements, and leftward postures and movements, respectively.

We more closely inspected the organization of behavioral features in tSNE space and indeed found that aspects of rudimentary pose and movement followed prevailing directions across the t-SNE embedding. Though not precise, there was an overall increase in running speed from the top to the bottom of the embedding, as well as changes along a left-right diagonal for the pitch of the head and back. We point out these patterns now in the legend of Supplementary Figure 4, where the map of actions in the tSNE embedding is introduced:

“The actions embedded in t-SNE space followed a coarse inherent organization of rudimentary features. These included an increase in running speed progressing from the top to the bottom of the tSNE map, as well as a tendency for the back to be low or hunched at the upper-left of the map, and raised vertically at the lower right of the map (i.e. at the rearing actions, 42 and 44). Head pitch followed a similar coarse diagonal

(head lowered in the upper left portions in t-SNE space and raised at lower and rightward regions).”

M3.-Explanation for why auditory cortical areas would primarily encode gravity relative head movements is not convincing.

We have expanded our interpretation of the potential use of gravity relative postural signals (i.e. for head roll and pitch) in auditory cortex. We explain that rodents rely on interaural loudness differences (ILDs) for localizing sound sources in the environment, and that an animals' head posture will strongly influence how these inputs are sampled. Strong head roll and pitch modulation (i.e. in the vertical plane) will facilitate detection of differences in loudness in vertical space, allowing for 3D sampling of sound in the environment. Conversely, if the head and ears are perfectly level, then any ILD corresponds to a change in horizontal localization, and the individual is “blind” to changes in vertical localisation. Once the head rolls, that changes, therefore knowing the pitch and roll of the head is probably critical to interpreting a localization cue. The revised text in the Discussion (lines 313-319) reads:

“The roll and pitch of the head might be heavily represented because these features would strongly affect how interaural loudness differences (ILDs), used by rodents to locate sound sources, are sampled in the environment. If an animal’s head is perfectly level, any ILD corresponds to a change in horizontal location, and the individual is “blind” to changes in vertical localization. Once the head rolls, this changes, and strong roll- and pitch-modulation would facilitate detection of ILDs in vertical space, allowing for 3D sampling.(Lauer et al., 2018). We note, however, that the precise influence of posture on sound localization in rodents remains largely untested.”

M4.-The main finding that visual areas strongly encode head movements and head posture seems to not be novel, and may merely represent subtraction of changes in the visual scene due to self-movement, rather than representation of ethological behaviors in these areas, per se.

We appreciate the concern regarding confounds due to visual scene changes during movement; however, we used recordings in darkness specifically to avoid the issue of visual scene changes. Since this is a critical methodological aspect, we note it again in the Discussion (below), in addition to the Results. Regarding the novelty of the visual cortex results, we added text to better contextualize our findings relative to prior work. In our view the novelty is that we measured several behavioral features at the same time and evaluate their relative importance, in addition to considering different frames of reference which the earlier studies did not do. The revised text from the Discussion (lines 275-280) is below:

“The activity of visual ensembles was best described by allocentric horizontal motion of the head and movement of the body over the surface of the arena, which confirms and extends earlier observations that features like angular head velocity (Vélez-Fort et al., 2018; Bouvier et al., 2020; Guitchounts et al., 2020), head direction (Guitchounts et al., 2020), and running (Keller et al., 2012) each modulate spiking activity in V1. The combination of these signals likely supports a variety of computations for stabilizing and predicting motion of the visual field, as well as optimizing visual processing during active movement (Ayaz et al., 2013; Dadarlat & Stryker, 2017).”

We also agree that movement representation in visual cortex would signal when visual scene changes are due to self-movement in lit conditions, which we note in the Discussion on functionally connected neurons (lines 290-295):

“More importantly, such fast spiking movement-responsive cells also inhibited posture-encoding units (e.g., a left turn cell inhibiting a cell encoding rightward posture), and posture-modulated neurons were shown to excite luminance-modulated and movement-responsive cells (e.g., a rightward pose cell exciting a leftward-rotation neuron). This might assist visual networks in distinguishing whether optic flow is likely generated by self-movement (McNaughton et al., 1994): if postural cells convey static head position, their direct inhibition would indicate that visual scene changes are likely self-generated.”

M5.-Examination of recurring synaptic motifs across cortical areas is extremely interesting, but the n used here are perhaps insufficient to make strong statistical conclusions concerning their relative prevalence. In addition, showing that manipulation or modulation of these synapses during particular behavioral actions and/or transitions would greatly contribute to the cohesiveness of the main findings of the study.

The total number of putative synaptic connections reported in the study is 650: 247 for visual, 107 for auditory, 35 for somatosensory and 261 for motor cortices. We agree in proceeding with caution regarding sampling that is limited, but we note that the inclusion criteria for counting cells as synaptically connected were strict (cross correlation p-values < 0.0001), as were criteria for significance in the rate of putative synaptic connections (exceeding 99% confidence intervals beyond shuffled data). We have acknowledged the limited sample size by modifying wording in the revised manuscript (lines 201-205):

“We identified a limited, but informative number of putative excitatory synaptic connections (driven by RS units) and inhibitory connections (driven by FS units) of varying strength within auditory (n = 107 total connections) and visual cortices (n = 247) (Fig. 6a, b; Methods) as well as somatosensory (n = 35) and motor cortices (n = 181) (Supplementary Data Fig. 21a, b).”

We nevertheless believe it is important to report these results for the following reasons: (1) The field of single-unit electrophysiology as a whole still very much relies on drawing statistical inferences on limited samples, e.g., in this study we make inferences based on samples of ~1000 neurons *per* region, though anatomically distinct areas are composed of different layers with tens of millions of neurons each. Given this, we believe it is still relevant to disclose our findings fully, given the particular rigor applied in constructing statistical arguments. (2) We would also argue that recordings as expansive as the ones we are reporting, in freely-moving and freely-sensing animals, can still be considered a valuable rarity. They offer new insights which point to fundamental network properties of cortical areas which lack recurrent connections characteristic of other networks (e.g. in the head direction or hippocampal systems), which significantly lowers the probability of detecting putative synaptic relationships, but are still valuable as a reference point for future studies. (3) Although the absolute number of reported putative connections is limited, we would argue that the analyses employed on such data still enable us to uncover meaningful effects. The shuffling method used in Figure 6c, (formerly Fig. 4c) compared observed data to that if the single units we recorded were randomly connected. Because each shuffle iteration constructs the same number of random pairs as those that have been observed, having more or fewer pairs does not make the shuffle test easier or harder to pass. Notably, previously reported findings of movement-inhibited sound-responsive units in A1 (Schneider et al., 2014) were replicated in our experiments, arguably lending credence to other synaptic motifs we are reporting.

As for manipulations—now that we have generated observations regarding the nature of the functional connections (e.g. posture → luminance modulated cells), it would be possible to generate testable hypotheses about the function of the connections. Testing this, however, poses a major technological challenge, requiring selective perturbations of neurons whose functional characteristics are determined *in vivo*. The only approach that could work for this, to our knowledge, would be all-optical, using holographic silencing (or stimulation) of cells that were functionally identified using 2-photon microscopy in freely moving subjects. These combined approaches are still just on the horizon for experiments in unrestrained animals. We feel that developing and applying such novel methods would be a major technological advance, but would constitute a new stand-alone study outside the scope of the present work. We include this as a direction for future work in the final paragraph of the Discussion (lines 324-328):

“Pinpointing the mechanisms by which behaviorally generated signals shape sensory circuit computations will be challenging in freely moving animals, requiring sufficiently resolved techniques, such as miniature 2-photon imaging (Zong et al., 2021; Zong et al., 2022) and holographic stimulation (Marshall et al., 2019), to identify then manipulate behaviorally-tuned neurons in vivo.”

M6.-Extracted behavioral primitives do a relatively poor job of explaining neural activity in somatosensory cortex (Fig 2D), and there is no good explanation for this. Is this due to a different layer recording position in somatosensory cortex, or to the lack of inclusion of fine whisker and facial movements in the overall model, for example?

This is a shrewd observation, and we now give more attention to the lower rates of observed tuning in S1 in the revised manuscript. To check if the underlying reason was due to recording in superficial layers, we compared tuning in layer 2/3 across somatosensory and motor cortices but still found a comparatively lower proportion of assigned covariates in S1:

We did, however, uncover differences when we split S1 recordings into FS and RS neurons, with larger fractions of FS neurons encoding movement of the back and self-motion:

Somatosensory cortex, layer 2/3

We note in Discussion that the propensity of FS neurons to represent movement features in S1 is quite consistent with earlier work from the Svoboda lab, reporting movement-driven increases in the spiking of parvalbumin neurons during active whisking in S1 barrel cortex (Yu et al., *Neuron*, 2019). Though we cannot prove the same circuit is at play in our recordings, our results are consistent, and we point this out in the Discussion (quoted below).

As for the overall larger number of unclassified neurons in S1, since it is unlikely attributable to the layers recorded, we suspect the unclassified cells respond to features which we did not track, such as the whiskers, face, limbs or tail. Given that the electrodes traversed the trunk and hindlimb regions of S1, we suggest that the limbs in particular could contribute, and have revised the Discussion (lines 260-270) accordingly:

“Somatosensory neurons, on the other hand, were better characterized by sparser models related to the trunk, and tuning was more common among FS than RS cells. Fast spiking neurons in particular encoded dynamic features such as trunk movement and self-motion, which fits broadly with known circuitry for movement-driven disinhibition of parvalbumin interneurons in S1, described in barrel cortex during active whisking (Yu et al., 2019). Overall, however, model performance in S1 was the lowest of all regions, which could be traced to different factors. It was not likely due to the recordings being confined to layer 2/3, since only 22% of layer 2/3 neurons in adjacent motor regions were unclassified. In this case, unclassified neurons could have been driven by features we did not track, such as the whiskers, face, limbs or tail. The hindlimbs may have been more influential in our recordings since the probes were placed in the hindlimb and trunk regions of S1, medial to barrel cortex, and the fact that trunk kinematics in quadrupeds are steadily affected by gross dynamics of the hindlimbs and hips (Halley et al., 2020).”

M7. Could the differences between areas result from different neurons being recorded? What if the same neurons (e.g. regular spiking cells in layer 5) were compared between areas – does that affect the results? What are the differences between layers within the same area? Are these differences similar to differences between areas?

These are excellent questions, and in response we have included multiple new analyses with the recordings separated by RS and FS cell types (as noted above) and by cortical layers, where recorded.

As for action coding, this revealed that the distribution of encoded actions and the proportions of tuned cells were similar for RS and FS neurons in all regions except visual cortex, where actions were encoded by a slightly higher fraction of FS neurons (Supplementary Figure 7a; shown below). In visual and motor areas, the distribution of encoded actions appeared similar across deep and superficial layers. The same was true between layer 4 and deep layers in auditory cortex (Supplementary Figure 7b). Similar percentages of cells encoded actions in superficial and deep layers of motor cortex, but there were notably fewer coding cells in superficial than deep layers in visual cortex (39% tuned cells in Layer 2/3, 41% in Layer 4, and 53% in layers 5&6) and auditory cortex (40% in layer 4, 58% in layers 5 & 6):

Supplementary Figure 7

For the lower-level features, GLM analyses showed that regional variations persisted across deep and superficial layers in visual and auditory cortices, which still primarily encoded head posture, head movement and self-motion (new Supplementary Figure 13, shown below). Similar to the action-encoding results, tuning was more widespread in deep layers of visual and auditory cortices which, at least for visual cortex, is consistent with earlier work showing that deep layer V1 neurons respond to head rotations in darkness (Velez-Fort et al. 2018; Bouvier et al. 2020). Both those and our findings point to the deeper cortical layers as a possible site of integration for motor input from midline motor cortex (Reep et al. 1987) and vestibular signals relayed by retrosplenial cortex (Velez-Fort et al. 2018), noted in the Discussion (lines 280-284).

In motor cortex, the laminar analysis uncovered differences across deep and superficial layers, with the trunk being encoded more strongly in superficial layers and the head represented more in deep layers (see Supp. Fig. 13, below). However, due to the tilt of the Neuropixels, layer 2/3 was sampled more posteriorly, and layer 5 was more anterior—which follows an established topography of head and trunk representation over the dorsal cortical surface in rats (Head & Lindholm, 1974; Neafsy et al., 1986). Further, our own topographical analysis (now main Figure 4) showed a continuous evolution of back-to-head representation progressing from posterior to anterior recording sites. So, while we cannot rule out differences in cell layers, the result can be accounted for by known anatomical gradients. We note the earlier anatomical mapping studies and topographical organization in our data in the Discussion (lines 270-272).

Supplementary Figure 13

In light of the new across-layer analyses, we also included additional examples of stable tuning curves from single cells in main Figure 2 (superficial or granular layers) and Supplementary Figure 10 (deep layers).

In separating recordings from FS and RS neurons, we found a larger proportion of FS neurons encoded dynamic features of the trunk in S1 (as discussed above), and a slightly larger fraction of FS neurons encoded allocentric head movement and self-motion in visual cortex (Supplementary Fig. 14, below). The potential functions of such fast-spiking activity in visual cortex are considered in the context of functional connectivity in the 5th paragraph of the Discussion (lines 273-284). FS and RS neurons in auditory and motor cortices encoded similar proportions of features.

Supplementary Figure 14

M8. How much of the widespread nature of activity related to behavior simply the result of the animal experiencing widespread changes in multiple inputs during behavior? E.g. when a animal moves, even in the dark, there is large scale changes in motor, somatosensory, auditory, vestibular, eye movement, etc. signals.

We appreciate this as a fundamental question, but one that is difficult to address concretely. For decades animal subjects have been treated as a single point in space with a location, bearing and speed, as with studies of spatial coding in the hippocampal system. Our goal was to take a step beyond traditional approaches, to track more, and to start by using the simplest environment we could provide: an open arena, in darkness, without sound or other overt stimuli. We point out that we

succeeded in finding stable tuning for the subsets of behavioral features we *could* track, then systematically manipulated inputs where feasible (e.g. light/dark sessions, sound stimuli, head weight).

It is also important to note that the regional differences in low-level features could not be explained by gross changes in inputs, since these would presumably have been the same across areas. To further test whether tuning across cortical regions changed following gross changes in environmental input, we performed new GLM analyses on recordings from all regions in dark and light conditions, and assessed whether the proportions of selected features changed with a two sample Z-test for proportions. This showed that only visual cortex was sensitive to the light-dark manipulation (Supplementary Figure 20), and the lack of change in other regions is included in the Results (lines 188-195).

It remains a major question in the field as to how the many additional features which we could not (yet) track influence spiking activity, or how neural tuning in different systems would change in more complex environments or during different motivational states. We suggest these as directions for future work in the last paragraph of the Discussion (lines 328-330).

Minor Points:

m1. It's not clear what the partial foraging paradigm adds to the study, other than to ensure a minimum amount of locomotion during the recordings. Also, it seems that the probability of naturalistic behaviors not directly related to foraging may have been decreased by this aspect of the experimental design.

The foraging paradigm was used to promote movement and exploration of the open recording arena, though it indeed probably influenced the proportion of foraging-relevant behaviors, such as bouts with lower speed, with the head lowered to retrieve food off the floor. This aspect of the task design is now acknowledged in the last sentence of the 2nd paragraph in the Results:

"The behavioral dynamics observed were also likely influenced by the foraging task, and would differ from purely exploratory patterns behavior in novel environments (Benjamini et al., 1996)."

m2. -The vagueness of the final sentence of the abstract speaks to the overall suggestive nature of the results and the lack of a cohesive overall finding.

We have changed the first and final sentences of the abstract to emphasize the main findings that (i) ongoing behavior is represented at multiple levels potentially throughout the dorsal cortex, and (ii) that low-level pose and movement representations vary by region and may support locally-relevant computations, particularly in sensory regions. The last sentence of the Abstract now reads:

"Together, our results indicate that ongoing behavior is encoded at multiple levels throughout the dorsal cortex, and that low-level features are differentially utilized by different regions to serve locally relevant computations."

We underscore these themes again in revised text in the last paragraph of the Introduction, and in motivating statements provided at the beginning of each section in the Results.

m3. -More in-depth treatment of behavioral action transition times may allow for analysis of activity motifs and synaptic activity that predict these transitions.

This is a great suggestion and prompted us to more closely investigate the neural encoding of transitions between specific actions. We found potential evidence of differential encoding of actions (action “B”) conditioned on the preceding action (action “A”), but ultimately could not conclude if apparent tuning to transitions were in fact tuning to only to action “B” due to limited sampling. The mean duration of behaviors conditioned on the preceding action was 0.09s (min=0.01s, max=1.9s, std=0.1s). This was for all combinations of behaviors for which there were at least 3 occurrences of the ordered pair (shown in the matrix directly below):

m4. -It's not clear if the authors looked for inter-areal synaptic connections (i.e. visual to auditory), or if they focused only on intra-areal connections (i.e. A1 to AuD).

Inter-areal (e.g. visual to auditory) synaptic connections were indeed too rare to perform quantitative analyses; we have modified the text to no longer mention inter-areal synaptic connections.

m5 -Use of alternative manifold embedding techniques, such as UMAP (see “BSOiD”, Hsu et al, 2021), may allow for better inter-subject identification of high-level behavioral motifs.

This is an interesting point, and prompted us to test whether identification of behavioral motifs was improved by other approaches such as UMAP. We took the watershed color-coded scatter plot of training points used for the tSNE embedding (below, left), and performed 2D embedding with UMAP (below, right).

By comparing them, we observed that the UMAP and tSNE embeddings performed similarly in preserving both local structure (*i.e.* tSNE classified point are clustered together in the UMAP space) and the global structure (the gradient of color in the two embeddings is the same, with the exception of “rearing” and “rearing up”). We tested if the distributions were uncorrelated using the Jaccard’s similarity for the k-nearest neighbors. At both micro (k=2) and meso (k=100) scales, the p-value associated with the median of the Jaccard’s similarity was 0.001 (or lower) under a randomization test, indicating that UMAP and tSNE embeddings were very unlikely to be uncorrelated. Since we did not find evidence that the embeddings differed, we respectfully prefer to maintain our original usage of tSNE and associated results.

The “BSOiD” study by Hsu et al. (2021) was published while we were preparing our manuscript last fall, but we now cite it with the other unsupervised approaches for behavioral classification in the Introduction.

m6 -Is it possible to achieve a clearer alignment between the identified elementary poses and species typical actions with assigned ethological meaning?

This is again an interesting suggestion. We also wished to know whether specific postural features aligned with the expression of specific actions, but did not find quantitative evidence to support it. Specifically, we tested whether postural variables occurred more or less often during specific behaviors vs. all other behaviors. A two-sided Mann Whitney U test showed that none of the medians of the posture variables conditioned on any of the actions was different from the median of the same postural variable across all the other actions. P values $< 5 \times 10^{-5} = 0.05/44/23$ (Bonferroni correction).

We did, however, re-align the lists of actions in Supplementary Figs. 5a and b (formerly 4a and b) to highlight commonalities in postural features among similar behaviors (*e.g.* when animals were walking with the head rolled, or while turning in place). The features by which actions are grouped are written in bold on the right of the groupings in the updated Supplementary Fig. 5.

m7 -A clearer explanation of the rationale for use of relative log likelihood ratio as a principal metric for assessing the degree to which each elemental behavioral is represented in each cortical region would be helpful to the general readership.

We used the relative log likelihood ratio (rLLR) because it provides a metric for how much the predictive power of the statistical model suffered when a behavioral covariate is removed. By setting the mean rLLR as the height of the wedges in the polar charts in Figure 3, the “importance” of each feature to the model is conveyed visually. We include this explanation in the Results (lines 98-99) and in the legend of Figure 3.

m8 -Use of a 15 g head weight to control for the possible encoding of muscle contractions as proxies for head position seems unnecessary and perhaps a bit cruel.

We can appreciate this point of concern for the animals’ welfare. However, the added weight to the head did not restrict the free movement of the animals during the experiments, and we limited the use of the weight to the time needed for the recordings. The only measurable behavioral difference was the tendency for the animals’ heads to roll slightly leftward during the weight sessions. We have added text in the Discussion (lines 252-259) to better motivate the experiments in light of long-standing debates in the motor cortex literature:

“The robust encoding of head features allowed us to test whether M1 neurons primarily encoded spatial kinematics or muscle exertion controlling the head, a question classically debated in the context of hand and arm movements in primates (Georgopoulos et al., 1982; Evarts, 1968; Crammond & Kalaska, 1996; Kakei et al., 1999). The addition of the head weight had only minor effects on tuning curve stability and statistical modeling of head-related covariates, indicating that spiking activity was linked more closely with spatial aspects of movement or specific postures (see Ward, 1938), rather the generation of mechanical force. Since the animals were freely moving, signals related to sensory feedback, planning or dynamic pattern generation may have also contributed and cannot be ruled out (Omrani et al., 2017; Aflalo & Graziano, 2006).”

m9-It would have been nice to provide passive somatosensory stimulation so that a hierarchical examination of behavioral vs. sensory encoding (see Fig 3e) could have been performed in somatosensory cortex as well as auditory and visual cortices, particularly given the preponderance of unclassified neurons in S1.

We agree that such experiments could have been informative, but there would have been serious practical complications in delivering sensory stimuli to S1 without impeding the animals’ natural movement in the arena. Unlike auditory or visual input, which are uniformly available throughout the recording arena and do not interfere with mobility, delivering somatosensory stimulation (e.g. stroking or applying vibration) to the body would have interfered with the animals’ movement and biased behavioral sampling. We therefore restricted our tests to the more readily controlled visual and auditory sense modalities, and note this in the Results (lines 171-173):

“We next sought to characterize the degree to which sensory and behavioral and signals overlapped in sensory cortices, focusing on visual and auditory modalities since these could be manipulated without disrupting the animals’ movement.”

m10-It is not specifically mentioned that the activity of visual cortex neurons could not be used to decode the presence or absence of the white noise stimulus, although it is mentioned that the activity of auditory cortex neurons could not be used to detect the luminance condition.

We re-checked the text and found that the result was included already. It can be found in the revision on lines 178-180:

“Decoding analyses confirmed that auditory, but not visual, units predicted sound stimulus presentation (Supplementary Fig. 19b, bottom).”

m11--Has it already been shown in other studies that back movements tend to be represented in caudal motor and somatosensory cortex, while head movements tend to be represented in corresponding rostral areas? If this is a novel finding, perhaps it warrants more discussion (page 3).

This is a relevant question to which we now give more attention in the revised Discussion (and noted above). Cortical microstimulation experiments in anesthetized rats (Hall & Lindholm, 1974; Neafsy et al. 1986) have shown an overall topographical organization of the back and head spanning the posterior-to-anterior extent of somatomotor cortex. This topography, to our awareness, had not been considered in relation to 3D pose or movement in freely moving animals prior to our study. Our findings and the early microstimulation experiments are mentioned in the Discussion, as noted above.

m12-Does 12% seem like a low percentage of neurons in auditory cortex to be modulated “exclusively” by sound? What is the corresponding percentage for all other recorded areas?

While 12% would seem low for the fraction of neurons exclusively modulated by a stimulus in a sensory cortical region, the total fraction of sound-modulated cells in our data was 35.3% (including neurons co-modulated by behavior). This number falls roughly in the middle of the range of sound modulated neurons reported prior studies in A1 of awake rats: Hromádka et al. 2008 reported 5-10% excited by white noise or pure tones; Abolafia et al. 2011 reported 70% responding neurons to white noise. In awake mice, Kato et al. (2015) reported 31% of layer 2/3 auditory cortical neurons responded to tone pips. By comparison, the total fraction of sound modulated neurons in visual areas was 7.4% (82 of 1107 tested neurons), which consisted of a subset of V2L neurons near the A1 border (see Figure 6c). We did not assess sound modulation in motor or somatosensory cortices.

m13-The increased prevalence of “movement inhibited posture units” in auditory cortical areas seems to be of particular interest, and to merit further discussion, particularly in terms of the ability of the model to explain cortical activity during periods of low movement / low arousal.

We have added additional text on this point in the Discussion (lines 309-313):

“Movement-encoding fast spiking cells were also shown to inhibit allocentric posture-encoding cells, which mostly occurred in A2D (e.g., a cell encoding leftward rolling of the head inhibiting a cell for right head roll posture). Circuits with this pattern of connectivity could report minute changes in posture whenever the animal moved and, conversely, maintain a strong readout of the current postural state when the rat was still.”

m14-Is it possible that the observed increased probability of synaptic connection between neurons with heterogeneous behavioral encoding in “V2L” may actually be due to the mis-assignment of neurons that are actually in posterior parietal cortex (PPC)?

We investigated this possibility by re-checking our anatomical delineations, and by conferring with anatomists with expertise in rat PPC. This re-affirmed that the recording locations in the right hemisphere (ranging from -5.5 to -6.5mm AP) were posterior to the caudal boundary of rat PPC, following regional boundaries of (Paxinos & Watson, 2013).

m15- Is it trivial that pairs of neurons receiving common synaptic inputs appear to be homogeneously tuned for behavior (see final point in results)? Or, is this merely an artifact of the analysis? Please provide brief additional discussion on this point.

The analysis used to establish the nature of connections between single units was agnostic to their functional properties, and relied solely on the spike train patterns. In that sense, neurons receiving common input did not have to be homogeneously tuned to behavior *a priori*. We wished to draw attention to the *discrepancy* between the synaptic and common input motifs that we observed, which in our view lends credence to these findings not being an artifact of the analysis.

The fact that the cells receiving common input were homogeneously tuned is relatable to findings where such properties have been hypothesized to be endowed by their upstream sources (e.g. Nelson et al., 2013). What is arguably not trivial is the *difference* between synaptic and common input pairs, where synaptic pairs form more functionally heterogeneous relationships, hopefully lending further credence to reporting the results in general (in relation to Point M5, above). In order to clarify the focus on this discrepancy, we added the following sentence in the Discussion (lines 306-309):

“The finding that homogeneously- and heterogeneously-tuned pairs showed different connectivity motifs, and that the analysis was agnostic to the functional properties of the cells, suggests that functional groupings were not an artifact of the analysis.”

m16-Do your results suggest that studies not using video of the back and hindlimbs of rodents will have dramatically insufficient ability to account for variance in neural activity of primary somatosensory cortex? (See Discussion, paragraph 2). Please provide brief additional discussion on this point.

We did not intend to suggest that behavioral tracking which omits the back and hindlimbs is dramatically insufficient to account for spiking variability in S1. We believe, however, that model performance will vary depending on where in S1 the recording takes place and which features are tracked. Tracking the hindlimbs, for example, may be of little service in explaining spiking activity in a barrel column. In our case the probes were placed in the trunk and hindlimb regions of S1, so tracking the trunk likely carries some explanation for our recordings. We include additional clarification of this point in the Discussion, lines 266-270 (quoted above in relation to Point M6).

m17-Separation of regular -and fast- spiking units (Supplementary Fig. 2) based on spike duration, peak-to-trough ratio, and end-slope is not very convincing.

The method employed to separate regular and fast spiking units was developed by Neill and Stryker (2008) to study visual cortical connections in mice, and has since been used by a number of labs to study behavior-related spiking in the rodent primary visual cortex. These include studies on head movement-driven spiking activity in mice (Bouvier G., et al., 2020) and rats (Guitchounts G. et al, 2020). The latter study, like ours, showed what FS/RS separation looked like in different datasets (Fig. S2H in Guitchounts et al.), which strongly resembled the separation we report. Because of the similar nature of the experiments, we reason that these studies and their methods provide useful points of comparison for our own work. Moreover, the unsupervised k-means clustering approach used to uncover the separating boundary positioned it at 0.5 ms of spike duration in our data, which closely matches previously described separating points between these subtypes of cells (e.g. McCormack DA, et al., 1985; Bartho P. et al., 2003).

m18--Was head position really only updated when the rat was running faster than 10 cm / s (Methods, page 12)? This seems like too high of a threshold and likely results in ignoring many real changes in head position in the overall model.

Head position tracking was continuously monitored at all running speeds. The 10 cm / s threshold was used only to center the coordinate systems for head direction and the animals' running direction, since the head and body are likely to be better aligned when the animal is walking forward or running. This threshold was not used in any other analyses. We have clarified the text accordingly in the Methods (lines 475-6):

"Time points where the speed exceeded 10 cm / s were used to center the coordinate systems for the animals' head direction and running direction."

m19-Use of the acronym "IMU" for "inertial measurement unit" is unnecessarily confusing. These instruments are typically referred to as "accelerometers".

We have changed our usage of "IMU" to accelerometer in the main text, but kept the term as used per manufacturer product information in the Methods.

m20- Please provide a definition of "XYZ Euler angle method".

We have added the following explanation of the XYZ Euler angle method in the Methods, lines 482-484:

"The XYZ Euler angle method indicates the three elemental rotations are intrinsic about the axes of the rotating coordinate system XYZ, solidly with the moving body, which changes its orientation after each elemental rotation."

m21--Behavioral feature vectors seem a bit overconstrained with 133 dimensions (Methods, page 13). Comparison of results after removal of redundant dimensions may be informative.

A 133-dimension feature vector is what was produced from the signal analysis, and we reduced redundant dimensions using PCA, keeping 97.2% of the variance from the

first 22 PCs. We hope this clarification addresses the concern. We have updated the text (Methods, lines 520-522) so that the steps are clearer:

"Redundant dimensions were removed using Principal Component Analysis (PCA), which indicated that the first 22 principal components explained 97.2% of the variance".

m22--Downsampling of feature vectors to 1 Hz (Methods, page 13) may be excessive and prevent proper high-resolution alignment of neural activity with behavioral action transitions. A comparison with results after downsampling to 10-30 Hz, a range that better matches the frequency components of natural rodent movement, would be informative.

We fully agree that downsampling to 1Hz would have prevented proper alignment of neural activity and behavioral transitions, but 1Hz downsampling was only performed in our data during embedding, which was necessary to avoid artifacts driven by temporal autocorrelations. Our analysis of behavior and its correlation with neural activity (encoding, decoding) operated at 120Hz. This point has been clarified in the Methods (lines 518-520):

"Feature vectors were downsampled to 1 Hz, and data were pooled across animals and conditions. Downsampling was performed only for the sake of embedding, and did not affect behavioral analyses or correlations with neural activity (e.g. encoding, decoding), which used 120Hz tracking data."

m23- The authors often state that movement drives or influences activity – however, this is only a correlation. I recommend the authors use more circumspect language such as "movement is associated with....".

We have adopted more circumspect language as suggested by the Reviewer and, where applicable, employed more exact terms referring to neural "encoding" of actions, or that movement was associated with neural spiking activity (e.g. line 7 of the Introduction, or line 237 of the Discussion).

m24- The results from the fast spiking category of cells is commented upon in the discussion but I couldn't find it presented in the main body of the results section.

We believe this referred to the functional connectivity analyses in the original Figure 4. We have revised the text in that section of the Results so it is explicit that excitatory connections were driven by regular spiking units and that inhibitory connections were driven by fast-spiking units (lines 201-205, quoted above in relation to point M5).

Reviewer #2 (Remarks to the Author):

General comments:

The goal of the study is to examine the neural representation of natural behaviors - both segmented 'momentary behaviors' (e.g., rearing, turning), and more "elementary" behavioral features (posture, movement, position, etc.). The reported findings are that (1) momentary behaviors were represented across all areas recorded, (2) behavioral features showed organization across areas, and (3) tuning properties of synaptically coupled cells differed across areas. The authors collected high-quality video recordings of freely behaving rats while recording single unit activity across multiple brain areas using Neuropixels probes. The methodology to quantify behavior appears sound and consistent with literature, and the statistical methodology behind fitting the array of decoders and GLMs in this paper appears sound.

The main weaknesses in the paper are that the primary evidence used to support claim (b) is based on models that appear to have very low performance in capturing neural responses and further that the differences in recording depth across different areas could be confounding the conclusions. Second, the data and analysis supporting claim (c), as presented in Figure 4, are somewhat weak. Finally, too much of the substance of the paper is in the Extended Data, making it very challenging to grasp the significance of the results as presented. This is a novel and extensive dataset, and the results of these analyses are of interest, but there are serious shortcomings in its current form.

Strengths of the paper:

It is clear that a tremendous amount of work went into acquiring, processing, and analyzing the data shown here, and the thoroughness of the documentation of technical details through the extended data figures is excellent. Figures 1 and 3 clearly convey results of interest.

To highlight some of the strengths in these parts of the paper: in Figure 1, the authors outline the methodology for segmenting natural behavior of freely moving rats, which produces a map of identifiable momentary behaviors (center of panel (c)). These momentary behaviors can be decoded from neural activity in all the brain areas recorded, with varying degrees of accuracy. Most momentary behaviors are decoded above a chance-level set by the prior distribution of momentary behaviors, some are decodable with very high accuracy. This is interesting, in that most previous decoding analysis have not focused on representation of a wide range of specific actions, and the fact that specific actions are represented is a novel finding. Figure 3 shows how visual inputs and auditory inputs modulate activity, and that neurons across both areas have similarly broadly distributed representation of behavior variables and sensory variables. This broad representation of behavioral alongside sensory information is important for understanding what cortical computation is – at the most basic level, the inputs and the outputs of these computations are more complex than typically appreciated.

We thank Reviewer 2 for their thorough consideration of the work, and we are encouraged to read that they feel our methods and analytical framework are sound, and that our dataset is novel and extensive. We agree with the Reviewer that it is interesting to see that a wide range of specific actions was broadly represented in cortex, and we were happy to read that they view the result showing side-by-side representation of sensory information and behavior as important to understanding cortical computation. We have worked to address each of the points of concern raised by the Reviewer, and respond to each of them below.

Detailed comments on major concerns:

M1 (multi-part; continues below). Regarding evidence supporting claim (2) The analysis supporting claim (b) is primarily in Figure 2 ("Distinct cortices show differential tuning to posture, movement, and self-motion"). The authors fit individual cells with GLMs through a model selection procedure to identify which behavioral features contributed the most to the model performance. This figure represents how the distribution of those features differs across different brain areas.

Before discussing further, Figure 2 is missing any information on how well the GLMs explained spiking activity, which appears in Extended Data Fig. 7. The measure of model performance is the pseudo- R^2 . Pseudo- R^2 would be 0 if the model explained no variance, and 1 if it exactly predicted the spiking of cell. Ext. Data Fig. 7 shows that a small minority of cells have a pseudo- R^2 greater than 0.1. The medians (reported in the main text) across different areas range from 0.01 to 0.03. Thus, to interpret these values, the authors need to explain what a "good" value of pseudo- R^2 is for single units during freely moving behavior. This is challenging, as there is no repeated trial structure to use to assess variance across repeated conditions to provide a ceiling to the possible model variance explained. One possibility is to follow a similar approach to the Stringer (Harris, Carandini) 2019 paper, in which they compared feature-derived predictions of neural activity to predictions generated by simultaneously recorded cells. In other words, if the GLM instead used neighboring cell activity as a predictive feature, how well would it perform? This could at least provide a scale by which to assess model performance. Further, I want to know, if you restrict analysis to cells for which the GLM explained a meaningful amount of variance, how does the feature analysis in Figure 2 change?

We wish to first clarify that the relative log-likelihood ratios (rLLR) values were indicated on the polar charts in original Figure 2 (ranging from 0.1 to 0.6), intended to convey mean importance of each covariate in the model. Due to space limitations, pseudo- R^2 values were shown fully in the original Supplementary Figure 7 (now Supplementary Fig. 12).

To address the concern regarding low pseudo- R^2 values from the original GLM (referred to here as the "Covariates Model"), we performed additional peer prediction analyses using spiking activity from simultaneously recorded cells (separated by >5 recording sites on the probe) as a predicted feature. We refer to this model as the "Peer Model", built according to the Stringer et al. (2019) paper suggested by the Reviewer.

We then compared the proportions of cells tuned to at least one behavioral feature from the Covariates Model vs. the subset of cells with larger pseudo- R^2 values than the corresponding Peer Models (*i.e.* cells with "good" pseudo- R^2 values). We found that the defining regional differences in encoding properties were upheld when only considering the cells with "good" pseudo- R^2 values. That is, head movement, head posture and self-motion still dominated in visual cortices; allo- and egocentric head posture still dominate in auditory cortex; motor regions still encoded the trunk and head, and somatosensory was still dominated by trunk representation (next page):

Proportion of tuning for all selected cells vs. with “high” R²

Since these additional analyses support our original findings and conclusions as they were, we respectfully suggest leaving the Figure and GLM portion of the paper as they were originally.

M1. Turning now to the presentation as-is in Figure 2, there are four groupings of plots, each consisting of a pie chart ('proportion of n = ## cells'), a bar plot, and a polar chart ('contribution to model performance'). The proportion of cells refers to the proportion of cells for which each of these behavioral features was the top covariate; between ~20% (Motor) and ~50% (Somatosensory) of cells are "unclassified," and presumably not included in the wedge plot at right or the bar plots below. The bar plots show how many covariates were included in the selected model for each cell. This is where one possible difference across areas becomes apparent, where some 40% of somatosensory cells only have one covariate and < 2% have 6 or more covariates; in motor areas especially, cells tend to have more covariates. This is potentially interesting, except that the cells sampled in the somatosensory areas are reported to

be primarily Layer 2/3, while the other areas are all sampling deeper layers, either instead of or in addition to L2/3 (manuscript, lines 82 - 85). Related, extended data Figure S8 shows V1 vs V2 and A1 vs A2D differences in the number of covariates, but also tractography showing the probe crossing into different layers. If the authors have enough cells sampled from L5/6 in multiple areas, could they repeat this analysis restricted to that subset? Similarly, with L2/3? Do these trends persist?

This is an excellent point, and was raised by other Reviewers as well. We have now included analyses with the data separated by superficial and deep layers in visual and motor cortices, as well as for layer 4 vs. deep layers in auditory cortex:

For the lower-level features, GLM analyses showed that regional variations persisted across deep and superficial layers in visual and auditory cortices, which mainly encoded head posture, head movement and self-motion (new Supplementary Figure 13, above). Similar to the action-encoding results, tuning was most widespread in deep layers of visual and auditory cortices which, at least for visual cortex, is consistent with earlier work showing that deep layer V1 neurons respond to head rotations in darkness (Velez-Fort et al. 2018; Bouvier et al. 2020). Both those and our findings point to the deeper cortical layers as a possible site of integration for motor input from midline motor cortex (Reep et al. 1987) and vestibular signals relayed by retrosplenial cortex (Velez-Fort et al. 2018), noted in the Discussion, lines 280-284.

Supplementary Figure 13:

In motor cortex, the laminar analysis uncovered differences across deep and superficial layers, with the trunk being encoded more strongly in superficial layers and the head represented more in deep layers (below). However, due to the tilt of the neuropixels, layer 2/3 was sampled more posteriorly, and layer 5 was more anterior— which follows an established topography of head and trunk representation over the dorsal cortical surface in rats (Head & Lindholm, 1974; Neafsey et al., 1986). Further, our own topographical analysis (now main Figure 4) showed a continuous evolution of back-to-head representation progressing from posterior to anterior recording sites. So, while we cannot rule out differences in cell layers, the result can be accounted for by known anatomy. We note these earlier anatomical mapping studies and the topographical organization in our recordings in the Discussion, lines 270-272.

To further substantiate that neurons in different layers encode low-level behavioral features, we included additional examples of stable tuning curves from single cells in main Figure 2 (superficial or granular layers) and Supplementary Figure 10 (deep layers).

M1. Finally, the polar chart in each subpanel is represents 'contribution to model performance' of behavioral features. I am somewhat confused on how it was constructed. The angular extent of each wedge is 'the fraction to which each covariate was represented among all those selected.' Does this mean that you take every model for every single unit in the selected brain area, enumerate all the behavioral features that were ever used, and then report the fraction of that pool by feature?

Is the contribution weighted by how much additional predictive power each feature contributed? Which cells are included in this analysis -- for instance, a substantial fraction of cells have "unclassified" top features, are these excluded? and are cells with pseudo- R^2 of 0.01 included on the same footing as cells with pseudo- R^2 of 0.1 or 0.2?

We apologize for not explaining sufficiently the construction of the polar charts in the original manuscript. For clarification:

- (i) The pie charts in the upper left of each panel show the proportion of all recorded cells which took a particular feature as the first covariate in the GLM and were assigned a color-coded wedge based on that covariate.
- (ii) The polar charts were made using only those cells from the colored pie wedges ("Unclassified" cells were excluded), and only considered features that were included in the full statistical model after model selection. For example, a neuron best explained by "allocentric head pitch + azimuth" would fall in a dark purple wedge in the pie chart, but the contribution of each feature of (head pitch and azimuth) are shown in separate wedges in the polar chart.

Cells with low and high pseudo- R^2 values are not separated, since separating them did not change the trends in tuning observed in each region.

The legend for Figure 3 has also been revised:

Fig. 3 | Distinct cortices show differential tuning to posture, movement and self-motion. a, (Top left) **Pie charts showing the fraction** of single units in visual cortices (V1, V2L and V2M) that incorporated specific behavioral features as the first covariate (largest mean cross-validated log-likelihood among single covariates models) in model selection (for feature identification, refer to the color-coded legend at bottom). (Lower

left) **Bar graph indicating** the percentages of single units statistically linked to one or any larger number of behavioral covariates. (Right) **Polar charts denoting the relative importance of individual covariates included in the full model, after model selection, with "unclassified" cells excluded.** Wedge length denotes the mean cross-validated rLLR (methods) of each covariate across the set of models it was included in. The width of each wedge reflects the **fraction of times that feature was selected among the other covariates. Wedge length and width are independent of each other (i.e. the area does not reflect effect size).**

M1. As a somewhat minor point, I would argue that it is perceptually confusing to set the 'length' of each wedge (rather than the area) to denote the relative log-likelihood ratio. Is 0.6 really four times higher than 0.3 for rLLRs? Again, this is averaged across all the models the covariate was included in, but how does it change if restricted to models with high pseudo-R²? It may be worth considering if polar charts are the best way to convey this information in this figure. As is, given the concerns described above, I do not know how to interpret the data shown.

Our goal with the polar charts in the current Figure 3 is to convey the importance of each behavioral feature in each cortical area in a visually intuitive way (which we now clarify in the Results, lines 98-99). We previously tried other conventions, such as bubble plots, but found polar charts to be the most efficient way to fit the results in a single figure.

The width of the wedges in the polar charts indicates the number of times that a particular feature was selected. The length of the wedge indicates how much the predictive power of the model suffered when that feature was removed. The length and width of the wedges should be viewed as 2 independent properties; we do not merge the values, and the area is not meant to convey the size of the effect. We clarify this point in the revised legend of Figure 3 (quoted above)

To further ease concerns about the way in which the data were presented, we include here comparisons of (i) polar charts showing mean rLLR values generated using full models as in the paper, and (ii) polar charts restricted to "good" cells (pseudo-R² values higher than corresponding peer-prediction models). This again confirms that the results presented in the paper hold up when only considering models with "good" performance (next page; note the difference in scale for Visual cortex polar charts):

M2. Regarding evidence supporting claim (d)

In the abstract, the authors state, "tuning properties of synaptically coupled cells also exhibited connection patterns suggestive of different uses of behavioral information within and between regions." This is primarily addressed in Figure 4.

In Figure 4, the authors use cross-correlograms between pairs of units to identify putative synaptic connections. Figure 4a shows an anatomical projection of auditory (green) and visual (pink) units, with grayscale lines showing putative connections. (The color bar (labeled $\log_{10} p$) uses the same darkness to represent a p-value of 1 and of 10^{-8} and should be reconsidered; at any rate, I cannot see where this colorscale is used in the figure.) The connectivity drawn in anatomical space is difficult to interpret; are cells truly unlikely to connect to their nearest neighbors, but likely to be connected across long A-P distance (roughly, left to right on this plot)? Figure 4b shows a scatter of synaptic strength against distance, with no significant trends.

To respond to the points in order: the color bar in the previous Figure 4a (now Figure 6a) indicated the significance of the cross correlogram values in each bin. The colors of the scale bar and data have been highlighted so that the color-code for significance is more visible than before. The middle of the scale bar reflects the significance threshold of $p = 0.0001$. We have grouped the scale bar with the cross-correlogram results so it is clearer that they go together.

We have adjusted the rendering of connectivity in anatomical space (Fig. 6a, right), having moved the colored dots (neurons) to the back so that the lines depicting anatomical connections are more visible. Cells were connected to nearby neighbors as well as to cells over longer (~50 micron) distances, which were limited by the total width of the shank. The plot is only intended to provide a sense of how the cells are positioned relative to each other,

and to show that they do not cluster only in one specific point along the probe. The scatter plots of synaptic strength and anatomical distance in panel 6b show that there were no significant trends between synaptic strength and distance. The lack of a trend is in our view also informative – strong connections, in any area we looked, were not limited to an immediate proximity, which motivated further functional analyses.

Figure 4c is where the main point is made: that tuning-specific connectivity suggests different uses of behavioral information within and between regions. First, with respect to "between regions," I do not see in this figure any analysis of synapses between regions, so perhaps something else is meant.

The Reviewer is indeed correct that the putative synaptic connections occurred between cells within the same region, and we have modified the text in the Abstract and Results accordingly.

The question is then, are there different tuning-specific patterns of connection within regions? The evidence for this is thin. Fig 4c represents the number of synapses found between functionally identified pairs of cells, where "function" is Mo (movement-responsive), Po (posture-responsive), LM (light-modulated) and SM (sound-modulated). (The authors claim that these subsets are largely non-overlapping, but this is not shown.)

The direct evidence of non-overlapping cells pairs proved difficult to convey visually in the original UMAP plots in 4c, so they were removed from the main figure. The functional results are now shown together in Supplementary Figure 21, along with supporting statistics showing a lack of preferential coupling between functionally similar neurons (Supplementary Fig. 21e, lower panel). We felt this was an important result to include since, as opposed to orientation tuning in visual cortex, in which similarly tuned neurons form functional sub-groups (e.g. Ko. et al., Nature, 2011), functionally disparate neurons in our data were just as likely to have strong connections as functionally similar neurons. This raises the prospect that communication between functionally disparate cells is important, possibly even more so when the content (behavioral vs. sensory signals) is different. We make this point now in the last sentence of the Results section in the revision.

In Fig. 4c, a shuffle test within the synapses found in each area was performed. Note that there were different numbers of cells in visual vs auditory areas, and about 2x as many synapses found in visual areas. The overall number of detected synapses is also small: from 5 to 15 in any auditory cortex group, and from about 10 to 40 in visual areas. If this is out of the ~10k possible synapses in the 100s of simultaneously recorded cells, this is a very small fraction, and I do not know how much weight I should give these results.

Regarding the number of putative synapses in visual and auditory areas, there were 3x more neurons recorded in visual areas, so this would be expected to be reflected in the number of putative synaptic connections identified as well. The total number of putative synaptic connections reported in the study is 650: 247 for visual, 107 for auditory, 35 for somatosensory and 261 for motor cortices. We agree in proceeding with caution regarding sampling that is limited, but we note that the inclusion criteria for counting cells as synaptically connected were strict (cross correlation p-values < 0.0001), as were criteria for significance in the rate of putative synaptic connections

(exceeding 99% confidence intervals beyond shuffled data). We have acknowledged the limited sample size by modifying wording in the revised manuscript:

“We found a limited, but informative number of putative excitatory synaptic connections (driven by RS units) and inhibitory connections (driven by FS units) of varying strength within auditory (n = 107 total connections) and visual cortices (n = 247) (Fig. 6a, b; Methods) as well as somatosensory (n = 35) and motor cortices (n = 181) (Supplementary Fig. 21a, b).”

We nevertheless believe it is important to report these results for the following reasons: (1) The field of single-unit electrophysiology as a whole still very much relies on drawing statistical inferences on limited samples, *e.g.*, in this study we make inferences based on samples of ~1000 neurons *per* region, though anatomically distinct areas are composed of different layers with tens of millions of neurons each. Given this, we believe it is still relevant to disclose our findings fully, given the particular rigor applied in constructing statistical arguments. (2) We would also argue that recordings as expansive as the ones we are reporting, in freely-moving and freely-sensing animals, can still be considered a valuable rarity. They offer new insights which point to fundamental network properties of cortical areas which lack recurrent connections characteristic of other networks (*e.g.* in the head direction or hippocampal systems), which significantly lowers the probability of detecting putative synaptic relationships, but are still valuable as a reference point for future studies. (3) Although the absolute number of reported putative connections is limited, we argue that the analyses employed on such data still enable us to uncover meaningful effects. Results reported in Figure 6c, (formerly Fig. 4c) compare observed data to that if single units we recorded were randomly connected. Notably, previously reported findings of movement-inhibited sound-responsive units in A1 (Schneider et al., 2014) were replicated in our experiments, arguably lending credence to other synaptic motifs we are reporting.

The shuffle test represents the null hypothesis that synapses occur with equal rate independent of functional cell class. In visual areas, there appear to be more Mo to Mo synapses (both excitatory and inhibitory), more excitatory Po to Mo synapses, and more inhibitory Po to Mo synapses. Comparing that result to the auditory result is challenging. Due to the smaller number of cells recorded in auditory areas, it is harder to detect functional patterns of synaptic connectivity over the shuffle baseline. Slightly more inhibitory Mo to Po synapses and Mo to SM synapses were found. The cells and significant connections, for both areas are shown in a functional space, which cannot be understood without referring to the details in Ext. Data Figure 13.

The shuffle test does not assume that synapses occur with an equal rate, independent of functional cell class. On the contrary, the most represented functional classes (where “not classified” is included also) would, on average, tend to form more random synapses, as they would be more likely to get chosen by chance. This is precisely what we were after: we wanted to know whether the relative differences in putative connection subtypes were an artifact of some functional subtypes occurring more often. As the Reviewer rightly observed, certain synapses appear more often than others (movement parameters are, for example, more prevalent in visual and postural ones in auditory areas, as established by the GLM (Figure 3)), and our results indicate that the patterns we observe do not match those that would be established by forming completely random connections. Moreover, we respectfully point out that the comment that it is harder to detect functional patterns of synaptic connectivity in auditory pairs because of lower numbers is not correct. The method we employed can be utilized regardless of the total number of pairs, because each shuffle iteration constructs the same number of random pairs as those that have been observed – in that sense, having more or fewer pairs does not make the shuffle test easier or harder to pass.

Regarding functional space plots, they are now only shown in Supplementary Figure 21, where the analysis is explained.

I would suggest that the authors consider whether Figure 4 truly belongs in this paper, and if it does, whether Ext. Data Fig. 13 should be included as well. How does it support the main point about 'encoding structure employed across neocortex'?

The message of the paper, which we hope is clearer in the revised manuscript, is about differences in tuning in different cortical areas, and the functional analyses in Figure 6 do not fall short in supporting that conclusion. We believe the added motivating text, along with additional explanation of the results, make it clearer that these analyses contribute to the narrative and should therefore remain in the paper.

Minor concerns:

m1.A general minor concern: burying the lede

There are several interesting statements that are almost throw-aways in the paper as written. For instance, recordings were performed in dark and light conditions, with the decoding analysis restricted to the dark condition [lines 53-54]. In Figure 3, the authors show a dramatic shift (in functional UMAP space, at least) of neural activity in visual areas in lights on vs lights off conditions. The analyses in Figure 2 also utilized the dark condition (lines 70-71), reporting that visual areas maintain information about head position in world coordinates. If this is really in the absence of visual inputs, how is a world-coordinate representation maintained? Potentially through some sort of angular path integration or triangulation with auditory inputs (which, like visual inputs, have sources that could provide stable world-reference coordinates). At the least, are the GLM features and GLM performances in visual areas different when trained on dark vs. lights-on recordings?

This is an interesting question and we are glad to provide more clarification. The world-based reference frame for pitch and roll here is defined by measuring rotations of the head relative to the arena floor (*i.e.* relative to the direction of gravity), so vestibular signaling, which is constant across lights-on and -off conditions, and which drives head-motion tuning in deep visual cortical layers (Velez-Fort et al., 2018), is very likely a key contributor to visual cortical tuning we found in darkness. Vestibular signals contributing to movement in azimuth will also be intact, regardless of whether it is measured relative to the room (allocentric) or the trunk (egocentric). Proprioceptive and efference copy signals also remain in darkness, the latter of which has also been shown to contribute to head movement representation in visual cortex (Guitchounts et al. 2020). Together, these signals would still give strong information about the position and movement of the head. As requested, we performed new GLM analyses on visual cortex recordings (as well as all other regions) and compared stability across light-light and light-dark sessions. Visual cortex was the least stable of the 4 areas (shown in Supplementary Figure 20), and we note these findings in lines 188-195 of the Results:

"We next compared the stability of behavioral representation across light and dark conditions, and found that GLM-selected features in visual regions were more stable across light sessions than between light and dark sessions (Supplementary Fig. 20). Planar body motion, allocentric head movement, and allo- and egocentric head posture were the least stable covariates, but were still maintained in darkness, presumably by vestibular, proprioceptive or efference copy signals (Rancz et al., 2015; Guitchounts et al., 2020; Bouvier et al., 2020; Véléz-Fort et al., 2018). Tuning in auditory cortex, on the other hand,

was stable across light and dark conditions (Supplementary Fig. 20), as was the case in somatosensory and motor cortices ($p > 0.05$, two sample Z-test for proportions; not shown).”

m2. Another result that seemed deserving of more attention was the perturbation experiment in which a 15 g weight was added to the head mount, with results reported in Extended Data Figure 10 and summarized in a few lines (106-113) in the results.

We have included additional context and discussion regarding the 15 g head weight experiments, both in the Results (lines 142-144) and Discussion, lines 252-259:

“The robust encoding of head features allowed us to test whether M1 neurons primarily encoded spatial kinematics or muscle exertion controlling the head, a question classically debated in the context of hand and arm movements in primates (Georgopoulos et al., 1982; Evarts, 1968; Crammond & Kalaska, 1996; Kakei et al., 1999). The addition of the head weight had only minor effects on tuning curves and statistical models of head-related covariates, indicating that spiking activity was linked more closely with spatial kinematics or specific postures (see Ward, 1938), rather than the generation of force. Since the animals were freely moving, signals related to sensory feedback, planning or dynamic pattern generation may have also contributed and cannot be ruled out (Omrani et al., 2017; Aflalo & Graziano, 2006).”

m3. Lines 44 -46: "The animals combined ethogram consisted of 44 independent modular actions ... whose sequence order was best described by a set of transition probabilities" • Is it shown that transition probabilities are the best description? My understanding of these ethograms is that the behavioral state dynamics tend to be highly non-Markovian, suggesting that transition probabilities alone are insufficient. See, e.g., Alba, Berman, Bialek and Shaevitz, arxiv, 2020.

We did not test whether transition probabilities were in fact the best description for the behavioral data, and indeed behavioral transitions in rats have been shown to be highly non-Markovian at time scales of seconds to minutes (e.g. Marshall et. al 2021). We revised the description of the action transitions to state that the estimated transition probabilities were highly stable across recording sessions in different lighting conditions (Pearson’s R of 0.968, $p = 0.0006$) (Results, lines 51-55):

“The animals’ combined ethogram consisted of 44 independent modular actions (Fig. 1c and Supplementary Fig. 4a) comprised of unique composites of rudimentary pose and movement features (Supplementary Fig. 5a, b), which followed characteristic transition probabilities that were conserved across light and dark recording conditions (Supplementary Fig. 4b).”

m4. Figure 1:

The top panel is relatively clear, as is the behavior map and example postures in the bottom panel. I do not know how to interpret actions 1 to 44 without a map, and I don't understand what new information the map provides that bar heights do not. Could you pick one or the other? Further, it is a non-intuitive choice to designate dark bars for accuracy of decoding with the prior and light bars for accuracy of decoding using neural activity, while in the maps of decoding, it appears to be light for low accuracy and dark for high accuracy.

We have revised Figure 1 substantially. We removed the circular bar plot and tSNE plots showing decoder accuracy, previously in Figure 1c. The decoder results are now shown in a stand-alone sub-panel in Figure 1e, with chance levels in the bar charts shown in dark grey. The tSNE plots with color-coded decoder accuracy were replaced by encoding results (Fig. 1c) to complement the decoding results, and the distributions of encoded actions, which did not differ between areas, are now shown in Figure 1d.

m5. A confusion matrix would be good to see in the extended data to understand how distinguishable similar behaviors are.

We have added confusion matrices for each brain region in a new Supplementary Figure 8 and refer to them in the Results (lines 76-77).

m6.Minor grammar: "of each covariate across the set of models it was included in" -> "of each covariate across the set of models in which it was included"

We have corrected the text in the Figure legend (now Figure 3a).

m7. Figure 3:

I do not understand why the average lines in the A1 plots in (B) are blue and yellow; do these correspond to blue and yellow clusters in the UMAP plots? I don't think so, as those seem more likely to correspond to the blue and yellow in panel (C), where light-modulation is demonstrated. Related, why is there no corresponding UMAP plot for sound modulation?

We have revised and simplified Figure 3 (now Figure 5), including changing the yellow and blue color schemes from before to red (luminance modulated) and blue (sound modulated) in Figure 5a and b. The color schemes match the luminance and sound modulation data shown on the Neuropixels schematic in 5c.

The UMAP plots, now in Supplementary Figure 19, are color-coded according to the same color scheme in the rest of the paper (auditory units in cyan, visual in pink/red). UMAP plots were generated to demonstrate light modulation effects because those data were not collected on a trial-to-trial basis– the room lights were on or off for the whole session. Sound modulation effects, on the other hand, could be shown clearly on a trial-by-trial basis in raw spike rasters each time the sound was presented, so UMAP plots were not necessary.

m8.The gradients in (D) are intriguing, but also seem to move in and out of different layers in different areas (as portrayed in Extended Data Fig 1D and f). This makes it difficult to determine whether it is a proximal-distal gradient, or a laminar gradient.

This is an interesting possibility but is difficult to parse purely by layer, since layers were not sampled evenly across visual and auditory regions, and ~3x more neurons were recorded in visual than auditory cortices. We did, however, calculate the proportion of sound- or luminance-modulated neurons by layer within each region. This showed that both sound and luminance modulation were highest in layer 4 of auditory and visual cortices, respectively, with slightly lower values in superficial and deep layers, where recorded. We have added these results to the legend of Figure 5:

“Sound- and light-modulation within auditory and visual cortices, respectively, were highest in layer 4 (L4). In A1, 48% of L4 neurons had significant sound modulation indices (SMIs), followed by layers 5 and 6, with 26% and 25%, respectively. In V1, 30% of layer 4 neurons had significant luminance modulation indices (LMIs); layer 5 had 27%, layer 6 had 24%, and layer 2/3 had 23%.”

m9. Figure 3(E) is nice: it appears there is a distributed representation of sensory inputs and behavior across auditory and visual areas, and that this is more or less similar between FS and RS units. It would be helpful to have unit counts at each branch.

We have added the proportions of FS and RS units at each level in the dendrograms in the Figure (now Figure 5d).

m10. Figure 4:

In Fig. 4C, use the same x limits on all subplots. It would also be useful to know the number of possible connections.

We respectfully oppose the suggestion to set the same x-limits in this case since doing so potentially helps obscure true effects that we try to present, and in datasets where the expected number of synaptic connections is different. Given that different areas and connection subtypes have different numbers of total pairs, we argue that the number of pairs itself is not as relevant as the presentation of whether that number exceeds what is expected by chance – for different areas/connection subtypes – this will be a different value.

As for the number of possible connections, we have included in the Methods that a total of 557,620 possible connections were tested in the right hemisphere (visual and auditory cortices), and 606,301 possible connections were tested in the left hemisphere (somatosensory and motor areas) (Methods, lines 697-699). We again point out that our inclusion criteria were strict ($p < 0.0001$ for CCG values in two consecutive bins within the ± 1.6 -4 ms window of the spike, and no threshold passing in the ± 1.6 -0 ms range). The proportion of detected connections relative to all possible connections says more about recurrent properties or propensity for strong connections in a network, and little about whether detected connection profiles could be observed by chance.

In Fig. 4 D, do not use the same color for FS/RS outline that you use in Fig 4B to indicate auditory regions. A1 and A2D are yellow and brown - why not shades of green to link to panel B and C?

We thank the Reviewer for catching this. The color scheme in the figure has been updated so that A1 and A2 are shades of green; FS neurons are outlined in magenta and RS neurons in purple (per the revised color scheme in Supplementary Figure 2).

The caption says 'We note that the outlined pair groups are largely separated.' for Fig 4C. This separation is not clear to me in the UMAP plot at right. More details on the functional subspace would be helpful for interpreting this plot. Extended Data Fig. 13 is useful in this regard, but Figure 4 needs to stand on its own.

Due to the difficulty in visually conveying that neurons forming different pair groups were largely distinct neurons (*i.e.*, there were not many neurons that fell in multiple categories), we removed the UMAP embedding from Fig. 6c. All functional subspace

analyses are in Supplementary Figure 21, where they are explained. We have added text to motivate the functional embedding analyses, and contextualizing text for the results (noted above in relation to point M2).

m11. Extended Data Fig. 6

All x-axes should be the same

The x-label says cross-validated log-likelihood, but the caption says cross-validated rLLRs.

The x-axes have been adjusted and the label has been updated, we thank the Reviewer for catching the error.

Reviewer #3 (Remarks to the Author):

Summary:

Mimica*, Tombaz* et al present an impressive and unique study that examines the functional coding properties of neurons in somatosensory, motor, visual, and auditory cortices during natural behavior. By first parsing the natural behavior into actions (e.g. rearing), the authors find that most actions can be decoded in each region. By then parsing the behavior into continuous movements, such as head orientation in egocentric and allocentric coordinates, the authors uncover coding properties specific to each region. E.g. visual cortex encodes allocentric head movement, while motor cortex encodes egocentric movement. The authors then show that neurons in visual and auditory cortices encode visual and auditory information alongside behavioral information, and that there is structure to the putative synaptic connections in visual and auditory regions.

Overall, the data collected are quite impressive, as are the wide range of presented analyses. Understanding neural activity during natural behavior is of high interest to many but an incredibly difficult task. Here the authors have very bravely waded into those waters and come out with some interesting results. However, I felt there were some issues with the manuscript in its current form. First, while in general I can get behind the sharing of results even if their full impact or interpretation isn't entirely clear yet (as every paper is a small part of a bigger story), I felt that some of the results presented here really didn't have quite enough contextualization, interpretation, or motivation. There are a lot of facts in this paper, and in some places it was hard to know what to take away, or what exactly was learned that wasn't clear before. More details on this point are listed below, but overall, I felt this applied most to results in Figures 3 and 4. This issue is also exacerbated by the current short format of the article (which I'm not sure is necessary, although I admit I don't know the character limits here). Further, some of the figures were hard to parse, and some analyses or methodological details were either missing or hard to find. These issues and others are detailed below.

We thank the Reviewer for their thorough and largely positive evaluation of the study. We have taken their concerns into account, in particular by giving additional background and motivating text to link the experiments and findings in the paper. We have also moved data from the Supplementary material into main figures in areas of the paper, in particular to substantiate our finding that all cortical regions encoded posture and movement features in different cortical layers, where sampled. Responses to the concerns and suggestions are listed point-by-point below.

Major comments:

As mentioned above, I felt that in a couple places the figures were difficult to parse. While some figure panels are mentioned in other comments, issues pertaining to the rest have been collected into the list below.

M1. Figure 1- Artistically speaking, the circle looks nice. But it is very hard to visualize the decoding accuracy or compare the size of the bars in different regions in this format. While less exciting, it might be easier on the reader (and better convey the points you make in the text) to show rows of bar plots. In addition, the comparison to the control decoder, in which decoding is based on the prior, is vital, but difficult to see here.

These are helpful suggestions; we agree that, although the original version of Figure 1 was visually appealing, it was difficult to parse. We removed the circular bar plot showing decoder accuracy in Figure 1c and show the decoding results now as a stand-alone sub-panel in Figure 1e. Chance levels in the bar charts are shown in dark gray. The tSNE plots with color-coded decoder accuracy have also been removed, and in their place we show encoding results to complement the decoding results. Further we show the distributions of encoded actions in each overarching area in Figure 1d.

M1b. Figure 3

i. I find the lines in Figure 3b,c to be visually confusing - I see it is a double axis, but it makes it quite hard to interpret.

We have simplified Figure 3 (now Figure 5) in several ways. In this case, we included additional text in the figure legend explaining the left axis (unit number) and right (colorimetric scale for unit activity relative to max).

“Fig. 5 | Auditory and visual cortices show similar prevalence and overlap of sensory and behavioral signals. a, (Top) White noise stimulation paradigm schematic. (Bottom) Trial averaged activity for single cells (arranged by unit number, left axis) and ensemble averaged activity (overlaid trace) of all significantly suppressed (left) and sound-excited (right) single units in auditory cortices during sequences of sound stimulation. Colorimetric axis denoting activity relative to max are shown to the right.”

ii. The proportion breakdown in the bottom of Fig 3b,c is really difficult to parse.

To make sub-panels 3b and c easier to parse, we moved them to their own Supplementary Figure 19, enlarged them, and changed the color scheme such that the colors for RS and FS cells (purple and magenta) are more visually distinct from colors indicating sound modulation (blue) or light modulation (red).

M1c. Extended Data Figures –

i. Extended Data Fig 4a,b are difficult to interpret. Can they be summarized or restructured to highlight the most meaningful relationships between behaviors?

Great suggestion. We re-grouped the actions in those figures (now Supplementary Figs. 5a and 5b) based on features which were most conspicuous when labeling the

actions from video (e.g. if the animals were sitting still, walking, head up or turned, etc.). The features by which the actions were grouped are written in bold on the right of the groupings in the figure.

M1dii. Extended Data 10 - this figure has a ton of information in it, but with the number of panels and subpanels, it's difficult to pull out the relevant facts. In some cases, I was not sure what was being plotted - e.g., what the 'information rate' in panel e is being computed from (the encoding model?). I would suggest substantially simplifying this figure.

We have simplified Extended Data figure 10 (now Supplementary Figure 16) to emphasize key points. It now consists of schematics and sample data in the top panel (a); the middle panel (b) shows stable tuning curves overlaid from each condition; (c) compares the properties of the tuning curves and spiking properties of the cells encoding head features; and in (d) the statistical results of compared features. We have added text in the figure legend to more clearly state that the comparisons were based on the tuning curves of the cells themselves and added that the information rate was calculated in bits / spike for each cell, per Skaggs et al. 1993.

The calculation of baseline stability was removed, but is still described in the Methods, and the GLM results were placed in their own Supplementary Figure 17.

M2. How were spikes sorted across multiple 'schedules' (the group of 4 sessions)? It's my understanding that the Neuropixels shank was implanted and in the same place across these schedules, but were the spikes sorted together? Or were cells sorted separately in each schedule? Unless I have missed something, it seems this latter method would lead to double-counting of the same cells.

To clarify: each animal had a total of 8 recording sessions. In the first 4 sessions we recorded from bank0 (distal part of the probe), then the animals received a break, and we continued with the next 4 sessions from bank 1 (more proximal on the probe). The 4 recording sessions that were run consecutively were combined and processed together in kilosort, which meant that cell IDs were maintained across the 4 recordings and prevented double-counting. This also allowed us to track whether the spiking profiles (e.g. amplitude) changed over the 4-session recording period.

M3. In Line 51, the authors claim that each action is encoded by some population of neurons in each region. However, in Extended Data Figure 3, it looks like there are quite a few actions that are not encoded by neurons in any region. (To me, this figure actually highlights that only a subset of actions seem to be encoded for in each region.) Perhaps this can be directly quantified somewhere, either in the main text or the figure legend, to make it clearer that there are neurons encoding these actions.

The Reviewer is indeed correct that several actions were not encoded in the examples in Extended Data Figure 3 (now Supplementary Fig. 6), which were from individual animals and single sessions. To the Reviewer's point specifically: Figure 1d now includes a complete summary of the fraction of cells encoding each action in each cortical region, and the most commonly encoded actions in each area are noted in the figure legend. It also highlights that the distributions of encoded actions were largely similar across cortical regions, and the Equivalence test in Suppl. Fig. 6b confirms this statistically. To address this point more generally, the revised manuscript gives more attention to action encoding results, including showing the overall prevalence of action

encoding (Fig. 1c) and inclusion of examples of action encoding by neurons in each region in Suppl. Fig. 6c. We hope the added data make it clearer that nearly all actions were encoded, and to differing degrees, by subsets of neurons in each region.

M4. Related to the decoding result in Figure 1 - the number of neurons will affect decoding accuracy (e.g. as in Figure 3b-c). However, I could not find the average number of simultaneously recorded cells (or how much this number varied across sessions), or whether this was controlled for when presenting decoding accuracy across regions. Also, given that motor regions encode egocentric movement the strongest, did the authors also observe higher decoding accuracy for actions here as well?

This is a good point, and we now include in the legend for Figure 1e that decoding analyses were restricted to 60 cells per session to match sampling across regions. We also added confusion matrices for each area in a new Supplementary Figure 8, and a new Supplementary Fig. 9 showing how decoding accuracy in each region increased with the number of simultaneously recorded cells. The best decoder performance was achieved when the highest numbers of cells were recorded simultaneously in motor cortex. Visual cortex had the lowest rates of decoding accuracy for a given number of cells, but all regions were well above chance levels. These points are noted in the Results, lines 77-79.

M5. Regarding the model fits - 20 minute sessions are quite short for fitting spike trains to natural behavior data, and the resulting pseudo-R² values are quite low with medians ~ 0.02. While this is to be expected to some degree for natural behavior no matter how long the recordings are, the authors may be able to provide additional support for the encoding properties they present if they are indeed tracking cells across schedules (or even across light1 and light2 sessions). In this case, are the encoding properties (variables encoded and their tuning) generally similar? If not, the data presented in Extended Data Figure 5 is relevant and convincing; however, there are only 4 example cells shown. How often are the tuning curves similar across session halves?

This is a fair point of concern, and was raised by other reviewers as well. There is no definition of what a “high” or “low” pseudo-R² is, so we took a suggestion from Reviewer to create a “Peer Model” GLM, in which spiking activity from simultaneously recorded cells (separated by >5 recording sites on the probe) was used as a predicted feature (as done by Stringer et al. (2019)). We then compared the proportions of cells tuned to at least one behavioral covariate in our original GLM (referred to here as the “Covariates Model”) against the subset of cells with “good” pseudo-R² values, *i.e.* larger than their corresponding Peer Models. We found that the defining regional differences in encoding properties were upheld when only considering the cells with “good” pseudo-R² values. That is, head movement, head posture and self-motion still dominated in visual cortices; allo- and egocentric head posture still dominate in auditory cortex; motor regions still encoded the trunk and head, and somatosensory cortex was still dominated by trunk representation:

Proportion of tuning for all selected cells vs. with “high” R²

We also found that the mean rLLRs (importance) of each feature in the polar plots were also maintained when comparing the full dataset against models with “good” pseudo-R² values:

To address the question of tuning curve stability, we calculated Pearson's R-values across even and odd minutes using tuning curves for whichever feature had the highest mutual information for that cell's spiking activity. This analysis included all cells, regardless of whether they appeared to show genuine tuning or not. We compared even-odd minute R-values against the distributions of values produced with shuffled data, and counted cells as "stable" if they exceed the 95th percentile of the shuffled distribution (this explanation is included in the Methods, lines 502-506). Tuning curves from 30% of cells in visual cortices exceeded this criterion; 37% in auditory cortex; 43% in somatosensory, and 66% in motor cortex. We include the fraction of cells with stable tuning curves in each cortical area in the legend of Figure 2.

We wish to point out that the criteria for a cell to have a "stable" tuning curve were more strict than for model selection by the GLM. This was due to practical differences in the way the analyses were constructed, including (i) tuning curves had 36 bins, whereas 1D features in the GLM consisted of 15 bins, (ii) we only considered one 1D covariate to test if a cell was stable, whereas the GLM considered all covariates, (iii) the GLM used 10-fold cross validation (train on 90%, test on 10%), whereas tuning curve stability was across session halves (testing 50% against 50% of sampled data).

M6. Regarding the result regarding the addition of a weight - in the main text, it seems that the addition of the weight did not change the encoding properties of motor cortex cells. However, in extended data fig 10, it is difficult to see the extent to which this is the case. In particular, while the proportion of cells encoding different features is similar across the population in panel G, it is unclear whether other aspects of the encoding, such as the weights or log-likelihoods, are similar. Perhaps a more straightforward comparison would be to train on a light1 session, test the model on the

weighted and light2 session, and then compare the resulting model fits. Further, one could directly compare the glm tuning for models fit with the same variables in each session. For what it is worth, I think this potentially a very interesting result, since this relates to ongoing arguments of encoding kinematics versus dynamics in motor cortex (which could also be referenced more directly in the text).

We are glad the Reviewer found the result of the weight vs. no weight experiments interesting. As noted above, we included examples of tuning curves from cells encoding postural features of the head to better convey the minor effects the added weight had on the cells' tuning (Supplementary Fig. 16b). Further, we performed new GLM analyses in which models were trained on data from the "light 1" session and tested on covariates in the "weight" and "light 2" sessions. The proportions of cells encoding ego- and allocentric head features were similar across weight and weight-free conditions, differing from 0.1% (for egocentric head posture) to 3.6% (for allocentric head posture)-- shown in Supplementary Figure 17, and noted in lines 150-153 of the Results. We also added background and discussion on these experiments, and the debates which motivated them in the Discussion, lines 252-259.

M7. Regarding the results in Figure 3 - currently, much more attention is paid to the overlap in sensory and behavioral variables in visual and auditory cortices than motor and somatosensory cortex. While I somewhat understand the logic for looking at sensory variables in sensory cortex, this has the unfortunate effect that the main results seem to be that visual cortex responds to visual stimuli, and auditory cortex responds to auditory stimuli. It's unclear what we've learned from this result, given that this would be expected. However, given that authors recorded from somatosensory and motor cortex during these trials as well, it might be interesting to include all four regions in these analyses and discuss the differences and similarities. For example, does the behavioral encoding in motor or somatosensory neurons depend on light/dark conditions? Are neurons in these regions modulated by luminance/sound and/or behavior?

Figure 3 in the original manuscript (Figure 5 in the revision) has been re-organized to emphasize the main result, that sensory and behavioral modulation overlap to similar extents in visual and auditory cortices, and in RS and FS neurons. The analyses demonstrating the specificity of visual tuning in visual cortex, and auditory sensitivity in auditory cortex, have been moved to a new Supplementary Fig. 19. Characterization of the overlap of sensory and behavioral tuning in Figure 5 is followed by analyses investigating *how* behavioral signals are integrated in each area in Figure 6.

To address the question of sensory modulation (at least for luminance) in all cortical regions, we performed new GLM analyses to quantify behavioral tuning across subsequent light-light and light-dark recordings in each cortical region. Neither motor, somatosensory nor auditory cortices differed in stability across the recording conditions, but tuning in visual cortex was significantly less stable across light-dark than light-light conditions (p-values calculated with a two sample Z-test for proportions). These findings are reported on lines 188-195 of the Results; the GLM results for auditory and visual cortices are shown in Supplementary Figure 20. We did not have the recordings to test whether other cortical regions were sound modulated.

M8. Extended Data Figure 12b does not include p-values for the changes in encoding properties of neurons in the light and dark conditions. Do you think that the results reflect an actual change in the encoding of movement features, or that the actual

features that the neurons are responding to are visual in nature and are only correlated with movement in the well-lit condition?

This is an excellent question, and (as stated above) we generated p-values for more rigorous comparisons of light-dark tuning changes in visual cortex. As a methodological note, we sought to simplify the question of whether apparent-behavioral tuning reflected the movement of the animal vs. movement of the visual scene by using recordings in darkness. Previous work by Guitchounts et al. 2020 (Neuron) and Bouvier et al. 2020 (Neuron) reported that luminance conditions dramatically modify the activity of visual cortical neurons during movement, and an even more recent paper by Parker et al., 2022 (Neuron) showed multiplicative encoding of head and eye position with visual features in the environment. Since quantifying interactions of visual scene movement and head movement would require eye tracking and gaze reconstruction, which we did not perform, we cannot speculate about this potential (and intriguing) contribution to instability across light and dark conditions in our experiments.

M9. There is a thread regarding FS/RS cells that I am not following. I think that part of the confusion is that FS/RS breakdowns in Figure 3 are not mentioned in the text, and then FS/RS properties relating to Figure 4 are mentioned in the discussion, but not mentioned in Figure 4 or the related main text in the results (unless I missed something). Further, it's not clear to me what the main results are regarding FS/RS cells. For example, in Figure 3, both types of cells seem to have very similar properties. What is the result that the authors are trying to highlight here?

We have revised the text and former Figure 3 (now Figure 5) to more clearly convey the main results: that the amount of overlap between sensory and behavioral signals was uniform in auditory and visual cortices, and occurred to similar relative extents in FS and RS neurons in each area. To this end, we also include the fractions of FS and RS neurons encoding behavior or sensory features in the dendrogram in Figure 5d.

Whereas the point of Figure 5 was to highlight the striking uniformity of sensory and behavioral overlap in visual and auditory regions, the point of Figure 6 was to uncover *how* behavioral signals might be utilized in these regions by considering functional connectivity (elaborated below). We also clarify in the text of the Results that inhibitory connections in the functional analyses (pertaining to Figure 6) were driven by FS neurons, and that excitatory connections were driven by RS neurons.

M10. In Figure 4, the take home message to me is not clear. I think perhaps part of the issue is that it is unclear what should be expected here, or how the results presented specifically update our knowledge of functional connectivity in visual and auditory cortex. I think additional text is needed to motivate and contextualize these results. There has been prior work on functional connectivity in at least visual cortex (Ko*, Hofer* et al 2011), which can be discussed here. Further, it is unclear what the functional space embeddings are supposed to convey.

Having explored the type and extent of behavioral modulations in three separate sensory cortices in the first two figures of the original manuscript, the final two figures (now Figures 5 and 6) are an attempt to connect those findings to the sensory functions of these areas, to motivate future investigations into how behavioral modulation assists sensory processing. As opposed to Figure 5, where the focus is on

determining *tuning overlap*, in Figure 6 we want to focus on *communication between cells*, as tuning overlap *per se* does not characterize whether, and if so, how information from behaviorally relevant signals flow within each network. Prior work on functional connectivity in the primary visual cortex (as in Ko*, Hofer* et al 2011) indeed established a relationship between connection probability and tuning similarity, where units with preferences for orthogonal stimuli had lower connection probabilities. In much the same vein, we wanted to learn whether functionally similar units exhibit greater likelihoods of forming synaptic connections. In our case, however, the functional subtypes could not be categorized based on firing rate differences along a variable that was essentially one-dimensional. Since we had numerous functional categories, each exhibiting directionality, it was not possible to visualize this plainly. Forming the functional space embeddings (*via* dimensionality reduction of filters emanating from our modeling analyses) was an attempt to visualize a potential relationship between connection probability and tuning similarity, a relationship which we ultimately show does not exist statistically, in the lower part of Supplementary Fig. 21e (no correlation between synaptic strength and functional distance).

The revised text motivates the analysis with a stronger introduction and reference to the earlier work on functional connectivity in visual cortex (Ko et al., 2011; Cossell et al., 2015; Lee et al., 2016), and refers to the lack of relation between functional similarity and synaptic strength at the end of the section. We felt this was important to report since it suggests that communication between functionally disparate cells is important, possibly even more so when the content (behavioral vs. sensory signals) is different.

a. In addition, it is not immediately clear to me why the focus is on auditory and visual cortex. The results in Extended Data 13 F-G could be compared to those in Figure 4 C-D. For example, does the increased precedence of Po->Mo excitatory connections in visual and somatosensory cortices as opposed to auditory and motor cortices align with a model similar to those described in the caption of Figure 4?

Our focus on primarily reporting results of auditory and visual areas in the main figure, versus those of other cortices, stemmed from our ability to quantify some aspects of sensory processing in these regions, which we believe would potentially be interesting to a larger audience, but also the desire to not over-complicate what is already a result-heavy figure. As for the similarities in excitatory Po->Mo connections in visual and somatosensory cortices—though the functions of such connections have not been tested empirically— we speculate that they might serve to prime sensory networks for impending movement. In visual cortex, as we noted, a cell encoding maximally rightward head posture would excite a cell encoding leftward head movement, which may prime circuits to anticipate self-generated changes in the visual scene. In somatosensory cortex, such connections could anticipate whether a given change in posture or stance was generated by the individual or, if not, signal postural instability, for example when an animal encounters an unexpected obstacle or change in footing (Nandakumar et al., 2021; Takakusaki, 2017). We note this possibility for somatosensory cortex in the Discussion, lines 298-300.

Minor comments:

m1. There were a few tiny typos:

- a. Line 442 – should be 'represent'
- b. Line 450 – 'features' is misspelled
- c. Line 454 -- there is an extra parentheses
- d. Line 65 – should be 'nearly all'

We thank the Reviewer for catching these typos; all have been corrected.

m2. Line 51 - it is not completely clear what the numbers in the parentheses refer to. Is this the number of neurons that encode at least behavior in each region? I think the wording should be changed, or text added, so this is more transparent.

We have modified the text to more clearly indicate which numbers indicate encoding rates in which cortical region (lines 64-66):

“We found stable encoding of nearly every considered action by individual neurons in each cortical region (51% of neurons in visual cortex, 55% in auditory, 58% in motor and 56% somatosensory regions; Fig. 1c), with most cells responding to multiple actions, and fewer to single actions (Supplementary Fig. 6c; Supplementary Videos 1-6; Methods)”

m3. To help compare the number of simultaneously encoded variables across regions, the axis of the bar plot in Figure 2d should be the same as the other panels.

The axis of the bar plot in revised Figure 3d has been adjusted to match those in 3a-c.

m4. Lines 94-98, Lines 118-122 - the authors report a gradient in encoded features, but it is unclear to me whether the observed gradient is surprising or fits with the current literature. I think a sentence or two contextualizing these results would help.

We have added contextualizing sentences in the Results and Discussion for the topography results. In lines 122-124 we point out that previous studies characterizing head-motion tuning in visual cortex (Velez-Fort et al., 2018; Bouvier et al. 2020; Guitchounts et al., 2020) focused only on V1, so it had not been tested whether gradients exist for behavioral features across visual and neighboring regions. For somatosensory and motor cortices, classical microstimulation experiments in anesthetized rats (Hall & Lindholm, 1974; Neafsy et al. 1986) showed an anterior-to-posterior topography of the head and back from motor to somatosensory cortex. Posture and movement tuning in our study followed a consistent pattern. We also acknowledge this in the Discussion, lines 270-272.

m5. Either the legend or the panel in Fig 3a is not correct – the legend says that the FS spikes are in green, but I do not see any no green spikes.

We have adjusted the wording (and color coding, for clarity) in the figure legend for what was shown in Figure 3a (now in Supplementary Fig. 19a):

“Supplementary Fig. 19: Modality specificity and decoding using auditory and visual cortical units. a, (Top) Spike rasters (grey) and peri-event time histograms (PETHs) of sound-suppressed (left) and sound-excited (right) auditory cortical single units (purple line and shading depict trial averaged firing rate ± 3 SEM for regular spiking (RS) units; and the same for fast spiking (FS) units is shown in magenta; sound stimulation in grey shading). (Bottom) Non-linear embedding of A1 population vector activity from light and dark recording sessions for an example rat (#26525); (right) same but for visual units.”

m6. Extended Data Fig 3 - it seems that panel c has actions in a different configuration than the rest of the panels? This makes it difficult to visually compare with the others.

We apologize for the confusing difference in configurations in the t-SNE embedding in the Extended Data figure. We have rearranged the embedding in the new Supplementary Figure 4 to match that of Figure 1.

m7. Extended Data Fig 6 - the x-axis should be labeled as 'relative log-likelihood' – just listing 'cross validated log-likelihood' had me confused for a little bit.

The x-axis has been corrected, we thank the reviewer for catching the error.

m8. Extended Data Fig 7 - it would help if all were plotted on the same axis.

The axes in the figure (now Supplementary Fig. 12) have been made the same.

m9. Extended Data Figure 13 – I believe the colors describing panels A and B in the caption are incorrect.

The caption has been corrected in the legend (now Supplementary Figure 21), we thank the reviewer for catching the error.

m10. Lines 418-422 - it is unclear how the two halves of the split-half analyses were obtained.

The description in the Methods (lines 534-536) has now been revised to state that the split-half analyses were obtained using odd vs. even time bins within each action:

“Additionally, the average firing rate of the cell was computed separately in two halves of the dataset: the odd vs. even time bins within each action.”

REVIEWER COMMENTS

Reviewer #2 (Remarks to the Author):

The revised manuscript has improved dramatically in clarity of presentation and message. In particular, the new attention to representing the anatomical locations and tuning properties of cells (e.g., new Fig. 4, supplementary figure 13) adds substantially to the interpretability of the data and results. Additionally, the modifications to Fig. 6 (former Fig. 4) are a large improvement. I had missed the significance of the point about heterogeneous connectivity with respect to tuning on the first read-through, and it is an interesting observation, especially (as the authors point out) in comparison to the Ko, Hofer results in mouse V1. Finally, I appreciate the additional analysis provided in the rebuttal of the rLLRs among high-pseudo- R^2 cells, and agree that the original analysis in the paper can remain as-is.

I have two very minor comments, below.

The authors fully addressed my questions about area vs layer specificity, to the extent that is possible from this dataset. One small request regarding Figure 4 (in the new MS) is to show the less-represented categories for each brain area in a supporting figure (e.g., egocentric head [red] for visual and auditory areas; planar motion [green] for motor and somatosensory).

Line 154-155: Did you mean to say "It had the largest overall proportion of unclassified units (51%)..." rather than smallest?

Reviewer #3 (Remarks to the Author):

Summary:

I greatly appreciate the authors' efforts in addressing our comments. I am mostly satisfied with their responses and believe the revised submission is an improvement over the previous submission. However, there are a few (4) outstanding issues that I believe should be addressed prior to publication.

Major comments:

1. While the dataset collected by the authors is impressive, the range of topics, experiments, and analyses covered is extremely broad. As a result, a cohesive message is still lacking. While this is

somewhat mitigated by the fact that the data and the results shown are very interesting and valuable even without a clear message, I believe including statements that better link the first half (Figs 1-3) and the second half (Figs 4-6) would help.

2. At several points in the manuscript there is speculation regarding the interpretation of results and their implications. While I suspect that these statements were likely intended to contextualize the results in response to the first round of comments, some extend beyond the scope of the study and should be reworked.

a. Line 28-29: "... in ways that could facilitate modality-specific functions, such as motion processing in visual cortices or sound localization in auditory cortex." Notably, as differences in sensory input cannot be ruled out to mediate the correlation of movement and postural variables with neural activity in visual and auditory cortex, it seems premature to conclude that the integration of these variables facilitates the proposed functions, particularly in the case of visual cortex. This point is supported by the authors' observation that the tuning of neurons in visual cortex changes between light and dark conditions. Walking back some of the related statements, or being more upfront with the caveats in the experiments, will help address this issue.

b. Lines 313-319: It is hard to tell whether the authors are suggesting that neurons in auditory cortex encoding posture and movement variables influence the detection and processing of the ILD (a function typically attributed to the midbrain), or simply that one must incorporate posture and movement to localize in 3D space using ILD. Regardless, while I agree that the literature on the influence of pose on auditory localization in rats is sparse, work implicating neural activity in auditory cortex to sound localization in other species could be cited to support the authors' argument.

c. Line 255-257: While this statement might have been included in response to a previous comment of mine, I believe this statement should actually be adjusted to focus on the finding that kinematic and postural encoding was not affected by the inclusion of a weight rather than suggesting that neural activity more closely resembles kinematic variables. There is a brief literature explaining why regression analyses like those performed in this manuscript are typically insufficient to distinguish between cortical control of force or end-point kinematics. I've included some examples below.

Todorov, E. Direct cortical control of muscle activation in voluntary arm movements: a model. *Nat Neurosci* 3, 391–398 (2000)

Churchland MM, Shenoy KV. Temporal complexity and heterogeneity of single-neuron activity in premotor and motor cortex. *J Neurophysiol.* 2007 Jun;97(6):4235-57. doi: 10.1152/jn.00095.2007. Epub 2007 Mar 21. PMID: 17376854.

3. I had some difficulty understanding the take-home message in Figure 4 and how it relates to the messages relayed in the main text. Overall, this figure seems to have been directly ported over from the Supplemental Figures, but with not quite enough rearrangement of the text to accommodate it. Some additional text that describes and contextualizes this figure would be extremely appreciated - the specific issues that I ran into are detailed below.

a. First, Figure 4 is barely mentioned in the text, which makes its inclusion as a main figure confusing. Fig 4a and Fig 4b are also introduced separately, which added to this confusion. Overall, adding more context and pointers to this figure will help.

b. Fig 4a is introduced in regards to head motion in visual cortex, but half of the panel is about planar body motion. If this distinction is relevant, it should be discussed in the text.

c. In Figure 4b, it is introduced with the general finding that egocentric head position and movement is encoded in motor cortex while back-related variables are encoded in somatosensory cortex. While this is clear from the middle-left panel of 4b, it's not clear what the other panels of 4b are meant to indicate.

d. More generally, the pie charts denoting the proportion of classified units in 4a or 4b are not referenced as far as I can tell. Do the authors mean to make a point about the gradient of classified units using this data? It seems like there could also be a gradient in the percentage of classified units in the motor cortex. Do the authors want to make a point about this?

e. Is there a main message we should draw from the far-right subplots that is distinct from the middle-left subplots?

4. Unfortunately, I still had difficulty with Figure 6. While I agree that, in general, examining whether functionally-connected cells have similar function properties is interesting, the results here seem to be a bit unclear or speculative. Perhaps the most feasible way to address this issue is to add more text describing what these results mean and what hypotheses they support or falsify, and to clarify some aspects of this analysis.

a. First, there is little explanation for why inhibitory (and not excitatory) Mo->Po and Mo->Sm connections may be more likely in auditory cortex. It is similarly unclear why excitatory Po->Mo and inhibitory Mo->Po connections may be more likely in visual cortex. Why might the system be organized specifically like this as opposed to any of the multitude of other ways one could combine motor and sensory information in these regions? Are there specific theories or previously-proposed mechanisms that could be supported or falsified? Perhaps if the authors provided some example potential circuit motifs, or further interpreted the ones found in light of what is known about circuit motifs in these regions, that would help.

b. In examining this figure further, I also realized that I was confused about the statistics for panel C. Specifically, I cannot figure out how the null distributions of connections between different functional cell types are generated based on the description in the Methods. I have two recommendations for this analysis, or perhaps two guesses as to how the authors may have generated the distribution.

Idea 1: If we consider the set of functional synaptic connections, one could create a null distribution by randomly permuting the functional category (Mo, Po, etc.) of the postsynaptic neurons. This would generate a null distribution of functional connections between populations of neurons that contain the same proportions of functional types. Accordingly, a significance test against this distribution would estimate the probability that the number of synaptic connections between functional categories observed in the real data occur in a population with a similar proportion of functional categories.

Idea 2: Using the Poisson spike generation used to determine whether a pair is significantly functionally connected (in panel A), one could generate a full null dataset of spike trains for all cells in the dataset, which would comply with the assumption that cells are not connected. One could then compute the expected number of pairs picked up by their analysis, and repeat 1000 times to generate a null distribution of pairs.

c. Just a note - the influence of sensory and motor neurons in visual/auditory cortex seems small, while the effects in the motor and somatosensory cortices shown in the supplement seem larger and much easier to interpret. It is still unclear to me why these results are not as thoroughly discussed.

Minor comments

1. Figure 2 - It is currently unclear how predictive the features are for neural activity for the neurons shown. I think adding pseudo-R² values for these neurons will help readers build intuition for that measure.
2. Figure 6 - I believe the dashed boxes in panel c should be removed.

We thank the editors for the invitation to submit a further revised draft of our manuscript, "Behavioral decomposition reveals rich encoding structure employed across neocortex" for publication in Nature Communications. We were glad to have addressed nearly all prior concerns from the Reviewers and appreciated their constructive suggestions for ways in which the manuscript could be further improved. Following their commentary, the most substantial changes in the current revision include (i) adding a subsection in the Results related to Figure 4 (anatomical topography), (ii) reformulating and adding statements to better link experiments in first and second half of the paper, (iii) revising wording to better clarify what can/not be interpreted of our findings with regard to sensory coding, (iv) placing our results and interpretation for the data in auditory cortex in context with other species, and (v) giving more text and discussion to functional connectivity results in somatosensory and motor cortices. We feel that these changes better convey the overall message of the article: that by studying animal behavior closely and at multiple levels, we can go beyond the observation that natural actions are encoded potentially globally in cortex, and start to understand how different systems utilize those signals, presumably in service of local sensory and motor computations.

Point-by-point responses to the concerns raised are listed below in blue:

Reviewer #2 (Remarks to the Author):

The revised manuscript has improved dramatically in clarity of presentation and message. In particular, the new attention to representing the anatomical locations and tuning properties of cells (e.g., new Fig. 4, supplementary figure 13) adds substantially to the interpretability of the data and results. Additionally, the modifications to Fig. 6 (former Fig. 4) are a large improvement. I had missed the significance of the point about heterogeneous connectivity with respect to tuning on the first read-through, and it is an interesting observation, especially (as the authors point out) in comparison to the Ko, Hofer results in mouse V1. Finally, I appreciate the additional analysis provided in the rebuttal of the rLLRs among high-pseudo- R^2 cells, and agree that the original analysis in the paper can remain as-is.

We are happy to have addressed the concerns of the reviewer and thank them for their time and effort to improve the quality and clarity of the manuscript.

I have two very minor comments, below.

The authors fully addressed my questions about area vs layer specificity, to the extent that is possible from this dataset. One small request regarding Figure 4 (in the new MS) is to show the less-represented categories for each brain area in a supporting figure (e.g., egocentric head [red] for visual and auditory areas; planar motion [green] for motor and somatosensory).

The features included in Figure 4 were the only ones that showed significant topographical variation over the length of the recording probes. The lesser-represented features were not topographically organized in either the left or right hemisphere animals, so we did not generate figures for them. This point is now clarified in the subsection for Figure 4 in the Results (see response below to comment 3 from Reviewer 3).

Line 154-155: Did you mean to say "It had the largest overall proportion of unclassified units (51%)..." rather than smallest?

We thank the Reviewer for catching the typo, it has been corrected.

Reviewer #3 (Remarks to the Author):

Summary:

I greatly appreciate the authors' efforts in addressing our comments. I am mostly satisfied with their responses and believe the revised submission is an improvement over the previous submission. However, there are a few (4) outstanding issues that I believe should be addressed prior to publication.

Major comments:

1. While the dataset collected by the authors is impressive, the range of topics, experiments, and analyses covered is extremely broad. As a result, a cohesive message is still lacking. While this is somewhat mitigated by the fact that the data and the results shown are very interesting and valuable even without a clear message, I believe including statements that better link the first half (Figs 1-3) and the second half (Figs 4-6) would help.

The range of experiments and data in the paper are indeed broad, and efforts were made in the previous revision to clarify how the experiments in each section are linked. We appreciate the Reviewer pointing out where these links needed further work and have adjusted different sections of the text. (i) We rewrote the last paragraph of the Introduction to more linearly convey how the results from one experiment were linked with the next and added a synthesis-statement to tie them together. (ii) We added text in the Results between the subsection for Figure 3 and a new sub-section for Figure 4 to state how the objectives in the first and second halves on the paper are connected. That is, Figures 1-3 establish *that* behavior and postural primitives are encoded throughout sensory and motor areas, whereas Figures 4-6 seek to place those findings in context by showing how the observed tuning was organized across regions, and rationalizing how the observed signals could support sensory processing where they were observed.

(i) The last paragraph of the Introduction now reads:

Here, we sought to leverage such advances to determine the extent to which momentary behavior was represented across four major sensory and motor cortical regions, and whether the coding was uniform or varied depending on the region in which it occurred. Whereas naturalistic actions were encoded ubiquitously in sensory and motor areas alike, finer-grained features of pose and movement varied from one region to the next, following extended topographies that overlaid neighboring areas. In relation to sensory processing, pose and movement signals were integrated to similar degrees among sound- and light-sensitive cell populations in auditory and visual cortices, but closer analyses of putative synaptically-connected pairs of neurons suggested different uses of pose and movement signals in each region. Thus, by considering neural tuning to behavior at different levels of complexity and among functionally connected neurons, we show that action representation may be a global feature of sensory and motor cortical systems, but that different regions preferentially encode different physical aspects of posture and motion, presumably to support locally unique computations during active sensing and movement.

(ii) We also made a standalone subsection for Figure 4 in the Results in response to comment 3 (below), which opens with statements to link the 1st and 2nd half of the Results:

Topographical mapping of behavioral features across cortical regions

Up to this point in the study, we found that discrete, naturalistic behaviors were represented throughout the cortical areas recorded, and were composed of simpler posture and movement primitives whose expression appeared to vary regionally. We therefore next sought to establish in the next series of experiments (i) how the neural coding of pose and movement was organized within and between cortical regions, (ii) the extent to which pose and movement signals were integrated with sensory inputs in sensory cortices, and (iii) how pose and movement signaling might be utilized in the different areas.

2. At several points in the manuscript there is speculation regarding the interpretation of results and their implications. While I suspect that these statements were likely intended to contextualize the results in response to the first round of comments, some extend beyond the scope of the study and should be reworked.

a. Line 28-29: "... in ways that could facilitate modality-specific functions, such as motion processing in visual cortices or sound localization in auditory cortex." Notably, as differences in sensory input cannot be ruled out to mediate the correlation of movement and postural variables with neural activity in visual and auditory cortex, it seems premature to conclude that the integration of these variables facilitates the proposed functions, particularly in the case of visual cortex. This point is supported by the authors' observation that the tuning of neurons in visual cortex changes between light and dark conditions. Walking back some of the related statements, or being more upfront with the caveats in the experiments, will help address this issue.

As the Reviewer points out, those interpretations of the results intended to offer context in response to the previous round of review. We have stepped back in the current revision on naming modality-specific perceptual processes which we did not test (e.g. visual motion processing, sound localization) in the Abstract and Introduction. We still feel it is important to explore those possibilities in the Discussion, however, where there is space to weigh both interpretations and the limitations of the experiments.

The 2nd to last sentence of the Abstract was changed to:

*"The tuning properties of synaptically coupled cells also exhibited connection patterns suggestive of area-specific uses of pose and movement signals, **particularly in visual and auditory regions.**"*

Also, as noted above, the last paragraph of the Introduction was changed and no longer speculates about which sensory processes the observed tuning could underlie. We also added the following statement in last paragraph of the Discussion:

*The widespread expression of such features speaks to the computational demand and importance of keeping sensory systems, and the individual, oriented while moving through complex environments, and may therefore reflect a general property of sensory coding in cortex. **Although we did not test how behavioral and sensory information are combined here, our observations could guide future work investigating how pose and movement signals inform perceptual functions like visual self-motion subtraction or sound localization. Pinpointing exactly how this integration happens at the circuit level may prove challenging in freely-behaving subjects, requiring sufficiently resolved techniques, such as miniature 2-photon imaging (Zong et al., 2021; Zong et al., 2022) and holographic stimulation (Marshall et al., 2019), to identify, then manipulate behaviorally-classified neurons in vivo.***

b. Lines 313-319: It is hard to tell whether the authors are suggesting that neurons in auditory cortex encoding posture and movement variables influence the detection and processing of the ILD (a function typically attributed to the midbrain), or simply that one must incorporate posture and movement to localize in 3D space using ILD. Regardless, while I agree that the literature on the influence of pose on auditory localization in rats is sparse, work implicating neural activity in auditory cortex to sound localization in other species could be cited to support the authors' argument.

Our interpretation was more in line with the latter alternative, that roll and pitch modulation may facilitate 3D localization of sound sources. We added support for this interpretation from observations made in barn owls and Tengmalm's owls; specifically, the ear openings in the heads of these species are offset vertically relative to each other, which has been shown to facilitate sound source localization using vertical ILDs (Norberg, 1978; Moiseff et al. 1989). We noted that it is possible that the mammalian brain also uses roll information in particular to facilitate localization of sound sources in elevation. We further added that auditory cortical neurons in several species of mammal signal source locations over a broad range of spatial locations (Middlebrooks, 2015), so having added modulation by head position or movement within auditory cortex could facilitate locating sound sources relative to the head of the individual. We note that further work is required to test this idea empirically.

The revised passage in the Discussion now reads:

*The roll and pitch of the head might be heavily represented because these features would strongly influence the detection of interaural loudness differences (ILDs) in a 3D environment. If a rat's head is perfectly level, for example, any ILD corresponds to a change in horizontal location, but the animal is blind to changes in vertical localization. Once the head rolls, this changes, and strong roll- and pitch-modulation would facilitate detection of ILDs in vertical space, allowing for 3D sampling (Lauer et al., 2018). **Further work will be needed to bear out if this is the case in rodents, but this interpretation is supported by observations in certain species of owl that have vertically offset ear openings in the head, which has been shown to facilitate sound localization using vertical ILDs (Norberg, 1978; Moiseff, 1989). Although additional recordings are also required to establish how postural signals contribute to sound localization at the level of cortex, auditory cortical neurons in mammals encode sound source locations uniformly and over broad ranges of spatial locations (Middlebrooks et al., 1994; Middlebrooks), 2015, so it is intuitive that added modulation by the position and motion of the head would help discriminate the location and direction of a sound relative to the individual (Wallach, 1940).***

c. Line 255-257: While this statement might have been included in response to a previous comment of mine, I believe this statement should actually be adjusted to focus on the finding that kinematic and postural encoding was not affected by the inclusion of a weight rather than suggesting that neural activity more closely resembles kinematic variables. There is a brief literature explaining why regression analyses like those performed in this manuscript are typically insufficient to distinguish between cortical control of force or end-point kinematics. I've included some examples below.

Todorov, E. Direct cortical control of muscle activation in voluntary arm movements: a model. *Nat Neurosci* 3, 391–398 (2000)

Churchland MM, Shenoy KV. Temporal complexity and heterogeneity of single-neuron activity in premotor and motor cortex. *J Neurophysiol.* 2007 Jun;97(6):4235-57. doi: 10.1152/jn.00095.2007. Epub 2007 Mar 21. PMID: 17376854.

We thank the Reviewer for clarifying this point, and for sharing these useful papers on the limitations of regression analyses in modeling highly interrelated features of motor behavior. We have adjusted the text away from a “kinematics vs. muscle force” distinction, and focus on the observation that kinematic & postural tuning were largely unaffected by the added head weight.

In the Results:

Our recordings also uncovered dense representation of head kinematics, particularly in the deep layers of motor cortex (Supplementary Fig. 13), which presented the opportunity to determine if neural encoding of spatial kinematics [46] changed with the added load of a 15 g weight on the head [47].

In the Discussion:

The robust encoding of head features allowed us to test whether the addition of weight on the head affected how neurons encoded posture or kinematics of the head, a question approached previously in the context of hand and arm movements in primates [46, 47, 71, 72]. We found that the added weight had only minor effects on the tuning properties of M1 neurons (Supplementary Figs. 16, 17), but acknowledge that signals related to sensory feedback, planning or dynamic pattern generation could also have contributed since the animals were moving freely [73, 74].

3. I had some difficulty understanding the take-home message in Figure 4 and how it relates to the messages relayed in the main text. Overall, this figure seems to have been directly ported over from the Supplemental Figures, but with not quite enough rearrangement of the text to accommodate it. Some additional text that describes and contextualizes this figure would be extremely appreciated - the specific issues that I ran into are detailed below.

- a. First, Figure 4 is barely mentioned in the text, which makes its inclusion as a main figure confusing. Fig 4a and Fig 4b are also introduced separately, which added to this confusion. Overall, adding more context and pointers to this figure will help.
- b. Fig 4a is introduced in regards to head motion in visual cortex, but half of the panel is about planar body motion. If this distinction is relevant, it should be discussed in the text.
- c. In Figure 4b, it is introduced with the general finding that egocentric head position and movement is encoded in motor cortex while back-related variables are encoded in somatosensory cortex. While this is clear from the middle-left panel of 4b, it's not clear what the other panels of 4b are meant to indicate.
- d. More generally, the pie charts denoting the proportion of classified units in 4a or 4b are not referenced as far as I can tell. Do the authors mean to make a point about the gradient of classified units using this data? It seems like there could also be a gradient in the percentage of classified units in the motor cortex. Do the authors want to make a point about this?
- e. Is there a main message we should draw from the far-right subplots that is distinct from the middle-left subplots?

We have rearranged the text to make a subsection for Figure 4 like the other figures in the paper, and broke the figure into 2 further sub-panels (a-d). We hope this and the added text in the new section address the questions raised above:

Topographical mapping of behavioral features across cortical regions

Up to this point in the study, we found that discrete, naturalistic behaviors were represented throughout the cortical areas recorded, and were composed of simpler posture and movement primitives whose expression appeared to vary regionally. We therefore next sought to establish in the next series of experiments: (i) how the neural coding of pose and movement was organized within and between cortical regions, (ii) the extent to which pose and movement signals were integrated with sensory inputs in sensory cortices, and (iii) how pose and movement signaling might be utilized in the different areas.

In regard to the first question, we specifically asked whether region-specific differences revealed by the GLM analysis emerged abruptly between areas, or followed continuous topographies that spanned cortical boundaries. The first covariates we considered were allocentric head posture and movement in visual and auditory areas, since they were represented most prominently, and had previously only been studied within V1 (Vélez-Fort et al., 2018; Bouvier et al., 2020; Guitchounts et al., 2020). This revealed a graded increase in allocentric head posture coding that progressed laterally from V1 to V2L and peaked in A2D, ($\chi^2(7)=29.5$, $p=4.8e-5$), as well as a peak in allocentric head movement tuning nearby in V2L ($\chi^2(7)=13.09$, $p=.04$; **Fig. 4a** and **Methods**; data from individual animals shown in **4b**). The representation of planar body motion features (e.g. self-motion and turning direction) also increased laterally across V1 and reached a maximum in deeper cortical layers at the border of V1 and V2L ($\chi^2(7)=18.2$, $p=.006$) (**Fig. 4a, b**; **Supplementary Fig. 13**). Together, these covariates largely accounted for the apex of coding cells around the visual-auditory cortical border (**Fig. 4a, pie charts**), since the lesser represented features (egocentric head pose, movement, back tuning) were not organized topographically ($p > .05$ χ^2 test; not shown). Across S1 and M1, on the other hand, we found continual gradients for egocentric head features and the back (**Fig. 4c**). Specifically, neurons encoding egocentric head posture and movement were more frequent in anterior than posterior M1 or in S1 (posture: $\chi^2(7)=37.7$, $p=1.3e-6$; movement: $\chi^2(7)=106.9$, $p=8.8e-21$), whereas back representations dominated in posterior motor areas and S1HL (posture: $\chi^2(7)=18.8$, $p=.004$; movement: $\chi^2(7)=41.02$, $p=2.9e-7$) (**Fig. 4c, d**). Features related to allocentric coding of the head and planar body motion were not organized topographically ($p > .05$ χ^2 test; not shown), and the total fraction of classified cells was higher in anterior locations (**Fig. 4c, pie charts**), peaking in M1.

The legend for Figure 4 has also been updated accordingly:

Fig. 4: Anatomically organized behavioral tuning gradients span sensory and motor regions. **a**, (Left) Probe positions relative to corresponding coronal atlas sections in all right hemisphere implanted animals; the dotted line marks the border between auditory and visual cortices. (Middle) Percentage of all recorded cells responsive to allocentric head posture, allocentric head movement and planar body motion along the mediolateral axis. (Right) The fraction of cells showing any type of behavioral tuning at a given location is indicated by the grey and black pie charts. **b**, Probes from each animal showing the percentage of cells encoding behavioral features along their length. **c**, (Left) Same as in **a** only for left hemisphere implanted animals, with the dotted line marking the border between primary somatosensory and motor cortices. (Middle) percentage of all recorded cells responsive to egocentric head posture and movement, and back posture and movement along the rostrocaudal axis. (Right) same as in **a**,

but for cells classified along the length of S1-M1. **d**, Same as **b**, but using the probe view along S1 and M1 for each animal.

Discussion

End of 3rd paragraph: This interpretation also fits the somatotopic organization of the trunk and limbs in rats (Hall & Lindholm, 1974; Neafsey et al., 1986), as well as the anatomical gradients we observed for trunk and head features, and the larger overall fraction of classified cells at anterior locations in motor cortex (Fig. 4c, right).

4. Unfortunately, I still had difficulty with Figure 6. While I agree that, in general, examining whether functionally-connected cells have similar function properties is interesting, the results here seem to be a bit unclear or speculative. Perhaps the most feasible way to address this issue is to add more text describing what these results mean and what hypotheses they support or falsify, and to clarify some aspects of this analysis.

a. First, there is little explanation for why inhibitory (and not excitatory) Mo->Po and Mo->Sm connections may be more likely in auditory cortex. It is similarly unclear why excitatory Po->Mo and inhibitory Mo->Po connections may be more likely in visual cortex. Why might the system be organized specifically like this as opposed to any of the multitude of other ways one could combine motor and sensory information in these regions? Are there specific theories or previously-proposed mechanisms that could be supported or falsified? Perhaps if the authors provided some example potential circuit motifs, or further interpreted the ones found in light of what is known about circuit motifs in these regions, that would help.

Inhibitory connections in auditory cortices (both Mo->Po and Mo->Sm) appeared more prevalent than would be expected by chance, sharply contrasting those observed for excitatory connections. The latter of these two sets (Mo->Sm) was likely the same phenomenon observed in prior studies in head-fixed animals (Schneider et al., 2014, 2018), positing a role for inhibition in controlling the effects of self-generated sounds on audition (an interpretation previously noted in the Discussion, lines 307-310). Feedforward inhibition plays a known role in gain control (e.g., Mejias et al., 2014), and might have an upper hand over feedforward excitation in the auditory system, which is responsible for solving challenging problems like sound detection, attribution and localization. We also provided a hypothesis for why movement-tuned neurons might be inhibiting posture-tuned neurons (Mo->Po) and relate this to sound localization in the Discussion (lines 314-318), as head posture informs the auditory system about the position of its sensors relative to the environment.

As for visual cortices, the emergent picture is more complex, given that both excitatory and inhibitory connections are more prevalent than would be expected by chance. We attribute the discrepancy in connectivity patterns between auditory and visual to differences in problem sets these systems are tasked with addressing. The visual system needs to continuously resolve visual flow from head movements (Velez-Fort et al., 2018), and this problem might require both feedforward excitation and inhibition (Bouvier et al., 2020; Guitchounts 2020), with cross-talk between movement and posture-modulated cells. Future experiments with eye tracking could offer an additional layer of behavioral resolution necessary to tease apart this issue in more detail. It is an open question how much of what we are observing in terms of either posture or movement is related to eye kinematics.

We also point out these circuit motifs explicitly in the revised Discussion (e.g. Po→Mo connection types are placed in the text where referenced).

We also sought to better motivate the experiments in Figure 6 in the revision by noting that one of the aims of the analyses is to uncover how feedforward connections differ in each region, which then gives clues as to what functions the connections might fulfill:

*The overlap of sensory and behavioral tuning, while substantial, was not informative as to how behavior-related signals were utilized within the networks, so we sought to characterize which behavioral signals were expressed between putative synaptically-connected cells. **This allowed us to discern whether feedforward excitatory and inhibitory signaling differed in relation to specific types of perceptual processing or motor behavior, and to test whether the likelihood of synaptic connections was higher between similarly tuned cells, as seen previously among orientation-selective neurons in visual cortex (Ko et al., 2011; Cossell et al., 2015; Lee et al., 2016).***

We also include the results for motor and somatosensory connections in the Results, as well as a new paragraph interpreting their possible functions in the Discussion (below).

b. In examining this figure further, I also realized that I was confused about the statistics for panel C. Specifically, I cannot figure out how the null distributions of connections between different functional cell types are generated based on the description in the Methods. I have two recommendations for this analysis, or perhaps two guesses as to how the authors may have generated the distribution.

Idea 1: If we consider the set of functional synaptic connections, one could create a null distribution by randomly permuting the functional category (Mo, Po, etc.) of the postsynaptic neurons. This would generate a null distribution of functional connections between populations of neurons that contain the same proportions of functional types. Accordingly, a significance test against this distribution would estimate the probability that the number of synaptic connections between functional categories observed in the real data occur in a population with a similar proportion of functional categories.

Idea 2: Using the Poisson spike generation used to determine whether a pair is significantly functionally connected (in panel A), one could generate a full null dataset of spike trains for all cells in the dataset, which would comply with the assumption that cells are not connected. One could then compute the expected number of pairs picked up by their analysis, and repeat 1000 times to generate a null distribution of pairs.

The null distribution we present in Fig. 6 and Suppl. Fig. 21 is closely aligned to "idea 1" suggested by the reviewer. In the Methods section (lines 748-753 in the current revision) we write:

"Assessment of whether the connection pair numbers in each category could have been observed by chance was done by subsampling pseudorandomly paired units 1000 times, provided that: (1) the anatomical distance between cells was shorter or equal to the maximal one observed in the true data, (2) there were equal numbers of excitatory and inhibitory connections as in the real data in each run, (3) the connection was physiologically plausible (excitatory/inhibitory connections could only be formed in the RS/FS cell was the presynaptic neuron, respectively)."

The reviewer suggested permuting the functional category of the post-synaptic neuron, but we opted for a broader approach, pseudorandomly subsampling both pre- and postsynaptic neurons in a given pair (with noted constraints), taking sample sizes that matched those empirically observed. The question we wanted to ask with this method was "if we observed biologically plausible but functionally random connections between neurons in our ensembles, how much would the emerging functional connections differ to those we actually observed". Our null distributions reliably trace what would be expected based on category sizes, confirming to us the validity of the approach. We hope this answers the question and suggest leaving the Methods and results of this analysis as-is.

c. Just a note - the influence of sensory and motor neurons in visual/auditory cortex seems small, while the effects in the motor and somatosensory cortices shown in the supplement seem larger and much easier to interpret. It is still unclear to me why these results are not as thoroughly discussed.

Although the connectivity effects in the motor and somatosensory cortices were larger, we found they were more difficult to interpret given the lack of a more elaborate task structure in case of the former, and the lack of additional sensory manipulations for the latter. But the effects were large and differed from the patterns observed in visual and auditory areas, and we note these results now in the 2nd paragraph of the Results section for Figure 6:

Different functional connection subtypes were uncovered in each area (Fig. 6c and Supplementary Fig. 21c-f), with the strongest connections in motor and somatosensory cortices being feed-forward excitation between movement-responsive neurons (movement—>movement), and excitatory (posture—>movement) connections in somatosensory cortex (Supplementary Fig. 21f). Relative to motor and somatosensory areas, visual pairs tended to be more homogeneous, and both visual and auditory synapses were weaker (Supplementary Fig. 21e, top).

As for interpreting these results, we added the following which is now the 2nd to last paragraph in the Discussion:

Lastly, the most abundant connections types in motor and somatosensory cortices supported excitatory cross-talk between movement-modulated neurons, though defining the functions of these connections here is difficult given the lack of a more elaborate task structure for the former, and the lack of additional sensory manipulations for the latter. Previous work in motor areas has shown that weak feedforward connections, in conjunction with synaptic noise, help sustain behavioral variability, as demonstrated in simpler organisms (e.g., Zhang et al., 2020), but the strengthening of such connections with sustained repetition leads to motor outputs being less variable (e.g., Garst-Orozco et al., 2014; Graziano et al., 2002). In that sense, the occurrence of feedforward excitation of similarly tuned cells, potentially in both motor and somatosensory regions, may be a feature of a network mechanism which ensures kinematic stability during different iterations of the same actions. Furthermore, excitatory posture -> movement connections between somatosensory neurons could assist in predicting expected movements for a given gait or starting posture, particularly among neurons encoding trunk or limb status, and signal postural instability when those expectations are violated (Nandakumar et al., 2021; Takakusaki, 2017).

Minor comments

1. Figure 2 - It is currently unclear how predictive the features are for neural activity for the neurons shown. I think adding pseudo-R² values for these neurons will help readers build intuition for that measure.

We have added the pseudo-R² values for each of the examples shown in Figures 2 and Suppl. Figure 10. Since the GLM analysis comes subsequently to when the tuning curves are shown in the paper, we listed the R² values in the Figure legend, where it is explained that they refer to output of the GLM analysis in the next section.

Fig. 2: Tuning curves for posture and movement in superficial or granular layers in each overarching cortical area. **a**, (Top left) Tuning curves from a layer 2/3 visual cortical neuron (Cell 1) preferring upward pitch of the head in allocentric coordinates (all examples recorded in darkness). Data from even and odd minutes of a 20 min recording session are shown adjacently (left and middle), and full-session data are shown to the right; the 99% CI of shuffled data are shown in grey. A total of 30% of visual cortical neurons had even-odd minute tuning curve stability higher than the 95th quantile of shuffled data (Methods). **In relation to the subsequent GLM analyses (in Fig. 3, Supplementary Figs. 11 and 12), the pseudo-R² value for allocentric head pitch for Cell 1 was 0.003.** (Top right) A layer 2/3 visual cortical neuron tuned to right head azimuth velocity in allocentric coordinates; **pseudo-R² value of 0.01**. (Lower left) A layer 4 auditory cortical neuron preferring rightward head roll (**pseudo-R², 0.004**), and (lower right) a L4 auditory neuron tuned to upward pitch of the head in allocentric coordinates (**pseudo-R² of 0.001**). 37% of auditory cortical neurons had tuning curves exceeding the shuffled distribution (as described above). **b**, Somatosensory cortical neurons in layer 2/3 tuned to leftward flexion velocity of the back (top left; **pseudo-R² of 0.03**), and upward pitch of the back (top right; **pseudo-R², 0.02**). 43% of S1 neurons were stable beyond the shuffled data. (Lower left) Tuning curves from layer 2/3 motor cortical neurons preferring upward pitch of the back (**pseudo-R² of 0.06**), and (lower right) left and right head roll relative to the trunk (**pseudo-R² of 0.01**). Tuning curves from 66% of M1 neurons were stable beyond the 95th quantile of shuffled data.

Supplementary Fig. 10: Stable posture and movement tuning curves in deep layers in each cortical area. **a**, (Top left) Tuning curves from a layer 6 visual cortical neuron (Cell 1) preferring leftward movement of the head in allocentric coordinates (all examples were recorded in darkness). As with Figure 2, data from even and odd minutes are shown in the left and middle, and the full session is shown to the right. Grey shading indicates the 99% CI of shuffled data. **For reference to GLM analyses (in Figure 3, Supplementary Figs. 11 and 12), the pseudo-R² value for allocentric head pitch for Cell 1 was 0.04.** (Top right) Stable tuning curves from a visual cortical neuron (layer 6) preferring left head azimuth in egocentric coordinates (i.e., relative to the trunk); **pseudo-R² of 0.13**. (Lower left) A layer 6 auditory cortical neuron showing stable tuning to rightward head roll (**pseudo-R² of 0.05**), and (lower right) another L6 auditory neuron preferring downward pitch of the head in allocentric coordinates (**pseudo-R² of 0.006**). **b**, A Layer 5 somatosensory cortical neurons tuned to upward pitch of the back (top left; **pseudo-R² of 0.07**), and an S1 neuron encoding left flexion of the back (top right; **pseudo-R² of 0.01**); (Lower left) Tuning curves from L5 motor cortical neurons preferring left flexion of the back (**pseudo-R² of 0.06**), and (lower right) leftward roll of the head relative to the trunk (**pseudo-R² of 0.01**).

2. Figure 6 - I believe the dashed boxes in panel c should be removed.

Since the dashed boxes in Fig. 6c highlight connections discussed in the Results and shown in the schematic in Fig. 6d, we instead tried giving additional direction to readers by numbering both the boxes in 6c and the connections to which they refer in 6d. We note the numbering of the examples in the figure legend:

Fig. 6: Synaptic connectivity patterns reveal behavioral information is employed differently across auditory and visual subregions.

d, (Top) The V2L network utilizes excitation and inhibition between postural and movement ensembles (connections 1 and 2, respectively; dashed boxes in c) which could serve to coordinate impending movements with visual flow. (Middle) A1 FS ensembles inherit movement information, which could enable gain-modulation of local sound modulated regular spiking (RS) units (connection 3) in response to self-generated sounds. (Bottom) A2D fast spiking (FS) movement modulated ensembles inhibit gravity-relative, posture-responsive units (connection 4), which could facilitate sound localization.

REVIEWERS' COMMENTS

Reviewer #3 (Remarks to the Author):

I appreciate the authors' responses to the previous comments, and the inclusion of text that more clearly links the results from each half of the paper together. The dataset that the authors have collected, and the analyses they performed, are very valuable and I believe will be of interest to a broad community. I support publication of this paper.